# Beyond Vanilla Variational Autoencoders: Detecting Posterior Collapse in Conditional and Hierarchical Variational Autoencoders

**Hien Dang**
FPT Software AI Center
danghoanghien1123@gmail.com

**Tho Tran**
FPT Software AI Center
thotranhuu99@gmail.com

**Tan Nguyen** [*]
Department of Mathematics
National University of Singapore
tanmn@nus.edu.sg

**Nhat Ho** [*]
Department of Statistics and Data Sciences
University of Texas at Austin
minhnhat@utexas.edu

## Abstract

The posterior collapse phenomenon in variational autoencoder (VAE), where the variational posterior distribution closely matches the prior distribution, can hinder the quality of the learned latent variables. As a consequence of posterior collapse, the latent variables extracted by the encoder in VAE preserve less information from the input data and thus fail to produce meaningful representations as input to the reconstruction process in the decoder. While this phenomenon has been an actively addressed topic related to VAE performance, the theory for posterior collapse remains underdeveloped, especially beyond the standard VAE. In this work, we advance the theoretical understanding of posterior collapse to two important and prevalent yet less studied classes of VAE: conditional VAE and hierarchical VAE. Specifically, via a non-trivial theoretical analysis of linear conditional VAE and hierarchical VAE with two levels of latent, we prove that the cause of posterior collapses in these models includes the correlation between the input and output of the conditional VAE and the effect of learnable encoder variance in the hierarchical VAE. We empirically validate our theoretical findings for linear conditional and hierarchical VAE and demonstrate that these results are also predictive for non-linear cases with extensive experiments.

## 1 Introduction

Variational autoencoder (VAE) (Kingma & Welling, 2013) has achieved successes across unsupervised tasks that aim to find good low-dimensional representations of high-dimensional data, ranging from image generation (Child, 2021; Vahdat & Kautz, 2020) and text analysis (Bowman et al., 2015; Miao et al., 2016; Guu et al., 2017) to clustering (Jiang et al., 2016) and dimensionality reduction (Akkari et al., 2022). The success of VAE relies on integrating variational inference with flexible neural networks to generate new observations from an intrinsic low-dimensional latent structure (Blei et al., 2017). However, it has been observed that when training to maximize the evidence lower bound (ELBO) of the data's log-likelihood, the variational posterior of the latent variables in VAE converges to their prior. This phenomenon is known as the *posterior collapse*. When posterior collapse occurs, the data does not contribute to the learned posterior distribution of the latent variables, thus limiting the ability of VAE to capture intrinsic representation from the observed data. It is widely claimed in the literature that the causes of the posterior collapse are due to: i) the Kullback–Leibler (KL) divergence regularization factor in ELBO that pushes the variational distribution towards the prior, and ii) the powerful decoder that assigns high probability to the training samples even when posterior collapse occurs. A plethora of methods have been proposed to mitigate the effect of the KL-regularization term in ELBO training process by modifying the training objective functions (Bowman et al., 2015; Huang et al., 2018; Sønderby et al., 2016; Higgins et al., 2016;

---

[*]Co-last authors. Please correspond to: danghoanghien1123@gmail.com

Razavi et al., 2019) or by redesigning the network architecture of the decoder to limit its representation capacity (Gulrajani et al., 2017; Yang et al., 2017; Semeniuta et al., 2017; Van Den Oord et al., 2017; Dieng et al., 2019; Zhao et al., 2020). However, the theoretical understanding of posterior collapse has still remained limited due to the complex loss landscape of VAE.

**Contribution:** Given that the highly non-convex nature of deep nonlinear networks imposes a significant barrier to the theoretical understanding of posterior collapse, linear VAEs are a good candidate model for providing important theoretical insight into this phenomenon and have been recently studied in (Wang & Ziyin, 2022; Lucas et al., 2019). Nevertheless, these works only focus on the simplest settings of VAE, which are the linear standard VAE with one latent variable (see Figure 1(a) for the illustration). Hence, the theoretical analysis of other important VAEs architectures has remained elusive.

In this paper, we advance the theory of posterior collapse to two important and prevalently used classes of VAE: Conditional VAE (CVAE) (Sohn et al., 2015) and Markovian Hierarchical VAE (MHVAE) (Luo, 2022). CVAE is widely used in practice for structured prediction tasks (Sohn et al., 2015; Walker et al., 2016). By conditioning on both latent variables and the input condition in the generating process, CVAE overcomes the limitation of VAE that the generating process cannot be controlled. On the other hand, MHVAE is an extension of VAE to incorporate higher levels of latent structures and is more relevant to practical VAE architecture that use multiple layers of latent variable to gain greater expressivity (Child, 2021; Vahdat & Kautz, 2020; Maaløe et al., 2019). Moreover, studying MHVAE potentially sheds light on the understanding of diffusion model (Sohl-Dickstein et al., 2015; Song & Ermon, 2019; Ho et al., 2020), since diffusion model can be interpreted as a simplified version of deep MHVAE (Luo, 2022). Following common training practice, we consider linear CVAE and MHVAE with adjustable hyperparameter $\beta$'s before each KL-regularization term to balance latent channel capacity and independence constraints with reconstruction accuracy as in the $\beta$-VAE (Higgins et al., 2016). Our contributions are four-fold:

1. We first revisit linear standard VAE and verify the importance of learnability of the encoder variance to posterior collapse existence. For unlearnable encoder variance, we prove that posterior collapse might not happen even when the encoder is low-rank (see Section 3).

2. We characterize the global solutions of linear CVAE training problem with precise conditions for the posterior collapse occurrence. We find that the correlation of the training input and training output is one of the factors that decides the collapse level (see Section 4.1).

3. We characterize the global solutions of linear two-latent MHVAE training problem and point out precise conditions for the posterior collapse occurrence. We study the model having separate $\beta$'s and find their effects on the posterior collapse of the latent variables (see Section 4.2).

4. We empirically show that the insights deduced from our theoretical analysis are also predictive for non-linear cases with extensive experiments (see Section 5).

**Notation:** We will use the following notations frequently for subsequent analysis and theorems. For input data $x \in \mathbb{R}^{D_0}$, we denote $\mathbf{A} := \mathbb{E}_x(xx^\top)$, the second moment matrix. The eigenvalue decomposition of $\mathbf{A}$ is $\mathbf{A} = \mathbf{P}_A \Phi \mathbf{P}_A^\top$ with $\Phi \in \mathbb{R}^{d_0 \times d_0}$ and $\mathbf{P}_A \in \mathbb{R}^{D_0 \times d_0}$ where $d_0 \leq D_0$ is the number of positive eigenvalues of $\mathbf{A}$. For CVAE, in addition to the eigenvalue decomposition of condition $x$ as above, we define similar notation for output $y$, $\mathbf{B} := \mathbb{E}_y(yy^\top) = \mathbf{P}_B \Psi \mathbf{P}_B^\top$ with $\Psi \in \mathbb{R}^{d_2 \times d_2}$ and $\mathbf{P} \in \mathbb{R}^{D_2 \times d_2}$ where $d_2 \leq D_2$ is the number of positive eigenvalues of $\mathbf{B}$. Moreover, we consider the whitening transformation: $\tilde{x} = \Phi^{-1/2} \mathbf{P}_A^\top x \in \mathbb{R}^{d_0}$ and $\tilde{y} = \Psi^{-1/2} \mathbf{P}_B^\top y \in \mathbb{R}^{d_2}$. It is clear that $\mathbb{E}_x(\tilde{x}\tilde{x}^\top) = \mathbf{I}_{d_0}$, $\mathbb{E}_y(\tilde{y}\tilde{y}^\top) = \mathbf{I}_{d_2}$ and $x, y$ can be written as $x = \mathbf{P}_A \Phi^{1/2} \tilde{x}$ and $y = \mathbf{P}_B \Phi^{1/2} \tilde{y}$. The KL divergence of two probability distributions $P$ and $Q$ is denoted as $D_{\mathrm{KL}}(P||Q)$.

## 2 BACKGROUND ON VARIATIONAL AUTOENCODERS

**Variational Autoencoder:** VAE represents a class of generative models assuming each data point $x$ is generated from an unseen latent variable. Specifically, VAE assumes that there exists a latent variable $z \in \mathbb{R}^{d_1}$, which can be sampled from a prior distribution $p(z)$ (usually a normal distribution), and the data can be sampled through a conditional distribution $p(x|z)$ that modeled as a decoder. Because the marginal log-likelihood $\log p(x)$ is intractable, VAE uses amortized variational inference (Blei et al., 2017) to approximate the posterior $p(z|x)$ via a variational distribution $q(z|x)$. The

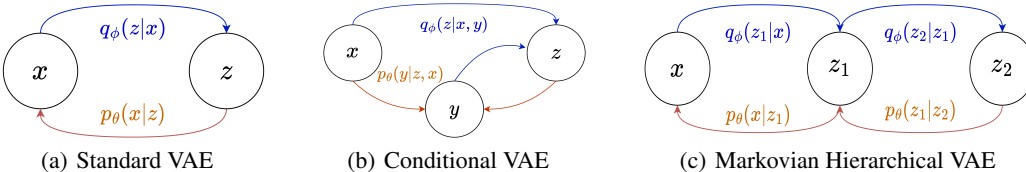

Figure 1: Graphical illustration of standard VAE, CVAE, and MHVAE with two latents.

variation inference $q(z|x)$ is modeled as an encoder that maps data $x$ into the latent $z$ in latent space. This allows tractable approximate inference using the evidence lower bound (ELBO):

$$\text{ELBO}_{\text{VAE}} := \mathbb{E}_{q(z|x)}[\log p(x|z)] - D_{\text{KL}}(q(z|x)||p(z)) \leq \log p(x). \quad (1)$$

Standard training involves maximizing the ELBO, which includes a reconstruction term $\mathbb{E}_{q(z|x)}[\log p(x|z)]$ and a KL-divergence regularization term $D_{\text{KL}}(q(z|x)||p(z))$.

**Conditional Variational Autoencoder:** One of the limitations of VAE is that we cannot control its data generation process. Assume that we do not want to generate some random new digits, but some certain digits based on our need, or assume that we are doing the image inpainting problem: given an existing image where a user has removed an unwanted object, the goal is to fill in the hole with plausible-looking pixels. Thus, CVAE is developed to address these limitations (Sohn et al., 2015; Walker et al., 2016). CVAE is an extension of VAE that include input condition $x$, output variable $y$, and latent variable $z$. Given a training example $(x, y)$, CVAE maps both $x$ and $y$ into the latent space in the encoder and use both the latent variable $z$ and input $x$ in the generating process. Hence, the variational lower bound can be rewritten as follows:

$$\text{ELBO}_{\text{CVAE}} := \mathbb{E}_{q_\phi(z|x,y)}\left[\log p_\theta(y|x,z)\right] + D_{\text{KL}}(q_\phi(z|x,y)||p(z|x)) \leq \log p(y|x), \quad (2)$$

where $p(z|x)$ is still a standard Gaussian because the model assumes $z$ is sampled independently of $x$ at test time (Doersch, 2016).

**Hierarchical Variational Autoencoder:** Hierarchical VAE (HVAE) is a generalization of VAE by introducing multiple latent layers with a hierarchy to gain greater expressivity for both distributions $q_\phi(z|x)$ and $p_\theta(x|z)$ (Child, 2021). In this work, we focus on a special type of HVAE named the Markovian HVAE (Luo, 2022). In this model, the generative process is a Markov chain, where decoding each latent $z_t$ only conditions on previous latent $z_{t+1}$. The ELBO in this case becomes

$$\mathbb{E}_{q_\phi(z_{1:T}|\boldsymbol{x})}\left[\log \frac{p(\boldsymbol{x}, z_{1:T})}{q_\phi(z_{1:T} \mid \boldsymbol{x})}\right] = \mathbb{E}_{q_\phi(z_{1:T}|\boldsymbol{x})}\left[\log \frac{p(z_T)\, p_\theta(\boldsymbol{x} \mid z_1) \prod_{t=2}^{T} p_\theta(z_{t-1} \mid z_t)}{q_\phi(z_1 \mid \boldsymbol{x}) \prod_{t=2}^{T} q_\phi(z_t \mid z_{t-1})}\right].$$

## 3 REVISITING LINEAR VAE WITH ONE LATENT AND MORE

We first revisit the simplest model, i.e., linear standard VAE with one latent variable, which has also been studied in (Lucas et al., 2019; Dai et al., 2020; Wang & Ziyin, 2022), with two settings: *learnable* and *unlearnable* (i.e., predefined and not updated during the training of the model) diagonal encoder variance. For both settings, the encoder is a linear mapping $\mathbf{W} \in \mathbb{R}^{d_1 \times D_0}$ that maps input data $x \in \mathbb{R}^{D_0}$ to latent space $z \in \mathbb{R}^{d_1}$. Applying the reparameterization trick (Kingma & Welling, 2013), the latent $z$ is produced by further adding a noise term $\xi \sim \mathcal{N}(\mathbf{0}, \boldsymbol{\Sigma})$, i.e., $z = \mathbf{W}x + \xi$ with $\boldsymbol{\Sigma} \in \mathbb{R}^{d_1 \times d_1}$ is the encoder variance. Thus, we have the recognition model $q_\phi(z|x) = \mathcal{N}(\mathbf{W}x, \boldsymbol{\Sigma})$. The decoder is a linear map that parameterizes the distribution $p_\theta(x|z) = \mathcal{N}(\mathbf{U}z, \eta_{\text{dec}}^2\mathbf{I})$ with $\mathbf{U} \in \mathbb{R}^{D_0 \times d_1}$ and $\eta_{\text{dec}}$ is unlearnable. The prior $p(z)$ is $\mathcal{N}(0, \eta_{\text{enc}}^2\mathbf{I})$ with known $\eta_{\text{enc}}$. Note that we do not include bias in the linear encoder and decoder, since the effect of bias term is equivalent to centering both input and output data to zero mean (Wang & Ziyin, 2022). Therefore, adding the bias term does not affect the main results. After dropping the multiplier $1/2$ and other constants, the negative ELBO becomes

$$\mathcal{L}_{\text{VAE}} = \frac{1}{\eta_{\text{dec}}^2}\mathbb{E}_x\left[\|\mathbf{U}\mathbf{W}x - x\|^2 + \text{trace}((\mathbf{U}^\top\mathbf{U} + \beta c^2\mathbf{I})\boldsymbol{\Sigma}) + \beta c^2\|\mathbf{W}x\|^2\right] - \beta \log |\boldsymbol{\Sigma}|, \quad (3)$$

where $c := \eta_{\text{dec}}/\eta_{\text{enc}}$. Our contributions are: i) we characterize the global solutions of linear VAE training problem for *unlearnable* $\Sigma$ with arbitrary elements on the diagonal, which is more general than the Proposition 2 in (Wang & Ziyin, 2022) where only the unlearnable isotropic $\Sigma = \sigma^2 \mathbf{I}$ is considered, and ii) we prove that for the case of unlearnable $\Sigma$, even when the encoder matrix is low-rank, posterior collapse may not happen. While it has been known that for learnable $\Sigma$, low-rank encoder matrix certainly leads to posterior collapse (Lucas et al., 2019; Dai et al., 2020). Thus, *learnable latent variance is among the causes of posterior collapse*, opposite to the results in Section 4.5 in Wang & Ziyin (2022) that "a learnable latent variance is not the cause of posterior collapse". We will explain this point further after Theorem 1. Recalling the notations defined in Section 1, the following theorem characterizes the global minima $(\mathbf{U}^*, \mathbf{W}^*)$ when minimizing $\mathcal{L}_{\text{VAE}}$ for unlearnable $\Sigma$ case. In particular, we derive the global minima's SVD with closed-form singular values, via the SVD and singular values of the matrix $\mathbf{Z} := \mathbb{E}_x(x\tilde{x}^\top)$ and other hyperparameters.

**Theorem 1** (Unlearnable $\Sigma$). *Let* $\mathbf{Z} := \mathbb{E}_x(x\tilde{x}^\top) = \mathbf{R}\Theta\mathbf{S}$ *is the SVD of* $\mathbf{Z}$ *with singular values* $\{\theta_i\}_{i=1}^{d_0}$ *in non-increasing order and define* $\mathbf{V} := \mathbf{W}\mathbf{P}_A\Phi^{1/2}$. *With unlearnable* $\Sigma = \text{diag}(\sigma_1^2, \dots, \sigma_{d_1}^2)$, *the optimal solution of* $(\mathbf{U}^*, \mathbf{W}^*)$ *of* $\mathcal{L}_{\text{VAE}}$ *is as follows:*

$$\mathbf{U}^* = \mathbf{R}\Omega\mathbf{T}^\top, \mathbf{V}^* := \mathbf{W}^*\mathbf{P}_A\Phi^{1/2} = \mathbf{T}\Lambda\mathbf{S}^\top, \tag{4}$$

*where* $\mathbf{T} \in \mathbb{R}^{d_1 \times d_1}$ *is an orthonormal matrix that sort the diagonal of* $\Sigma$ *in **non-decreasing** order, i.e.,* $\Sigma = \mathbf{T}\Sigma'\mathbf{T}^\top = \mathbf{T}\text{diag}(\sigma_1'^2, \dots, \sigma_{d_1}'^2)\mathbf{T}^\top$ *with* $\sigma_1'^2 \leq \dots \leq \sigma_{d_1}'^2$. $\Omega \in \mathbb{R}^{D_0 \times d_1}$ *and* $\Lambda \in \mathbb{R}^{d_1 \times d_0}$ *are rectangular diagonal matrices with the following elements,* $\forall i \in [d_1]$:

$$\omega_i^* = \sqrt{\max\left(0, \frac{\sqrt{\beta}\eta_{dec}}{\eta_{enc}\sigma_i'}\left(\theta_i - \sqrt{\beta}\sigma_i'\frac{\eta_{dec}}{\eta_{enc}}\right)\right)}, \lambda_i^* = \sqrt{\max\left(0, \frac{\eta_{enc}\sigma_i'}{\sqrt{\beta}\eta_{dec}}\left(\theta_i - \sqrt{\beta}\sigma_i'\frac{\eta_{dec}}{\eta_{enc}}\right)\right)}.$$

*If* $d_0 < d_1$, *we denote* $\theta_i = 0$ *for* $d_0 < i \leq d_1$.

The proof of Theorem 1 is in Appendix D.1. We note that our result allows for arbitrary predefined values of $\{\sigma_i\}_{i=1}^{d_1}$, thus is more general than the Proposition 2 in (Wang & Ziyin, 2022) where $\sigma_i$'s are all equal to a constant. Under broader settings, there are two notable points from Theorem 1 that have not been captured in the previous result of (Wang & Ziyin, 2022): i) at optimality, the singular matrix $\mathbf{T}$ sorts the set $\{\sigma_i\}_{i=1}^{d_1}$ in non-decreasing order, and ii) singular values $\omega_i$ and $\lambda_i$ are calculated via the $i$-th smallest value $\sigma_i'$ of the set $\{\sigma_i\}_{i=1}^{d_1}$, not necessarily the $i$-th element $\sigma_i$.

**The role of learnability of the encoder variance to posterior collapse:** From Theorem 1, we see the ranks of both the encoder and decoder depend on the sign of $\theta_i - \sqrt{\beta}\sigma_i'\eta_{\text{dec}}/\eta_{\text{enc}}$. The model becomes low-rank when the sign of this term is negative for some $i$. However, low-rank $\mathbf{V}^*$ is *not sufficient* for the occurrence of posterior collapse, it also depends on the sorting matrix $\mathbf{T}$ that sorts $\{\sigma_i\}_{i=1}^{d_1}$ in non-decreasing order. For the isotropic $\Sigma$ case where all $\sigma_i$'s are all equal, $\mathbf{T}$ can be any orthogonal matrix since $\{\sigma_i\}$ are in non-decreasing order already. Therefore, although $\Lambda$ and $\Lambda\mathbf{S}^\top$ has zero rows due to the low-rank of $\mathbf{V}^*$, $\mathbf{V}^* = \mathbf{T}\Lambda\mathbf{S}^\top$ might have no zero rows and $\mathbf{W}x = \mathbf{V}\tilde{x}$ might have no zero component. For the $k$-th dimension of the latent $z = \mathbf{W}x + \xi$ to collapse, i.e., $p(z_k|x) = \mathcal{N}(0, \eta_{\text{enc}}^2)$, we need the $k$-th dimension of the mean vector $\mathbf{W}x$ of the posterior $q(z|x)$ to equal 0 for any $x$. Thus, posterior collapse might not happen. This is opposite to the claim that posterior collapse occurs in this unlearnable isotropic $\Sigma$ setting from Theorem 1 in (Wang & Ziyin, 2022). If all values $\{\sigma_i\}_{i=1}^{d_1}$ are distinct, $\mathbf{T}$ must be a permutation matrix and hence, $\mathbf{V}^* = \mathbf{T}\Lambda\mathbf{S}^\top$ has $d_1 - \text{rank}(\mathbf{V}^*)$ zero rows, corresponding with dimensions of latent $z$ that follow $\mathcal{N}(0, \sigma_i^2)$.

For the setting of *learnable* and *diagonal* $\Sigma$, the low-rank structure of $\mathbf{U}^*$ and $\mathbf{V}^*$ surely leads to posterior collapse. Specifically, at optimality, we have $\Sigma^* = \beta\eta_{\text{dec}}^2(\mathbf{U}^{*\top}\mathbf{U}^* + \beta c^2\mathbf{I})^{-1}$, and thus, $\mathbf{U}^{*\top}\mathbf{U}^*$ is diagonal. As a result, $\mathbf{U}^*$ can be decomposed as $\mathbf{U}^* = \mathbf{R}\Omega$ with orthonormal matrix $\mathbf{R} \in \mathbb{R}^{D_0 \times D_0}$ and $\Omega$ is a rectangular diagonal matrix containing its singular values. Hence, $\mathbf{U}^*$ has $d_1 - r$ zero columns with $r := \text{rank}(\mathbf{U}^*)$. Dai et al. (2017) claimed that there exists an inherent mechanism to prune these superfluous columns to exactly zero. Looking at the loss $\mathcal{L}_{\text{VAE}}$ at Eqn. (3), we see that these $d_1 - r$ zero columns of $\mathbf{U}^*$ will make the corresponding $d_1 - r$ dimensions of the vector $\mathbf{W}x$ to not appear in the reconstruction term $\|\mathbf{U}\mathbf{W}x - x\|^2$, and they only appear in the regularization term $\|\mathbf{W}x\|^2 = \|\mathbf{V}\tilde{x}\|^2 = \|\mathbf{V}\|_F^2$. These dimensions of $\mathbf{W}x$ subsequently becomes zeroes at optimality. Therefore, these $d_1 - r$ dimensions of the latent $z = \mathbf{W}x + \xi$ collapse to its prior $\mathcal{N}(0, \eta_{\text{enc}}^2)$. The detailed analysis for *learnable* $\Sigma$ case is provided in the Appendix D.2.

# 4 BEYOND STANDARD VAE: POSTERIOR COLLAPSE IN LINEAR CONDITIONAL AND HIERARCHICAL VAE

## 4.1 CONDITIONAL VAE

In this section, we consider linear CVAE with input condition $x \in \mathbb{R}^{D_0}$, the latent $z \in \mathbb{R}^{d_1}$, and output $y \in \mathbb{R}^{D_2}$. The latent $z$ is produced by adding a noise term $\xi \sim \mathcal{N}(\mathbf{0}, \mathbf{\Sigma})$ to the output of the linear encoder networks that maps both $x$ and $y$ into latent space, i.e., $z = \mathbf{W}_1 x + \mathbf{W}_2 y + \xi$, where $\mathbf{\Sigma} \in \mathbb{R}^{d_1 \times d_1}$ is the encoder variance, $\mathbf{W}_1 \in \mathbb{R}^{d_1 \times D_0}$, and $\mathbf{W}_2 \in \mathbb{R}^{d_1 \times D_2}$. Hence, $q_\phi(z|x, y) = \mathcal{N}(\mathbf{W}_1 x + \mathbf{W}_2 y, \mathbf{\Sigma})$. The decoder parameterizes the distribution $p_\theta(y|z, x) = \mathcal{N}(\mathbf{U}_1 z + \mathbf{U}_2 x, \eta_{\text{dec}}^2 \mathbf{I})$, where $\mathbf{U}_1 \in \mathbb{R}^{D_2 \times d_1}, \mathbf{U}_2 \in \mathbb{R}^{D_2 \times D_0}$, and predefined $\eta_{\text{dec}}$. We set the prior $p(z) = \mathcal{N}(0, \eta_{\text{enc}}^2 \mathbf{I})$ with a pre-defined $\eta_{\text{enc}}$. An illustration of the described architecture is given in Figure 1(b). We note that the linear standard VAE studied in (Wang & Ziyin, 2022; Lucas et al., 2019) does not capture this setting. Indeed, let us consider the task of generating new pictures. The generating distribution $p(y|z)$ considered in (Wang & Ziyin, 2022; Lucas et al., 2019), where $z \sim \mathcal{N}(0, \mathbf{I})$, does not condition on the input $x$ on its generating process.

Previous works that studied linear VAE usually assume $\mathbf{\Sigma}$ is data-independent or only linearly dependent to the data (Lucas et al., 2019; Wang & Ziyin, 2022). We find that this constraint can be removed in the analysis of linear standard VAE, CVAE, and MHVAE. Particularly, when each data has its own learnable $\mathbf{\Sigma}_x$, the training problem is equivalent to using a single $\mathbf{\Sigma}$ for all data (see Appendix C for details). Therefore, for brevity, we will use the same variance matrix $\mathbf{\Sigma}$ for all samples. Under this formulation, the negative ELBO loss function in Eqn. (2) can be written as:

$$\mathcal{L}_{\text{CVAE}}(\mathbf{W}_1, \mathbf{W}_2, \mathbf{U}_1, \mathbf{U}_2, \mathbf{\Sigma}) = \frac{1}{\eta_{\text{dec}}^2} \mathbb{E}_{x,y}\big[\|(\mathbf{U}_1 \mathbf{W}_1 + \mathbf{U}_2)x + (\mathbf{U}_1 \mathbf{W}_2 - \mathbf{I})y\|^2$$

$$+ \text{trace}(\mathbf{U}_1 \mathbf{\Sigma} \mathbf{U}_1^\top) + \beta c^2(\|\mathbf{W}_1 x + \mathbf{W}_2 y\|^2 + \text{trace}(\mathbf{\Sigma}))\big] - \beta d_1 - \beta \log|\mathbf{\Sigma}|, \quad (5)$$

where $c := \eta_{\text{dec}}/\eta_{\text{enc}}$ and $\beta > 0$. Comparing to the loss $\mathcal{L}_{\text{VAE}}$ of standard VAE, minimizing $\mathcal{L}_{\text{CVAE}}$ is a more complicated problem due to the fact that the architecture of CVAE requires two additional mappings, including the map from the output $y$ to latent $z$ in the encoder and the map from condition $x$ to output $y$ in the decoder. Recall the notations defined in Section 1, we find the global minima of $\mathcal{L}_{\text{CVAE}}$ and derive the closed-form singular values of the decoder map $\mathbf{U}_1$ in the following theorem. We focus on the rank and the singular values of $\mathbf{U}_1$ because they are important factors that influence the level of posterior collapse (i.e., how many latent dimensions collapse), which we will explain further after the theorem.

**Theorem 2** (Learnable $\mathbf{\Sigma}$). *Let* $c = \eta_{dec}/\eta_{enc}$, $\mathbf{Z} = \mathbb{E}_{x,y}(\tilde{y}\tilde{x}^\top) \in \mathbb{R}^{d_0 \times d_2}$ *and define* $\mathbf{E} := \mathbf{P}_B \Psi^{1/2}(\mathbf{I} - \mathbf{Z}^\top \mathbf{Z})\Psi^{1/2}\mathbf{P}_B^\top = \mathbf{P}\Theta\mathbf{Q}$ *be the SVD of* $\mathbf{E}$ *with singular values* $\{\theta_i\}_{i=1}^{d_2}$ *in non-increasing order. The optimal solution of* $(\mathbf{U}_1^*, \mathbf{U}_2^*, \mathbf{W}_1^*, \mathbf{W}_2^*, \mathbf{\Sigma}^*)$ *of* $\mathcal{L}_{CVAE}$ *is as follows:*

$$\mathbf{U}_1^* = \mathbf{P}\Omega\mathbf{R}^\top, \quad \mathbf{\Sigma}^* = \beta\eta_{dec}^2(\mathbf{U}_1^{*\top}\mathbf{U}_1^* + \beta c^2\mathbf{I})^{-1},$$

*where* $\mathbf{R} \in \mathbb{R}^{d_1 \times d_1}$ *is an orthonormal matrix.* $\Omega$ *is the rectangular singular matrix of* $\mathbf{U}_1^*$ *with diagonal elements* $\{\omega_i^*\}_{i=1}^{d_1}$ *and variance* $\mathbf{\Sigma}^* = \text{diag}(\sigma_1^2, \sigma_2^2, \ldots, \sigma_{d_1}^2)$ *with:*

$$\omega_i^* = \frac{1}{\eta_{enc}}\sqrt{\max(0, \theta_i - \beta\eta_{dec}^2)}, \quad \sigma_i' = \begin{cases} \sqrt{\beta}\eta_{enc}\eta_{dec}/\sqrt{\theta_i}, & \text{if } \theta_i \geq \beta\eta_{dec}^2 \\ \eta_{enc}, & \text{if } \theta_i < \beta\eta_{dec}^2 \end{cases}, \; \forall \, i \in [d_1] \quad (6)$$

*where* $\{\sigma_i'\}_{i=1}^{d_1}$ *is a permutation of* $\{\sigma_i\}_{i=1}^{d_1}$*, i.e.,* $\text{diag}(\sigma_1', \ldots, \sigma_{d_1}') = \mathbf{R}^\top \text{diag}(\sigma_1, \ldots, \sigma_{d_1})\mathbf{R}$*. If* $d_2 < d_1$*, we denote* $\theta_i = 0$ *for* $d_2 < i \leq d_1$*. The other matrices obey:*

$$\begin{cases} \mathbf{U}_2^* \mathbf{P}_A \Phi^{1/2} = \mathbf{P}_B \Psi^{1/2} \mathbf{Z}^\top \\ \mathbf{W}_2^* \mathbf{P}_B \Psi^{1/2}(\mathbf{I} - \mathbf{Z}^\top \mathbf{Z}) = (\mathbf{U}_1^{*\top}\mathbf{U}_1^* + \beta c^2 \mathbf{I})^{-1}\mathbf{U}_1^{*\top}\mathbf{P}_B \Psi^{1/2}(\mathbf{I} - \mathbf{Z}^\top \mathbf{Z}) \\ \mathbf{W}_1^* \mathbf{P}_A \Phi^{1/2} = -\mathbf{W}_2^* \mathbf{P}_B \Psi^{1/2} \mathbf{Z}^\top \end{cases} \quad (7)$$

The proof of Theorem 2 is given in Appendix E. First, we notice that the rank of $\mathbf{U}_1^*$ depends on the sign of $\theta_i - \beta\eta_{\text{dec}}^2$. When there are some $i$'s that the sign is negative, the map $\mathbf{U}_1^*$ from $z$ to $y$ in the generating process becomes low-rank. Since the encoder variance $\mathbf{\Sigma}$ is *learnable*, the posterior collapse surely happens in this setting. Specifically, assuming $r$ is the rank of $\mathbf{U}_1^*$ and $r < d_1$, i.e., $\mathbf{U}^*$ is low-rank, then $\mathbf{U}_1^*$ has $d_1 - r$ zero columns at optimality. This makes the

corresponding $d_1 - r$ dimensions of $\mathbf{W}_1 x$ and $\mathbf{W}_2 y$ no longer appear in the reconstruction term $\|(\mathbf{U}_1 \mathbf{W}_1 + \mathbf{U}_2)x + (\mathbf{U}_1 \mathbf{W}_2 - \mathbf{I})y\|$ of the loss $\mathcal{L}_{\text{CVAE}}$ defined in Eqn. (5). These dimensions of $\mathbf{W}_1 x$ and $\mathbf{W}_2 y$ can only influence *the term* $\|\mathbf{W}_1 x + \mathbf{W}_2 y\|$ *coming from the KL-regularization*, and thus, this term forces these $d_1 - r$ dimensions of $\mathbf{W}_1 x + \mathbf{W}_2 y$ to be 0. Hence, the distribution of these $d_1 - r$ dimensions of latent variable $z = \mathbf{W}_1 x + \mathbf{W}_2 y + \xi$ collapses exactly to the prior $\mathcal{N}(0, \eta_{\text{enc}}^2 \mathbf{I})$, which means that posterior collapse has occurred. Second, the singular values $\theta$'s of $\mathbf{E}$ decide the rank of $\mathbf{U}_1^*$ and therefore determine the level of the posterior collapse of the model. If the data $(x, y)$ has zero mean, for example, we can add bias term to have this effect, then $\mathbf{E} = \mathbf{P}_B \Psi^{1/2}(\mathbf{I} - \text{Cov}(\tilde{y}, \tilde{x}) \text{Cov}(\tilde{y}, \tilde{x})^\top)\Psi^{1/2}\mathbf{P}_B^\top$. Thus, given the same $y$, *the larger correlation (both positive and negative) between $x$ and $y$ leads to the larger level of posterior collapse*. In an extreme case, where $x = y$, we have that $\mathbf{E} = \mathbf{0}$ and $\mathbf{U}_1 = \mathbf{0}$. This is reasonable since $y = x$ can then be directly generated from the map $\mathbf{U}_2 x$, while the KL-regularization term converges to 0 due to the complete posterior collapse. Otherwise, if $x$ is independent of $y$, the singular values of $\mathbf{E}$ are maximized, and posterior collapse will be minimized in this case. Lastly, Theorem 2 implies that a sufficiently small $\beta$ or $\eta_{\text{dec}}$ will mitigate posterior collapse. This is aligned with the observation in (Lucas et al., 2019; Dai et al., 2020; Wang & Ziyin, 2022) for the linear VAE model. We also note that the $i$-th element $\sigma_i$ does not necessarily correspond with the $i$-th largest singular $\theta_i$, but it depends on the right singular matrix $\mathbf{R}$ of $\mathbf{U}^*$ to define the ordering.

## 4.2 MARKOVIAN HIERARCHICAL VAE

We extend our results to linear MHVAE with two levels of latent. Specifically, we study the case where the encoder variance matrix of the first latent variable $\boldsymbol{\Sigma}_1$ is *unlearnable* and *isotropic*, while the encoder variance of the second latent variable $\boldsymbol{\Sigma}_2$ is either: i) *learnable*, or, ii) *unlearnable isotropic*. In these settings, the encoder process includes the mapping from input $x \in \mathbb{R}^{D_0}$ to first latent $z_1 \in \mathbb{R}^{d_1}$ via the distribution $q_\phi(z_1|x) = \mathcal{N}(\mathbf{W}_1 x, \boldsymbol{\Sigma}_1), \mathbf{W}_1 \in \mathbb{R}^{d_1 \times D_0}$, and the mapping from $z_1$ to second latent $z_2 \in \mathbb{R}^{d_2}$ via $q_\phi(z_2|z_1) = \mathcal{N}(\mathbf{W}_2 z_1, \boldsymbol{\Sigma}_2), \mathbf{W}_2 \in \mathbb{R}^{d_2 \times d_1}$. Similarly, the decoder process parameterizes the distribution $p(z_1|z_2) = \mathcal{N}(\mathbf{U}_2 z_2, \eta_{\text{dec}}^2 \mathbf{I}), \mathbf{U}_2 \in \mathbb{R}^{d_1 \times d_2}$, and $p(x|z_1) = \mathcal{N}(\mathbf{U}_1 z_1, \eta_{\text{dec}}^2 \mathbf{I}), \mathbf{U}_1 \in \mathbb{R}^{D_0 \times d_1}$. The prior distribution is given by $p(z_1) = \mathcal{N}(0, \eta_{\text{enc}}^2 \mathbf{I})$. A graphical illustration is provided in Fig. 1(c). Our goal is to minimize the following negative ELBO (the detailed derivations are at Appendix F.2):

$$\mathcal{L}_{\text{HVAE}} = -\mathbb{E}_x \big[ \mathbb{E}_{q(z_1, z_2|x)}(\log p(x|z_1)) - \beta_1 \mathbb{E}_{q(z_2|z_1)}(D_{\text{KL}}(q(z_1|x)\|p(z_1|z_2))) - \beta_2 \mathbb{E}_{q(z_1|x)}(D_{\text{KL}}(q(z_2|z_1)\|p(z_2))) \big].$$

$$= \frac{1}{\eta_{\text{dec}}^2} \mathbb{E}_x \bigg[ \|\mathbf{U}_1 \mathbf{W}_1 x - x\|^2 + \text{trace}(\mathbf{U}_1 \boldsymbol{\Sigma}_1 \mathbf{U}_1^\top) + \beta_1 \|\mathbf{U}_2 \mathbf{W}_2 \mathbf{W}_1 x - \mathbf{W}_1 x\|^2 + \beta_1 \text{trace}(\mathbf{U}_2 \boldsymbol{\Sigma}_2 \mathbf{U}_2^\top)$$

$$+ \beta_1 \text{trace}((\mathbf{U}_2 \mathbf{W}_2 - \mathbf{I})\boldsymbol{\Sigma}_1(\mathbf{U}_2 \mathbf{W}_2 - \mathbf{I})^\top) + c^2 \beta_2 \big( \|\mathbf{W}_2 \mathbf{W}_1 x\|^2 + \text{trace}(\mathbf{W}_2 \boldsymbol{\Sigma}_1 \mathbf{W}_2^\top) + \text{trace}(\boldsymbol{\Sigma}_2) \big) \bigg]$$

$$- \beta_1 \log|\boldsymbol{\Sigma}_1| - \beta_2 \log|\boldsymbol{\Sigma}_2|.$$

Although the above encoder consists of two consecutive linear maps with additive noises, the ELBO training problem must have an extra KL-regularizer term between the two latents, i.e., $D_{\text{KL}}(q_\phi(z_1|x)\|p_\theta(z_1|z_2))$. This term, named "consistency term" in (Luo, 2022), complicates the training problem much more, as can be seen via the differences between $\mathcal{L}_{\text{HVAE}}$ and $\mathcal{L}_{\text{VAE}}$ in the standard VAE setting in Eqn. (3). We note that this model shares many similarities with diffusion models where the encoding process of diffusion models also consists of consecutive linear maps with injected noise, and their training process requires to minimize the consistency terms at each timestep. We characterize the global minima of $\mathcal{L}_{\text{HVAE}}$ for *learnable* $\boldsymbol{\Sigma}_2$ in the following theorem. Similar as above theorems, Theorem 3 derives the SVD forms and the closed-form singular values of the encoder and decoder maps to analyze the level of posterior collapse via the hyperparameters.

**Theorem 3** (Unlearnable isotropic $\boldsymbol{\Sigma}_1$, Learnable $\boldsymbol{\Sigma}_2$). *Let $c = \eta_{dec}/\eta_{enc}$, $\mathbf{Z} = \mathbb{E}_x(x\tilde{x}^\top) = \mathbf{R}\Theta\mathbf{S}^\top$ is the SVD of $\mathbf{Z}$ with singular values $\{\theta_i\}_{i=1}^{d_0}$ in non-increasing order, and unlearnable $\boldsymbol{\Sigma}_1 = \sigma_1^2 \mathbf{I}$ with $\sigma_1 > 0$. Assuming $d_0 \geq d_1 = d_2$, the optimal solution of $(\mathbf{U}_1^*, \mathbf{U}_2^*, \mathbf{W}_1^*, \mathbf{W}_2^*, \boldsymbol{\Sigma}_2^*)$ of $\mathcal{L}_{HVAE}$ is*

$$\mathbf{V}_1^* = \mathbf{W}_1^* \mathbf{P}_A \Phi^{1/2} = \mathbf{P}\Lambda\mathbf{R}^\top, \mathbf{U}_2^* = \mathbf{P}\Omega\mathbf{Q}^\top,$$

$$\mathbf{W}_2^* = \mathbf{U}_2^{*\top}(\mathbf{U}_2^* \mathbf{U}_2^{*\top} + c^2 \mathbf{I})^{-1}, \mathbf{U}_1^* = \mathbf{Z}\mathbf{V}_1^{*\top}(\mathbf{V}_1^* \mathbf{V}_1^{*\top} + \boldsymbol{\Sigma}_1)^{-1},$$

*and $\boldsymbol{\Sigma}_2^* = \frac{\beta_2}{\beta_1} \eta_{dec}^2 (\mathbf{U}_2^{*\top} \mathbf{U}_2^* + c^2 \mathbf{I})^{-1}$ where $\mathbf{P}, \mathbf{Q}$ are square orthonormal matrices. $\Lambda$ and $\Omega$ are rectangular diagonal matrices with the following elements, for $i \in [d_1]$:*

a) If $\theta_i^2 \geq \frac{\beta_2 \eta_{dec}^2}{\sigma_1^2} \max(\sigma_1^2, \frac{\beta_2}{\beta_1} \eta_{dec}^2)$: $\lambda_i^* = \frac{\sigma_1}{\sqrt{\beta_2} \eta_{dec}} \sqrt{\theta_i^2 - \beta_2 \eta_{dec}^2}$, $\omega_i^* = \sqrt{\frac{\sigma_1^2 \theta_i^2}{\beta_2 \eta_{enc}^2 \eta_{dec}^2} - \frac{\beta_2}{\beta_1} \frac{\eta_{dec}^2}{\eta_{enc}^2}}$.

b) If $\theta_i^2 < \frac{\beta_2 \eta_{dec}^2}{\sigma_1^2} \max(\sigma_1^2, \frac{\beta_2}{\beta_1} \eta_{dec}^2)$ and $\sigma_1^2 \geq \frac{\beta_2}{\beta_1} \eta_{dec}^2$: $\lambda_i^* = 0$, $\quad \omega_i^* = (\sigma_1^2 - \eta_{dec}^2 \beta_2 / \beta_1) / \eta_{enc}^2$.

c) If $\theta_i^2 < \frac{\beta_2 \eta_{dec}^2}{\sigma_1^2} \max(\sigma_1^2, \frac{\beta_2}{\beta_1} \eta_{dec}^2)$ and $\sigma_1^2 < \frac{\beta_2}{\beta_1} \eta_{dec}^2$: $\lambda_i^* = \sqrt{\max\left(0, \frac{\sigma_1}{\sqrt{\beta_1}} \left(\theta_i - \sqrt{\beta_1} \sigma_1\right)\right)}$, $\omega_i^* = 0$.

The detailed proof of Theorem 3 is in Appendix F.1. The proof uses zero gradient condition of critical points to derive $\mathbf{U}_1$ as a function of $\mathbf{V}_1$ and $\mathbf{W}_2$ as a function of $\mathbf{U}_2$ to reduce the number of variables. Then, the main novelty of the proof is that we prove $\mathbf{V}_1 \mathbf{V}_1^\top$ and $\mathbf{U}_2 \mathbf{U}_2^\top$ are *simultaneously diagonalizable*, and thus, we are able to convert the zero gradient condition into relations of their singular values $\lambda$'s and $\omega$'s. Thanks to these relations between $\lambda$'s and $\omega$'s, the loss function now can be converted to a function of singular values. The other cases of the input and latent dimensions, e.g., $d_0 < d_1$, are considered with details in Appendix F.1.

Theorem 3 identifies precise conditions for the occurrence of posterior collapse and the low-rank structure of the model at the optimum. There are several interesting remarks can be drawn from the results of the above theorem. First, regarding the posterior collapse occurrence, since $\mathbf{\Sigma}_2$ is learnable and diagonal, if there are some $i$ that $\omega_i = 0$, i.e., $\mathbf{U}_2^*$ is low-rank, the second latent variable will exhibit posterior collapse with the number of non-collapse dimensions of $z_2$ equal the number of non-zero $\omega_i$'s. However, the first latent variable might not suffer from posterior collapse even when $\mathbf{V}_1^*$ is low-rank due to the unlearnable isotropic $\mathbf{\Sigma}_1$, with the same reason that we discuss in the standard VAE case in Section 3. Second, all hyperparameters, including $\eta_{\text{dec}}, \beta_1, \beta_2, \sigma_1$ but except $\eta_{\text{enc}}$, are decisive for the rank of the encoders/decoders and the level of posterior collapse. In particular, the singular value $\omega_i > 0$ when either $\theta_i \geq \frac{\beta_2 \eta_{\text{dec}}^2}{\sqrt{\beta_1} \sigma_1}$ or $\sigma_1^2 \geq \frac{\beta_2}{\beta_1} \eta_{\text{dec}}^2$. Therefore, having a sufficiently small $\beta_2$ and $\eta_{\text{dec}}$ or a sufficiently large $\beta_1$ can mitigate posterior collapse for the second latent. Given $\omega_i > 0$, $\lambda_i > 0$ if $\theta_i^2 - \beta_2 \eta_{\text{dec}}^2 > 0$. Hence, a sufficiently small $\beta_2$ and $\eta_{\text{dec}}$ will also increase the rank of the mapping from the input data $x$ to the first latent $z_1$. In summary, decreasing the value of $\beta_2$ and $\eta_{\text{dec}}$ or increasing the value of $\beta_1$ can avoid posterior collapse, with the former preferred since it also avoids the low-rank structure for the first latent variable. We also characterize the global minima of two-latent MHVAE with both latent variances that are unlearnable and isotropic in Appendix F.2. In this setting, posterior collapse might not happen in either of the two latent variables.

**Remark 1.** *If the second latent variable $z_2$ suffers a complete collapse, the generating distribution becomes $p_\theta(z_1|z_2) = \mathcal{N}(0, \eta_{dec}^2 \mathbf{I})$ since all columns of $\mathbf{U}_2$ are now zero columns. Therefore, the MHVAE model now becomes similar to a standard VAE with the prior $\mathcal{N}(0, \eta_{dec}^2 \mathbf{I})$. We conjecture this observation also applies to MHVAE with more layers of latent structures: when a complete posterior collapse happens at a latent variable, its higher-level latent variables become inactive.*

### 4.3 MITIGATE POSTERIOR COLLAPSE

The analysis in Sections 4.1 and 4.2 identifies the causes of posterior collapse in conditional VAE and hierarchical VAE and implies some potential ways to fix it. Although the methods listed in Table 1 are the implications drawn from results with linear setting, we empirically prove their effectiveness in non-linear regime in the extensive experiments below in Section 5 and in Appendix A.

## 5 EXPERIMENTS

In this section, we demonstrate that the insights from the linear regime can shed light on the behaviors of the nonlinear CVAE and MHVAE counterparts. Due to the space limitation, we mainly present experiments on non-linear networks in the main paper. Experiments to verify our theorems for the linear case and additional empirical results for nonlinear VAE, CVAE and HVAE along with hyperparameter details can be found in Appendix A.

### 5.1 LEARNABILITY OF ENCODER VARIANCE AND POSTERIOR COLLAPSE

In this experiment, we demonstrate that the learnability of the encoder variance $\mathbf{\Sigma}$ is important to the occurrence of posterior collapse. We separately train two linear VAE models on MNIST dataset. The first model has data-independent and learnable $\mathbf{\Sigma}$, while the second model has fixed and unlearnable

| Conditional VAE | Markovian Hierarchical VAE |
|---|---|
| • A sufficiently small $\beta$ or $\eta_{\text{dec}}$ can mitigate posterior collapse. | • A sufficiently small $\beta_2$ or $\eta_{\text{dec}}$ can both mitigate collapse of the second latent and increase the rank of encoder/decoder of the first latent. |
| • Unlearnable encoder variance can prevent collapse. This is also true for MHVAE. | • Surprisingly, using larger $\beta_1$ value can mitigate the collapse for the second latent. |
| • Since high correlation between the input condition and the output leads to strong collapse, decorrelation techniques can help mitigate the collapse. | • Create separate maps between the input and each latent in case of a complete collapsed latent causes higher-level latents to lose information of the input. |

Table 1: Insights to mitigate posterior collapse drawn from our analysis

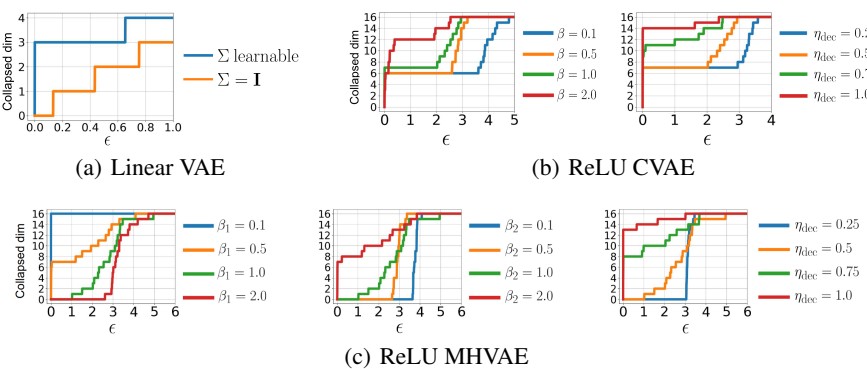

(a) Linear VAE  (b) ReLU CVAE

(c) ReLU MHVAE

Figure 2: Graphs of $(\epsilon, \delta)$-collapse with varied hyperparameters ($\delta = 0.05$). (a) For learnable $\mathbf{\Sigma}$, 3 (out of 5) latent dimensions collapse immediately at $\epsilon = 8 \times 10^{-5}$, while collapse does not happen with unlearnable $\mathbf{\Sigma} = \mathbf{I}$. (b) Larger value of $\beta$ or $\eta_{\text{dec}}$ makes more latent dimensions to collapse, and (c) Larger value of $\beta_2$ or $\eta_{\text{dec}}$ triggers more latent dimensions to collapse, whereas larger value of $\beta_1$ mitigates posterior collapse.

$\mathbf{\Sigma} = \mathbf{I}$. The latent dimension is 5, and we intentionally choose $\beta = 4, \eta_{\text{dec}} = 1$ to have 3 (out of 5) singular values $\theta$ equal 0. To measure the degree of posterior collapse, we use the $(\epsilon, \delta)$-collapse definition in (Lucas et al., 2019). Specifically, a latent dimension $i$ of latent $z$ is $(\epsilon, \delta)$-collapsed if $\mathbb{P}_x[D_{\text{KL}}(q(z_i|x)\|p(z_i)) < \epsilon] \geq 1 - \delta$. It is clear from Fig. 2(a) that with learnable $\mathbf{\Sigma}$, 3 out of 5 dimensions of the latent $z$ collapse immediately at small $\epsilon$, while the unlearnable variance does not.

## 5.2 CVAE EXPERIMENTS

We perform the task of reconstructing the MNIST digits from partial observation as described in (Sohn et al., 2015). We divide each digit image in the MNIST dataset into four quadrants: the bottom left quadrant is used as the input $x$ and the other three quadrants are used as the output $y$.

**Varying $\beta, \eta_{\text{dec}}$ experiment.** We train a ReLU CVAE that uses two-layer networks with ReLU activation as its encoder and decoder with different values of $\beta$ and $\eta_{\text{dec}}$. Fig. 2(b) demonstrates that decreasing $\beta$ and $\eta_{\text{dec}}$ can mitigate posterior collapse, as suggested in Section 4.3.

**Collapse levels on MNIST dataset.** Theorem 2 implies that training linear CVAE on a dataset with a lower set of $\theta_i$'s, i.e., the singular values of matrix $\mathbf{E}$ defined in Theorem 2, is more prone to posterior collapse. To verify this insight in nonlinear settings, we separately train multiple CVAE models, including linear CVAE, ReLU CVAE, and CNN CVAE, on three disjoint subsets of the MNIST dataset. Each subset contains all examples of each digit from the list $\{1, 7, 9\}$. To compare the values of $\theta_i$'s between datasets, we take the sum of the top-16 largest $\theta_i$'s and get the list $\{6.41, 13.42, 18.64\}$ for the digit $\{1, 9, 7\}$, respectively. The results presented in Fig. 3(a) empirically show that the values of $\theta_i$'s are negatively correlated with the degree of collapse.

## 5.3 MHVAE EXPERIMENTS

**Varying $\beta_1, \beta_2, \eta_{\text{dec}}$ experiment.** We train a ReLU MHVAE that uses two-layer networks with ReLU activation as the encoder and decoder, with multiple values of $\beta_1, \beta_2$ and $\eta_{\text{dec}}$. Fig. 2(c)

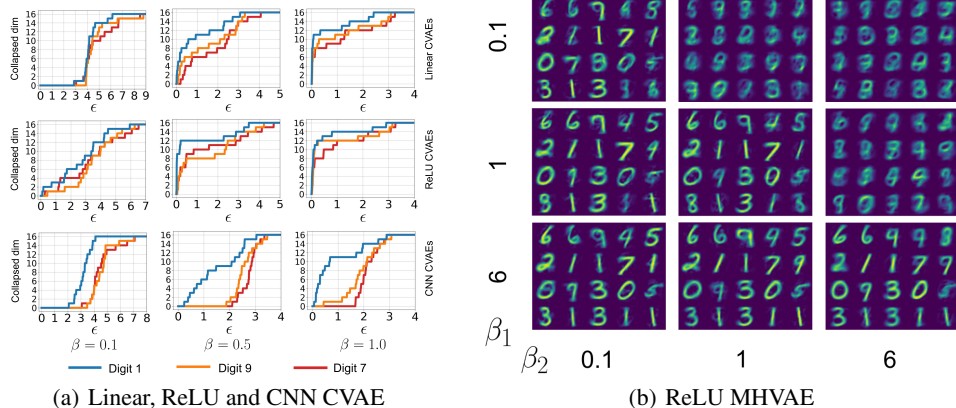

(a) Linear, ReLU and CNN CVAE        (b) ReLU MHVAE

Figure 3: (a) Graphs of $(\epsilon, \delta)$-collapse for several CVAEs trained separately on each of three digit $\{1, 9, 7\}$ subsets of MNIST ($\delta = 0.05$). Dataset with smaller $\theta_i$'s ($1 \to 9 \to 7$ in increasing order) has more collapsed dimensions, and (b) Samples reconstructed by nonlinear MHVAE. Smaller $\beta_2$ alleviates collapse and produces better samples, while smaller $\beta_1$ has the reverse effect.

demonstrates that decreasing $\beta_2$ and $\eta_{\text{dec}}$ reduce the degree of posterior collapse, while decreasing $\beta_1$ has the opposite effect, as Theorem 3 suggests.

**Samples reconstructed from ReLU MHVAE with varied $\beta_1$ and $\beta_2$.** We train the ReLU MHVAE with $\Sigma_1 = 0.5^2 \mathbf{I}$ and parameterized learnable $\Sigma_2(x) = (\text{Tanh}(\text{MLP}(z_1)))^2$ on MNIST dataset. Fig. 3(b) aligns to the insight discussed in Section 4.3 that decreasing the value of $\beta_2$ help mitigate collapse, and thus, produce better samples, while decreasing $\beta_1$ leads to blurry images. The full experiment with $\beta_1, \beta_2 \in \{0.1, 1.0, 2.0, 6.0\}$ can be found in Fig. 4 in Appendix A.

## 6   RELATED WORKS

To avoid posterior collapse, existing approaches modify the training objective to diminish the effect of KL-regularization term in the ELBO training, such as annealing a weight on KL term during training (Bowman et al., 2015; Huang et al., 2018; Sønderby et al., 2016; Higgins et al., 2016) or constraining the posterior to have a minimum KL-distance with the prior (Razavi et al., 2019). Another line of work avoids this phenomenon by limiting the capacity of the decoder (Gulrajani et al., 2017; Yang et al., 2017; Semeniuta et al., 2017) or changing its architecture (Van Den Oord et al., 2017; Dieng et al., 2019; Zhao et al., 2020). On the theoretical side, there have been efforts to detect posterior collapse under some restricted settings. (Dai et al., 2017; Lucas et al., 2019; Rolinek et al., 2019) study the relationship of VAE and probabilistic PCA. Specifically, (Lucas et al., 2019) showed that linear VAE can recover the true posterior of probabilistic PCA. (Dai et al., 2020) argues that posterior collapse is a direct consequence of bad local minima. The work that is more relatable to our work is (Wang & Ziyin, 2022), where they find the global minima of linear standard VAE and find the conditions when posterior collapse occurs. Nevertheless, the theoretical understanding of posterior collapse in important VAE models such as CVAE and HVAE remains limited. Due to space limitation, we defer the full related work discussion until Appendix B.

## 7   CONCLUDING REMARKS

Despite their prevalence in practical use as generative models, the theoretical understanding of CVAE and HVAE has remained limited. This work theoretically identifies causes and precise conditions for posterior collapse occurrence in linear CVAE and MHVAE from loss landscape perspectives. Some of our interesting insights beyond the results in linear standard VAE include: i) the strong correlation between the input conditions and the output of CVAE is indicative of strong posterior collapse, ii) posterior collapse may not happen if the encoder variance is unlearnable, even when the encoder network is low-rank. The experiments show that these insights are also predictive of nonlinear networks. One limitation of our work is the case of both encoder variances are learnable in two-latent MHVAE is not considered due to technical challenges and left as future works. Another limitation is that our theory does not consider the training dynamics that lead to the global minima and how they contribute to the collapse problem.

ACKNOWLEDGMENTS

This research/project is supported by the National Research Foundation Singapore under the AI Singapore Programme (AISG Award No: AISG2-TC-2023-012-SGIL). NH acknowledges support from the NSF IFML 2019844 and the NSF AI Institute for Foundations of Machine Learning.

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

# Appendix for "Beyond Vanilla Variational Autoencoders: Detecting Posterior Collapse in Conditional and Hierarchical Variational Autoencoders"

**Table of Contents**

# A    ADDITIONAL EXPERIMENTS AND NETWORK TRAINING DETAILS

We define $l_{rec}$ as the reconstruction loss for both CVAE and MHVAE. For CVAE, $l_{\text{KL}}$ is the KL-divergence $D_{\text{KL}}(q_\phi(z|x,y)||p(z|x))$. Similarly for MHVAE, $l_{KL_1}$ and $l_{KL_2}$ are the KL-divergence terms $D_{\text{KL}}(q_\phi(z_1|x)||p_\theta(z_1|z_2))$ and $D_{\text{KL}}(q_\phi(z_2|z_1)||p_\theta(z_2))$, respectively. To measure the discrepancy between the empirical singular values and the theoretical singular values of the encoder and decoder networks, we define the metric $\mathcal{D}_{\text{MA}}(\{u_i\},\{v_i\}) = \frac{1}{D}\sum_{i=1}^{D}|u_i - v_i|$ to be the mean absolute difference between two sets of *non-increasing* singular values $\{u_i\}_{i=1}^D, \{v_i\}_{i=1}^D$ (if the number of nonzero singular values is different between two sets, we extend the shorter set with 0's to match the length of the other set).

## A.1    DETAILS OF NETWORK TRAINING AND HYPERPARAMETERS IN SECTION 5 IN MAIN PAPER

In this subsection, we provide the remaining training details and hyperparameters for experiments shown in main paper. Unless otherwise stated, all the experiments in Section 5 are trained for 100 epochs with ELBO loss using Adam optimizer, learning rate of $1 \times 10^{-3}$, and batch size of 128.

### A.1.1    LEARNABILITY OF ENCODER VARIANCE AND POSTERIOR COLLAPSE

**Effect of learnable and unlearnable $\Sigma$ on posterior collapse (Fig. 2(a)):** In this experiment, we demonstrate that the learnability of the encoder variance $\Sigma$ is important to the occurrence of posterior collapse. We separately train 2 Linear VAE models on MNIST dataset. The first model has shared and learnable $\Sigma$, while the second model has fixed $\Sigma = \mathbf{I}$. We set $d_0 = 784, d_1 = 5, \eta_{\text{enc}} = \eta_{\text{dec}} = 1$, hidden dim $h = 256$, optimized with Adam optimizer for 100 epochs, learning rate set to $1 \times 10^{-4}$ for both models. It is clear from Fig. 2(a) that with learnable $\Sigma$, 3 out of 5 dimensions of the latent $z$ collapse with the KL divergence smaller than $8 \times 10^{-5}$, while in the same setting, the unlearnable encoder variance does not result in posterior collapse.

### A.1.2    CVAE EXPERIMENTS

**Varying $\beta, \eta_{\textbf{dec}}$ experiment (Fig. 2(b)):** In this experiment, we train ReLU CVAE with data-independent and learnable $\Sigma$ on the task of reconstructing the original MNIST digit from the bottom left quadrant. ReLU CVAE model is obtained by replacing all the linear layers in both the encoder and the decoder in the linear CVAE by two-layer MLP with ReLU activation. We set $d_0 = 196, d_1 = 16, d_2 = 588, \eta_{\text{enc}} = 1.0$ and hidden dimension $h$ of the MLPs is set to 16, learning rate set to $1 \times 10^{-4}$. We first run the experiment with fixed $\eta_{\text{dec}} = 1.0$ and $\beta$ chosen from the set $\{0.1, 0.5, 1.0, 2.0\}$, then we run the other experiment with fixed $\beta = 1.0$ and vary $\eta_{\text{dec}}$ from the set $\{0.25, 0.5, 1.0, 2.0\}$.

**Collapse level of MNIST digit datasets (Fig. 3(a)):** We separately train 3 CVAs models, namely linear CVAE, ReLU CVAE and CNN CVAE on 3 subsets of the MNIST dataset, each subset contains all examples of each digit $\{1, 7, 9\}$. The linear CVAE and ReLU CVAE have $\eta_{\text{enc}} = \eta_{\text{dec}} = 0.5, \beta = 1.0$ and other parameters the same as the experiment of Fig. 2(b). For CNN CVAE model, we replace all hidden layers in ReLU CVAE models by convolutional layers (with ReLU activation) and other settings stay the same. For the encoder of CNN CVAE, we use convolutional layers with kernel size $3 \times 3 \times 32, 3 \times 3 \times 16$, and stride $= 2$. For its decoder, we use transposed convolutional layers with kernel sizes $3 \times 3 \times 32, 3 \times 3 \times 16, 3 \times 3 \times 1$, and stride $= 2$.

### A.1.3    MHVAE EXPERIMENTS

**Varying $\beta_1, \beta_2, \eta_{\textbf{dec}}$ experiment with Relu MHVAE (Fig. 2(c)):** In this experiment, we use 2-layer MLP with ReLU and Tanh activation functions to replace all the linear layers in both encoder and decoder of Linear MHVAE. We train the model on MNIST dataset with $\Sigma_2$ parameterized by a 2-layer MLP with latent $z_1$ as the input. We set $\eta_{\text{enc}} = 0.5, \Sigma_1 = 0.5^2\mathbf{I}$, the latent dimensions and the hidden dimension are $d_1 = d_2 = 16$ and $h = 256$, respectively. We run 3 sub-experiments as follow: i) we fixed $\beta_2 = 1, \eta_{\text{dec}} = 0.5$ and then vary $\beta_1$ from the set $\{0.1, 0.5, 1.0, 2.0\}$, ii) we fixed $\beta_1 = 1, \eta_{\text{dec}} = 0.5$ and then vary $\beta_2$ from the set $\{0.1, 0.5, 1.0, 2.0\}$, and iii) we fixed $\beta_1 = \beta_2 = 1$ and then vary $\eta_{\text{dec}}$ from the set $\{0.25, 0.5, 1.0, 2.0\}$.

**Samples reconstructed from ReLU MHVAE with varied $\beta_1$ and $\beta_2$ (Fig. 4):** We vary $\beta_1, \beta_2$ from the set $\{0.1, 1.0, 2.0, 4.0\}$, $\eta_{\text{enc}}$ and $\eta_{\text{dec}}$ is set to 0.5. Other hyperparameters and the architecture of ReLU MHVAE model is identical to the experiment in Fig. 2(c).

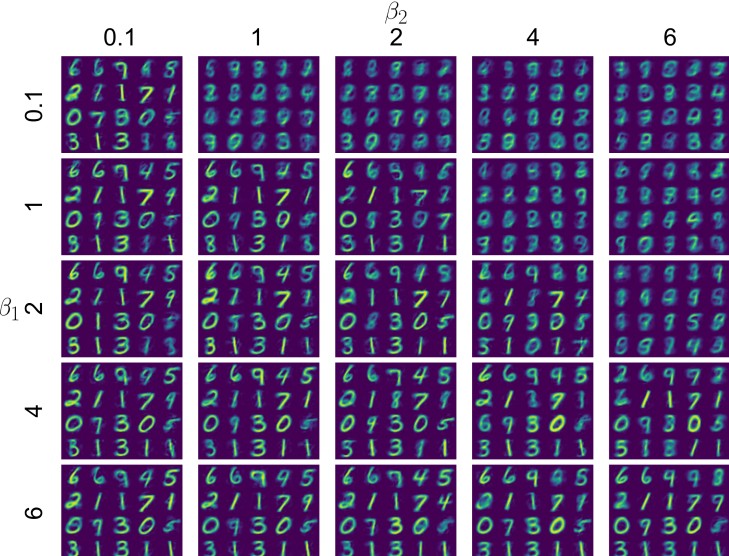

Figure 4: Samples reconstructed by nonlinear MHVAE with different $(\beta_1, \beta_2)$ combinations. Smaller $\beta_2$ alleviates collapse and produces better samples, while smaller $\beta_1$ has the reverse effect.

| | Log-likelihood | KL | AU |
|---|---|---|---|
| Learnable $\mathbf{\Sigma}$ | $-744.99$ | 9.54 | 69% |
| Unlearnable $\mathbf{\Sigma}$ | $-743.63$ | 9.80 | 100% |

Table 2: Test log-likelihood and posterior collapse degree of ReLU VAE on MNIST with learnable and unlearnable encoder variance.

## A.2 ADDITIONAL EXPERIMENTS

In this part, we empirically verify our theoretical results for linear VAE, CVAE and MHVAE by conducting experiments on both synthetic data and MNIST dataset. Furthermore, we continue to show that the insights drawn from our analysis are also true for non-linear settings.

### A.2.1 ADDITIONAL EXPERIMENT FOR VAE

**Log-likelihood, KL and AU of VAE with learnable and unlearnable $\mathbf{\Sigma}$ on MNIST (Table 2):** In this experiment, we use 2-layer MLP with ReLU activation to replace all linear layers in Linear VAE. We set $\mathbf{\Sigma} = 0.5^2 \mathbf{I}$ and other settings for this experiment to be identical to the experiment in Fig. 2(a). We evaluate the model performance on generative tasks using the importance weighted estimate of **log-likelihood** on a separate test set. To evaluate posterior collapse, we use two metrics: 1) the **KL divergence** between the posterior and the prior distribution, $D_{KL}(q(z|x)||p(z))$ and 2) the **active units (AU)** percentage (Wang et al., 2023; Burda et al., 2015) with $\epsilon = 0.01$. Higher AU percentage means that more latent dimensions are utilized by the model. In this experiment, we set $D_0 = 784, d_1 = 16, \eta_{\text{enc}} = \eta_{\text{dec}} = 1$. In Table 2, the unlearnable encoder variance with 100% AU (compared to 69% AU of the learnable case) indicates that unlearnable encoder variance can help to mitigate posterior collapse. This experiment provides further evidences that unlearnable encoder variance can help alleviate posterior collapse.

**ResNet-18 VAE with learnable and unlearnable $\mathbf{\Sigma}$ on CIFAR10 (Fig. 5):** In this experiment, we train ResNet-18 VAE model on CIFAR10 dataset (Krizhevsky et al., 2009). The model utilizes ResNet-18 architecture (He et al., 2016) to transform the input image $x$ into the latent vector $z_1$ in the encoder. In the decoder, transposed convolution layers are employed with kernel sizes of $3 \times 3 \times 32, 3 \times 3 \times 8, 3 \times 3 \times 3$, and a stride of 2. To maintain the appropriate dimensions for the ResNet-18 architecture and transposed convolution layers, two 2-layer MLPs with Relu activation are utilized for intermediate transformations. For this experiment, we set $D_0 = 3072, d_1 = 128, d_{\text{hidden}} = 512, \eta_{\text{enc}} = \eta_{\text{dec}} = 1.0, \beta = 2, \mathbf{\Sigma}_{\text{unlearnable}} = \mathbf{I}$, and train the model for 100 epochs with Adam optimizer and learning rate of $1 \times 10^{-3}$ with batch size 128.Fig. 5

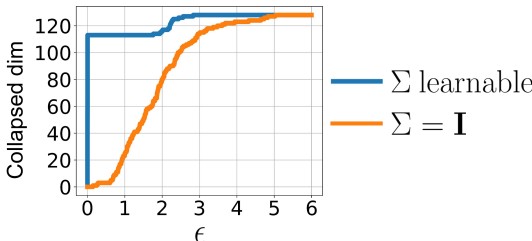

Figure 5: Graph of $(\epsilon, \delta)$-collapse of ResNet-18 VAE model with learnable $\boldsymbol{\Sigma}$ and unlearnable $\boldsymbol{\Sigma} = \mathbf{I}$, $\eta_{\text{enc}} = 1$ ($\delta = 0.01$). Learnable $\boldsymbol{\Sigma}$ suffers posterior collapse when most of the latent dimensions collapse to the prior at small $\epsilon$, while unlearnable $\boldsymbol{\Sigma}$ does not.

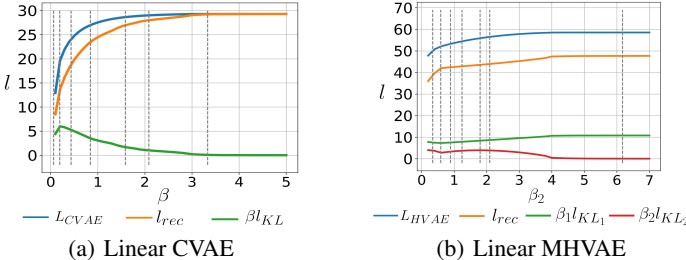

(a) Linear CVAE          (b) Linear MHVAE

Figure 6: Linear CVAE and MHVAE losses on MNIST dataset with $\beta$ and $\beta_2$ vary, respectively. Our theory correctly predicts complete posterior collapse at $\beta = 3.33$ for CVAE, and at $\beta_2 = 6.17$ for MHVAE.

illustrates that the unlearnable encoder variance model exhibits fewer collapsed latent dimensions compared to the learnable encoder variance model with a same $\epsilon$ threshold. These results justify that using an unlearnable encoder variance can alleviate posterior collapse, as pointed out in the paper.

### A.2.2 ADDITIONAL EXPERIMENTS FOR CVAE

**Linear CVAE (Fig. 6(a)):** In this experiment, we train linear CVAE model to verify the theoretical results by checking the sign of $\theta - \beta\eta_{\text{dec}}^2$ for posterior collapse described in Theorem 2. The top-1, 2, 4, 8, 16, 32, 64 leading singular vales $\theta_i$'s of MNIST dataset are $\{3.33, 2.09, 1.59, 0.84, 0.44, 0.19, 6.2 \times 10^{-2}\}$. In this experiment, we set $d_0 = 196, d_1 = 64, d_2 = 588, \eta_{\text{enc}} = \eta_{\text{dec}} = 1$, learning rate set to $1 \times 10^{-4}$. Thus, to determine the value of $\beta$ that cause a mode to collapse, we simply set $\beta = \theta$. Fig. 6(a) demonstrate that the convergence of $\beta l_{\text{KL}}$ to 0 agrees precisely with the threshold obtained from Theorem 2.

**Verification of Theorem 2 (Fig. 7, 8):** To verify Theorem 2, we measure the difference between the empirical singular values and theoretical singular values $\omega$ and variances $\sigma^2$ in two experiments for linear CVAE: synthetic experiment and MNIST experiment.

In the synthetic experiment, we optimize the matrix optimization problem derived in the proof of Theorem 2, which is equivalent to the minimizing negative ELBO problem. We randomly initialize each index of $x, y$ by sampling from $\mathcal{N}(0, 0.1^2)$ and optimize the matrix optimization objective with $d_0 = d_1 = d_2 = 5$ and $\eta_{\text{enc}} = \eta_{\text{dec}} = \beta = 1$. We use Adam optimizer for 200 iterations with learning rate 0.1. Fig. 7 corroborates Theorem 2 by demonstrating that $\mathcal{D}_{\text{MA}}(\{\omega_i\}, \{\omega_i^*\})$ and $\mathcal{D}_{\text{MA}}(\{\sigma_i\}, \{\sigma_i'\})$ converges to 0, which indicates the learned singular values $\{\omega_i\}$ and learned variances $\{\sigma_i^2\}$ converges to the theoretical singular values $\{\omega_i^*\}$ and variances $\{\sigma_i'^2\}$.

In the MNIST experiment, we train linear CVAE with ELBO loss on the task of reconstructing three remaining quadrants from the bottom left quadrant of MNIST dataset. Then, we compare the set of singular values $\{\omega_i\}$ and variances $\{\sigma_i^2\}$ with the theoretical solutions $\{\omega_i^*\}, \{\sigma_i'^2\}$ described in Theorem 2. In this experiment, $d_0 = 196, d_1 = 64, d_2 = 588$, and we set $\eta_{\text{enc}} = \eta_{\text{dec}} = \beta = 1$. The linear CVAE network is trained for 100 epochs with Adam optimizer, learning rate $1 \times 10^{-4}$, and batch size 128. From Fig. 8, it is evident that both $\mathcal{D}_{\text{MA}}(\{\omega_i\}, \{\omega_i^*\})$ and $\mathcal{D}_{\text{MA}}(\{\sigma_i\}, \{\sigma_i'\})$

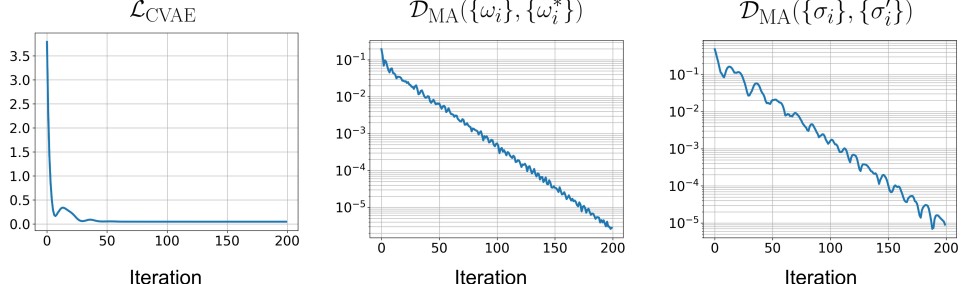

Figure 7: Evolution of $\mathcal{D}_{\mathrm{MA}}$ metrics across training iterations for linear CVAE on synthetic dataset.

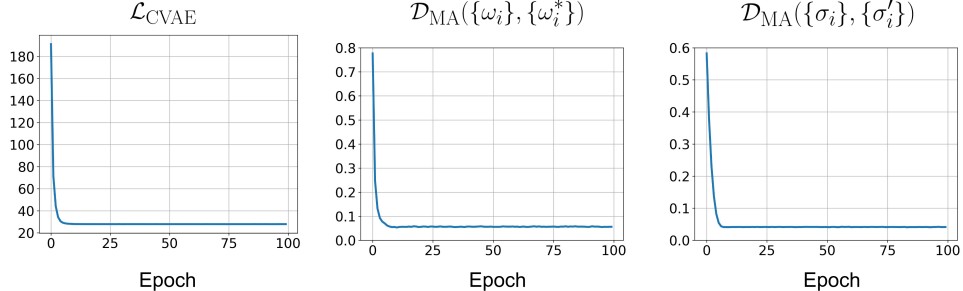

Figure 8: Evolution of $\mathcal{D}_{\mathrm{MA}}$ metrics across training epochs for linear CVAE trained on MNIST dataset.

|  |  | Log-likelihood | KL | AU |
|---|---|---|---|---|
|  | 0.25 | $-177.14$ | 18.85 | 56% |
|  | 0.5 | $-174.72$ | 15.42 | 56% |
| $\beta$ | 1.0 | $-173.93$ | 10.22 | 44% |
|  | 2.0 | $-174.32$ | 6.37 | 37% |
|  | 3.0 | $-176.43$ | 3.57 | 25% |
|  | 0.25 | 142.56 | 17.58 | 50% |
|  | 0.5 | $-173.93$ | 10.22 | 44% |
| $\eta_{\mathrm{dec}}$ | 0.75 | $-392.68$ | 5.64 | 38% |
|  | 1.0 | $-553.32$ | 2.10 | 13% |
|  | 2.0 | $-951.16$ | 0.00 | 0% |

Table 3: Test log-likelihood and posterior collapse degree of ReLU CVAE on MNIST. As Table 1 stated, smaller $\beta$ and $\eta_{\mathrm{dec}}$ mitigate collapse and have more active units.

converge to low value (less than 0.08 at the end of training), which indicates that the values $\{\omega_i\}$, $\{\sigma_i\}$ approaches the theoretical solution.

**Log-likelihood, KL and AU of CVAE with varied $\beta, \eta_{\mathbf{dec}}$ (Table 3):** All settings in this experiment are identical to the experiment in Fig. 2(b). We measure the log-likelihood, KL divergence of the model and AU of the model in Table 3. It is clear that decreasing $\beta, \eta_{\mathrm{dec}}$ alleviate posterior collapse. We observe that varying $\eta_{\mathrm{dec}}$ greatly affects the log-likelihood of the model, while changing $\beta$ has mixed effects on this metric.

**Correlation of $x, y$ and posterior collapse (Table 4):** We train ReLU CVAE model on synthetic dataset $(x, y)$, which is generated by sampling 1000 samples $y \in \mathbb{R}^{128} \sim \mathcal{N}(\mathbf{0}, \mathbf{I})$ and then sampling $x$ with different correlation level with $y$ as depicted in Table 4. In this experiment, we set $d_0 = d_1 = d_2 = 128, \eta_{\mathrm{enc}} = 1.0, \eta_{\mathrm{dec}} = 0.5$ and train the models with Adam optimizer for 1000 epochs with batch size 16. The results in Table 4 justify that higher correlation between $x$ and $y$ leads to a stronger collapse degree (higher AU percentage and KL divergence), as pointed out in our paper and especially, this insight also applies for non-linear conditional VAE.

| $\mathbf{y}, \mathbf{u} \sim \mathbb{N}(\mathbf{0}, \mathbf{I})$ | KL | AU |
|---|---|---|
| $\mathbf{x} = \mathbf{y}$ (correlation $= \mathbf{I}$) | 0.01 | 0.1% |
| $\mathbf{x} = \frac{1}{2}\mathbf{y} + \frac{\sqrt{3}}{2}\mathbf{u}$ (correlation $= 0.5\mathbf{I}$) | 66.20 | 54.1% |
| $\mathbf{x} = \frac{1}{4}\mathbf{y} + \frac{\sqrt{15}}{4}\mathbf{u}$ (correlation $= 0.25\mathbf{I}$) | 77.10 | 63.0% |
| $\mathbf{x} = \frac{1}{8}\mathbf{y} + \frac{\sqrt{63}}{8}\mathbf{u}$ (correlation $= 0.125\mathbf{I}$) | 79.64 | 64.7% |
| $\mathbf{x} \sim \mathbb{N}(\mathbf{0}, \mathbf{I})$ | 80.37 | 67.2% |

Table 4: Correlation and posterior collapse degree of ReLU CVAE on synthetic data. Higher correlation between the input condition $x$ and output $y$ leads to a stronger collapse.

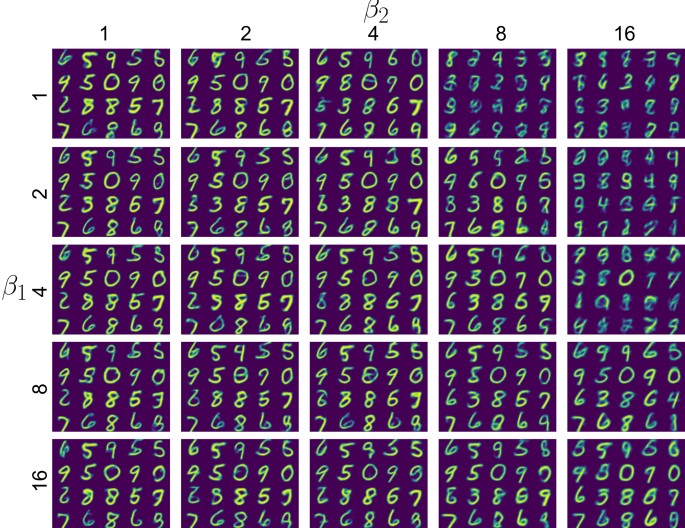

Figure 9: Samples generated by CNN Hierarchical VAE (with ReLU activation) with different $(\beta_1, \beta_2)$ combinations. Smaller $\beta_2$ alleviates collapse and produces better samples, while smaller $\beta_1$ has the reverse effect.

### A.2.3 ADDITIONAL EXPERIMENTS FOR MHVAE

**Samples reconstructed from CNN MHVAE with varied $\beta_1$ and $\beta_2$ (Fig. 9):** Similar to the experiment studying the quality of samples reconstructed of ReLU MHVAE with different combinations of $\beta_1$ and $\beta_2$ in Fig. 4, we train the CNN MHVAE model with $\mathbf{\Sigma}_1 = \sigma_1^2\mathbf{I}$ and parameterized $\mathbf{\Sigma}_2(x)$ depends on $z_1$. In this experiment, we replace all hidden layers in ReLU MHVAE models by convolutional layers (with ReLU activation). The encoder of CNN MHVAE uses convolutional layers with kernel size $3 \times 3 \times 64$, $3 \times 3 \times 32$ and stride $= 2$. The decoder of CNN MHVAE consists of transposed convolutional layers with kernel size $3 \times 3 \times 64$, $3 \times 3 \times 32$, $3 \times 3 \times 1$, and stride $= 2$. We set $\eta_{\text{enc}} = \eta_{\text{dec}} = 0.5, \sigma_1 = 0.5$. Fig. 9 illustrates that decreasing the value of $\beta_2$ help mitigate collapse and produce better samples, while decreasing $\beta_1$ causes the images to be blurry, which is similar to the result in the case of of ReLU MHVAE.

**Varying $\beta_1, \beta_2, \eta_{\text{dec}}$ experiment with ResNet-18 MHVAE on CIFAR10 (Fig. 10):** In this experiment, we train ResNet-18 MHVAE on CIFAR10 dataset for 100 epochs with Adam optimizer, learning rate of $1 \times 10^{-3}$, and batch size of 128. Within the model, ResNet-18 architecture is utilized to map the input image $x$ to the latent vector $z_1$ in the encoder, the transformation of $z_1$ to $y$ in the decoder is parameterized by transposed convolutional layers with kernel size $3 \times 3 \times 32$, $3 \times 3 \times 8$, $3 \times 3 \times 3$ and stride $= 2$. Similar to ResNet-18 VAE, two 2-layer MLPs with Relu activation are utilized for intermediate transformations. Furthermore, the mappings from $z_1$ to $z_2$ and $z_2$ to $z_1$ are implemented using 2-layer MLPs with Relu activation and hidden dimension 1024. We set $d_0 = 3072, d_1 = d_2 = 64, \eta_{\text{enc}} = \sigma_1 = 0.1$. Similar to the varying $\beta_1, \beta_2, \eta_{\text{dec}}$ experiment in Section 5.3, we alternatively run 3 sub-experiments as follow: i) we fixed $\beta_2 = 1, \eta_{\text{dec}} = 0.1$ and then vary $\beta_1$ from the set $\{0.1, 0.5, 1.0, 2.0\}$, ii) we fixed $\beta_1 = 1, \eta_{\text{dec}} = 0.1$ and then vary $\beta_2$ from the set $\{0.1, 0.5, 1.0, 2.0\}$, and iii) we fixed $\beta_1 = \beta_2 = 1$ and then vary $\eta_{\text{dec}}$ from the set

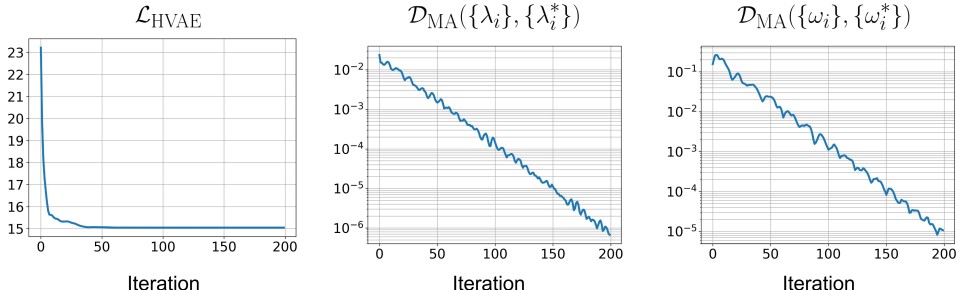

Figure 10: Graph of $(\epsilon, \delta)$-collapsed for ResNet-18 MHVAE model trained on CIFAR10 dataset with varying hyperparameters $\beta_1$, $\beta_2$ and $\eta_{\text{dec}}$. ($\delta = 0.05$). As Table 1 suggests, smaller $\beta_2$ and $\eta_{\text{dec}}$ mitigate collapse and have more active units, while $\beta_1$ has the reverse effect.

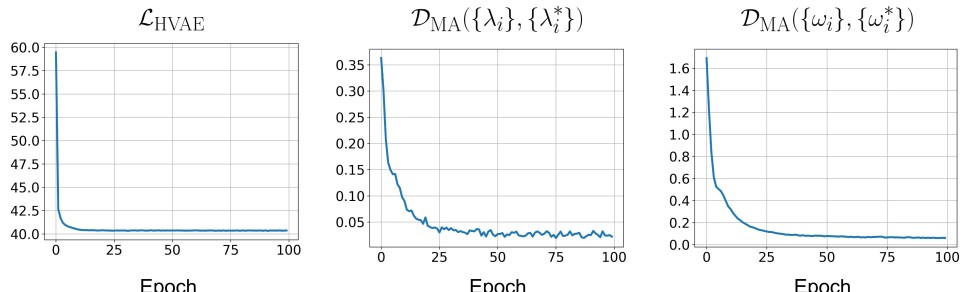

Figure 11: Evolution of $\mathcal{D}_{\text{MA}}$ metrics across training iterations for linear MHVAE with unlearnable isotropic $\boldsymbol{\Sigma}_1$ and learnable $\boldsymbol{\Sigma}_2$ on synthetic dataset.

Figure 12: Evolution of $\mathcal{D}_{\text{MA}}$ metrics across training epochs for linear MHVAE with unlearnable isotropic $\boldsymbol{\Sigma}_1$ and learnable $\boldsymbol{\Sigma}_2$ trained on MNIST dataset.

$\{0.25, 0.5, 1.0, 2.0\}$. The experiments depicted in Fig. 10 clearly support Theorem 3 by demonstrating that decreasing $\beta_2$ and $\eta_{\text{dec}}$ reduces the degree of posterior collapse, while decreasing $\beta_1$ has the opposite effect.

**Linear MHVAE (Fig. 6(b)):** In this experiment, we train the two-latent linear MHVAE model with unlearnable $\boldsymbol{\Sigma}_1 = \mathbf{I}$ and learnable $\boldsymbol{\Sigma}_2$ on the MNIST dataset with latent dimensions $d_1 = d_2 = 64$. The experiment aims to check the threshold that cause $\omega_i^*$'s to be 0 described in Theorem 3. We keep $\beta_1 = 1$ and then gradually increase $\beta_2$ to check whether posterior collapse happens as the threshold predicted. Fig. 6(b) shows that the convergence of KL loss for $z_2$ to zero agrees with the threshold obtained from Theorem 3. In which, the top-1, 2, 4, 8, 16, 32, 64 leading singular $\theta_i$'s used for computing $\beta_2$ thresholds are $\{6.17, 2.10, 1.80, 1.24, 0.89, 0.58, 0.34\}$.

**Verification of Theorem 3 (Fig. 11, 12):** To verify Theorem 3 for linear MHVAE with learnable $\boldsymbol{\Sigma}_2$, we further perform two experiments: synthetic experiment and MNIST experiment.

In the synthetic experiment for linear MHVAE with learnable $\boldsymbol{\Sigma}_2$, we initialize each index of $x, y$ by sampling from $\mathcal{N}(0, 0.1^2)$. We optimize the matrix optimization problem in the proof of Theorem 3, which is equivalent to the minimizing negative ELBO problem. We choose $d_0 = d_1 = d_2 = 5$, $\boldsymbol{\Sigma}_1 = \mathbf{I}$ and $\eta_{\text{enc}} = \eta_{\text{dec}} = \beta_1 = 1, \beta_2 = 2$. We use Adam optimizer for 200 iterations with learning rate 0.1. The convergence of $\mathcal{D}_{\text{MA}}(\{\lambda_i\}, \{\lambda_i^*\})$ and $\mathcal{D}_{\text{MA}}(\{\omega_i\}, \{\omega_i^*\})$ to 0 depicted in Fig. 11 empirically corroborate Theorem 3.

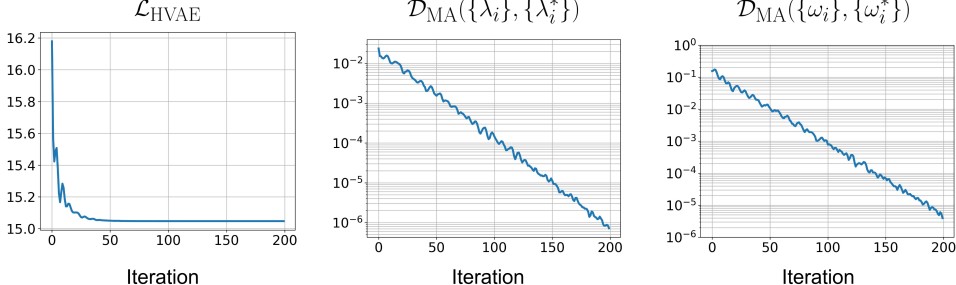

Figure 13: Evolution of $\mathcal{D}_{\text{MA}}$ metrics across training iterations for linear MHVAE with unlearnable isotropic $\boldsymbol{\Sigma}_1, \boldsymbol{\Sigma}_2$ on synthetic dataset.

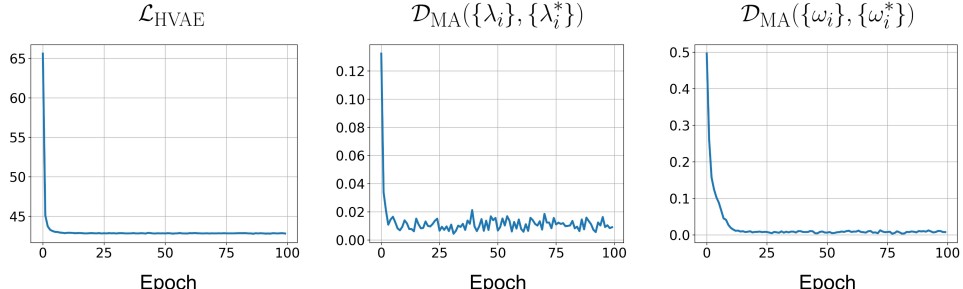

Figure 14: Evolution of $\mathcal{D}_{\text{MA}}$ metrics across training epochs for linear MHVAE with unlearnable isotropic $\boldsymbol{\Sigma}_1, \boldsymbol{\Sigma}_2$ trained on MNIST dataset.

In the MNIST experiment with learnable $\boldsymbol{\Sigma}_2$, we train a two-latent linear MHVAE, learnable and data-independent $\boldsymbol{\Sigma}_2$ by minimizing the negative ELBO. In this experiment, we set $d_0 = 784, d_1 = d_2 = 10, \eta_{\text{enc}} = \eta_{\text{dec}} = 0.5, \boldsymbol{\Sigma}_1 = 0.5^2\mathbf{I}$. The ELBO loss is optimized with Adam optimizer with learning rate $1 \times 10^{-3}$ and batch size 128 for 100 epochs. Fig. 12 demonstrates that both $\mathcal{D}_{\text{MA}}(\{\lambda_i\}, \{\lambda_i^*\})$ and $\mathcal{D}_{\text{MA}}(\{\omega_i\}, \{\omega_i^*\})$ converges to low value, which empirically verifies Theorem 3.

**Verification of Theorem 5 (Fig. 13, 14):** Similar as above, but in the setting of unlearnable isotropic $\boldsymbol{\Sigma}_1, \boldsymbol{\Sigma}_2$ studied in Theorem 5, we also verify it by performing two similar experiments on synthetic dataset and MNIST dataset.

In the synthetic experiment, except pre-defined $\boldsymbol{\Sigma}_2 = \mathbf{I}$, the other settings of this experiment is identical to the synthetic experiment for MHVAE with learnable $\boldsymbol{\Sigma}_2$ above. The clear convergence of $\mathcal{D}_{\text{MA}}(\{\lambda_i\}, \{\lambda_i^*\})$ and $\mathcal{D}_{\text{MA}}(\{\omega_i\}, \{\omega_i^*\})$ to 0 demonstrated in Fig. 13 empirically verified Theorem 5.

In the MNIST experiment, linear MHVAE is trained with pre-defined and unlearnable $\boldsymbol{\Sigma}_1 = \boldsymbol{\Sigma}_2 = 0.5^2\mathbf{I}$. Other hyperparameters and training settings is identical to the MNIST experiment for linear MHVAE with learnable $\boldsymbol{\Sigma}_2$ above. Fig. 14 also corroborate the convergence of the sets of singular values $\{\lambda_i\}, \{\omega_i\}$ to the theoretical values.

**Log-likelihood, KL and AU of HVAE with varied $\beta, \eta_{\text{dec}}$ (Table 5):** All settings in this experiment are identical to the experiment in Fig. 2(c). Table 5 demonstrates that increasing $\beta_1$ alleviate posterior collapse and increasing $\beta_2$ and $\eta_{\text{dec}}$ have the opposite effect. We also notice that changing $\eta_{\text{dec}}$ greatly affects the log-likelihood of the model, while varying $\beta_1, \beta_2$ has mixed effects on this metric.

## B   RELATED WORKS

**Posterior collapse:** To avoid posterior collapse, existing approaches modify the training objective to diminish the effect of KL-regularization term in the ELBO training. This includes heuristic approaches such as annealing a weight on KL term during training (Bowman et al., 2015; Huang et al., 2018; Sønderby et al., 2016; Higgins et al., 2016), finding tighter bounds for the marginal log-

|          |      | Log-likelihood | KL    | AU    |
|----------|------|----------------|-------|-------|
|          | 0.25 | $-229.88$      | 1.09  | 34%   |
|          | 0.5  | $-225.89$      | 4.59  | 89%   |
| $\beta_1$ | 1.0  | $-225.82$      | 8.77  | 100%  |
|          | 2.0  | $-226.64$      | 13.00 | 100%  |
|          | 3.0  | $-227.49$      | 16.19 | 100%  |
|          | 0.25 | $-228.66$      | 18.07 | 100%  |
|          | 0.5  | $-226.72$      | 13.04 | 100%  |
| $\beta_2$ | 1.0  | $-225.82$      | 8.77  | 100%  |
|          | 2.0  | $-226.05$      | 4.24  | 82%   |
|          | 3.0  | $-227.41$      | 1.98  | 40%   |
|          | 0.25 | 211.79         | 18.07 | 100%  |
|          | 0.5  | $-225.82$      | 8.77  | 100%  |
| $\eta_{\text{dec}}$ | 0.75 | $-522.61$      | 3.45  | 68%   |
|          | 1.0  | $-740.24$      | 0.49  | 18%   |
|          | 2.0  | $-1278.18$     | 0.00  | 0%    |

Table 5: Test log-likelihood and posterior collapse degree of ReLU two-latent MHVAE trained on MNIST dataset. As Table 1 suggests, smaller $\beta_2$ and $\eta_{\text{dec}}$ mitigate collapse and have more active units, while $\beta_1$ has the reverse effect.

likelihood (Burda et al., 2015) or constraining the posterior family to have a minimum KL-distance with the prior (Razavi et al., 2019). Another line of work avoids this phenomenon by limiting the capacity of the decoder (Gulrajani et al., 2017; Yang et al., 2017; Semeniuta et al., 2017) or changing its architecture (Van Den Oord et al., 2017; Dieng et al., 2019; Zhao et al., 2020). (Kinoshita et al., 2023) proposes a potential way to control posterior collapse by using inverse Lipchitz network in the decoder. Using hierarchical VAE is also demonstrated to alleviate posterior collapse with good performances (Child, 2021; Sohn et al., 2015; Maaløe et al., 2017; Vahdat & Kautz, 2020; Maaløe et al., 2019). However, (Kuzina & Tomczak, 2023) empirically observes that this issue is still present in current state-of-the-art hierarchical VAE models. On the theoretical side, there have been efforts to characterize posterior collapse under some restricted settings. (Dai et al., 2017; Lucas et al., 2019; Rolinek et al., 2019) study the relationship of VAE and probabilistic PCA. Specifically, (Lucas et al., 2019) showed that linear VAE can recover the true posterior of probabilistic PCA. They also prove that ELBO does not introduce additional bad local minima with posterior collapse in linear VAE model. (Dai et al., 2020) argues that posterior collapse is a direct consequence of bad local minima of the loss surface and prove that a small nonlinear perturbations from the linear VAE can produce such minima. The work that is more relatable to our work is (Wang & Ziyin, 2022), where they find the global minima of linear standard VAE and find the conditions when posterior collapse occurs. Nevertheless, the theoretical understanding of posterior collapse in important VAE models such as CVAE and HVAE remains limited.

**Linear network:** Analyzing deep linear networks is an important step in studying deep nonlinear networks. The theoretical analysis of deep nonlinear networks is very challenging and, in fact, there has been no rigorous theory for deep nonlinear networks yet to the best of our knowledge. Thus, deep linear networks have been studied to provide insights into the behavior of deep nonlinear networks. For example, using only linear regression, (Hastie et al., 2022) can recover several phenomena observed in large-scale deep nonlinear networks, including the double descent phenomenon (Nakkiran et al., 2021). (Saxe et al., 2013; Kawaguchi, 2016; Laurent & von Brecht, 2018; Hardt & Ma, 2017) empirically show that the optimization of deep linear models exhibits similar properties to those of the optimization of deep nonlinear models. As pointed out in Saxe et al. (2013), despite the linearity of their input-output map, deep linear networks have nonlinear gradient descent dynamics on weights that change with the addition of each new hidden layer. This nonlinear learning phenomenon is proven to be similar to those seen in deep nonlinear networks.

In practice, deep linear networks can help improve the training and performance of deep nonlinear networks Huh et al. (2023); Guo et al. (2021); Arora et al. (2018). Specifically, Huh et al. (2023) empirically proves that linear overparameterization in nonlinear networks improves generalization on classification tasks (see Section 4 in Huh et al. (2023)). In particular, Huh et al. (2023) expands

each linear layer into a succession of multiple linear layers and does not include any non-linearities in between. Guo et al. (2021) applies a similar strategy for compact networks, and their experiments show that training such expanded networks yields better results than training the original compact networks. Arora et al. (2018) shows that linear overparameterization, i.e., the use of a deep linear network in place of a classic linear model, induces on gradient descent a particular preconditioning scheme that can accelerate optimization. The preconditioning scheme that deep linear layers introduce can be interpreted as using momentum and adaptive learning rate.

## C  SAMPLE-WISE ENCODER VARIANCE

### C.1  CONDITIONAL VAE

In this section, we extend the minimize problem in Eqn. (5) to data-dependent encoder variance $\Sigma(x)$. Indeed, assume the training samples are $\{(x_i, y_i)\}_{i=1}^N$ and $q(z_i|x,y) \sim \mathcal{N}(\mathbf{W}_1 x_i + \mathbf{W}_2 y_i, \Sigma_i)\ \forall i \in [N]$, we have:

$$
\begin{aligned}
&-\text{ELBO}_{CVAE}(\mathbf{W}_1, \mathbf{W}_2, \mathbf{U}_1, \mathbf{U}_2, \Sigma_1, \ldots, \Sigma_N) \\
&= -\mathbb{E}_{x,y}\big[\mathbb{E}_{q_\phi(z|x,y)}\left[p_\theta(y|x,z)\right] - \beta D_{KL}(q_\phi(z|x,y)||p(z|x))\big] \\
&= \frac{1}{N}\sum_i \mathbb{E}_{q_\phi(z|x,y)}\left[\frac{1}{\eta_{\text{dec}}^2}\|\mathbf{U}_1 z_i + \mathbf{U}_2 x_i - y_i\|^2 - \beta \xi_i^\top \Sigma_i^{-1}\xi_i - \beta\log|\Sigma_i| + \frac{\beta}{\eta_{\text{enc}}^2}\|z_i\|^2\right] \\
&= \frac{1}{N}\sum_i\left(\frac{1}{\eta_{\text{dec}}^2}\left[\|(\mathbf{U}_1\mathbf{W}_1 + \mathbf{U}_2)x_i + (\mathbf{U}_1\mathbf{W}_2 - \mathbf{I})y_i\|^2 + \text{trace}(\mathbf{U}_1\Sigma_i\mathbf{U}_1^\top)\right.\right. \\
&\left.\left. + \beta c^2(\|\mathbf{W}_1 x_i + \mathbf{W}_2 y_i\|^2 + \text{trace}(\Sigma_i))\right] - \beta d_1 - \beta\log|\Sigma_i|\right).
\end{aligned}
$$

Taking the derivative w.r.t each $\Sigma_i$, we have:

$$
\begin{aligned}
-\frac{\partial \text{ELBO}}{\partial \Sigma_i} &= \frac{1}{\eta_{\text{dec}}^2}(\mathbf{U}_1^\top\mathbf{U}_1 + \beta c^2\mathbf{I}) - \beta\Sigma_i^{-1} = \mathbf{0} \\
&\Rightarrow \Sigma_i = \beta\eta_{\text{dec}}^2(\mathbf{U}_1^\top\mathbf{U}_1 + \beta c^2\mathbf{I})^{-1}.
\end{aligned}
\tag{8}
$$

We have $\Sigma_i = \Sigma$ for all $i$ at optimal, and thus, the above minimizing negative ELBO problem is equivalent to the training problem in Eqn. (5) that use the same $\Sigma$ for all data.

### C.2  MARKOVIAN HIERARCHICAL VAE

Similarly, we consider the negative ELBO function for MHVAE two latents with data-dependent encoder variance $\Sigma$. Indeed, assume training samples are $\{x_i\}_{i=1}^N$, and dropping the multiplier $1/2$ and some constants in the negative ELBO, we have:

$$
\begin{aligned}
&-\text{ELBO}_{\text{HVAE}}(\mathbf{W}_1, \mathbf{W}_2, \mathbf{U}_1, \mathbf{U}_2, \{\Sigma_{1,i}\}_{i=1}^N, \{\Sigma_{2,i}\}_{i=1}^N) = -\mathbb{E}_x\big[\mathbb{E}_{q_\phi(z_1|x)q_\phi(z_2|z_1)}(\log p_\theta(x|z_1)) \\
&\quad - \beta_1\mathbb{E}_{q_\phi(z_2|z_1)}(D_{\text{KL}}(q_\phi(z_1|x)||p_\theta(z_1|z_2)) - \beta_2\mathbb{E}_x\mathbb{E}_{q_\phi(z_1|x)}(D_{\text{KL}}(q_\phi(z_2|z_1)||p_\theta(z_2)))\big] \\
&= \frac{1}{N}\left(\sum_{i=1}^N \frac{1}{\eta_{\text{dec}}^2}\left[\|\mathbf{U}_1\mathbf{W}_1 x_i - x_i\|^2 + \text{trace}(\mathbf{U}_1\Sigma_{1,i}\mathbf{U}_1^\top) + \beta_1\|\mathbf{U}_2\mathbf{W}_2\mathbf{W}_1 x_i - \mathbf{W}_1 x_i\|^2\right.\right. \\
&\left. + \beta_1\text{trace}(\mathbf{U}_2\Sigma_{2,i}\mathbf{U}_2^\top) + \beta_1\text{trace}((\mathbf{U}_2\mathbf{W}_2 - \mathbf{I})\Sigma_{1,i}(\mathbf{U}_2\mathbf{W}_2 - \mathbf{I})^\top)\right] \\
&\left. + c^2\beta_2\big(\|\mathbf{W}_2\mathbf{W}_1 x\|^2 + \text{trace}(\mathbf{W}_2\Sigma_{1,i}\mathbf{W}_2^\top) + \text{trace}(\Sigma_{2,i})\big) - \beta_1\log|\Sigma_{1,i}| - \beta_2\log|\Sigma_{2,i}|\right),
\end{aligned}
$$

where the details of the above derivation are from the proof in Appendix F.1.

Taking the derivative w.r.t each $\mathbf{\Sigma}_{1,i}$ and $\mathbf{\Sigma}_{2,i}$, $\forall i \in [N]$, we have at critical points of $-\text{ELBO}_{\text{HVAE}}$:

$$-N\frac{\partial \text{ELBO}}{\partial \mathbf{\Sigma}_{1,i}} = \frac{1}{\eta_{\text{dec}}^2}(\mathbf{U}_1^\top \mathbf{U}_1 + \beta_1(\mathbf{U}_2\mathbf{W}_2 - \mathbf{I})^\top(\mathbf{U}_2\mathbf{W}_2 - \mathbf{I}) + c^2\beta_2\mathbf{W}_2^\top\mathbf{W}_2) - \beta_1\mathbf{\Sigma}_{1,i}^{-1} = \mathbf{0}$$

$$\Rightarrow \mathbf{\Sigma}_{1,i} = \beta_1\eta_{\text{dec}}^2(\mathbf{U}_1^\top\mathbf{U}_1 + \beta_1(\mathbf{U}_2\mathbf{W}_2 - \mathbf{I})^\top(\mathbf{U}_2\mathbf{W}_2 - \mathbf{I}) + c^2\beta_2\mathbf{W}_2^\top\mathbf{W}_2)^{-1}.$$

$$-N\frac{\partial \text{ELBO}}{\partial \mathbf{\Sigma}_{2,i}} = \frac{1}{\eta_{\text{dec}}^2}(\beta_1\mathbf{U}_2^\top\mathbf{U}_2 + c^2\beta_2\mathbf{I}) - \beta_2\mathbf{\Sigma}_{2,i}^{-1} = \mathbf{0}$$

$$\Rightarrow \mathbf{\Sigma}_{2,i} = \frac{\beta_2}{\beta_1}\eta_{\text{dec}}^2(\mathbf{U}_2^\top\mathbf{U}_2 + c^2\frac{\beta_2}{\beta_1}\mathbf{I})^{-1}. \tag{9}$$

Thus, we have at optimal, $\mathbf{\Sigma}_{1,i}$ and $\mathbf{\Sigma}_{2,i}$ are all equal for all input data. Hence, we can consider the equivalent problem of minimizing the negative ELBO with same encoder variances $\mathbf{\Sigma}_1$ and $\mathbf{\Sigma}_2$ for all training samples. Similar conclusions for linear standard VAE can be obtained by letting $\mathbf{U}_2 = \mathbf{W}_2 = \mathbf{\Sigma}_{2,1} = \ldots = \mathbf{\Sigma}_{2,N} = \mathbf{0}$ in the above arguments.

## D  PROOFS FOR STANDARD VAE

In this section, we prove Theorem 1 in Section D.1. We also derive the similar results for learnable $\mathbf{\Sigma}$ case in Section D.2.

Recall that $\mathbf{A} := \mathbb{E}_x(xx^\top) = \mathbf{P}_A\mathbf{\Phi}\mathbf{P}_A^\top$, $\tilde{x} = \mathbf{\Phi}^{-1/2}\mathbf{P}_A^\top x$ and $\mathbf{Z} := \mathbb{E}_{\tilde{x}}(x\tilde{x}^\top) \in \mathbb{R}^{D_0 \times d_0}$. Also, let $\mathbf{V}_1 = \mathbf{W}_1\mathbf{P}_A\mathbf{\Phi}^{1/2} \in \mathbb{R}^{d_1 \times D}$.

We minimize the negative ELBO loss function (after dropping multiplier $1/2$ and some constants):

$$\mathcal{L}_{VAE} = \mathbb{E}_x\big(-\mathbb{E}_{q(z|x)}[\log p(x|z)] + \beta D_{KL}(q(z|x)||p(z))\big)$$

$$= \frac{1}{\eta_{\text{dec}}^2}\mathbb{E}_x\Big[\|\mathbf{U}\mathbf{W}x - y\|^2 + \text{trace}(\mathbf{U}\mathbf{\Sigma}\mathbf{U}^\top) + \beta c^2(\|\mathbf{W}x\|^2 + \text{trace}(\mathbf{\Sigma}))\Big] - \beta\log|\mathbf{\Sigma}|$$

$$= \frac{1}{\eta_{\text{dec}}^2}\Big[\|\mathbf{U}\mathbf{V} - \mathbf{Z}\|_F^2 + \text{trace}(\mathbf{U}^\top\mathbf{U}\mathbf{\Sigma}) + \beta c^2\|\mathbf{V}\|_F^2 + \beta c^2\,\text{trace}(\mathbf{\Sigma})\Big] - \beta\log|\mathbf{\Sigma}|. \tag{10}$$

### D.1  UNLEARNABLE DIAGONAL ENCODER VARIANCE $\mathbf{\Sigma}$

*Proof of Theorem 1.* Since the $\mathbf{\Sigma} = \text{diag}(\sigma_1^2, \sigma_2^2, \ldots, \sigma_{d_1}^2)$ is fixed, we can drop the term $\beta c^2\,\text{trace}(\mathbf{\Sigma})$:

$$\mathcal{L}_{VAE} = \frac{1}{\eta_{\text{dec}}^2}\Big[\|\mathbf{U}\mathbf{V} - \mathbf{Z}\|_F^2 + \text{trace}(\mathbf{U}^\top\mathbf{U}\mathbf{\Sigma}) + \beta c^2\|\mathbf{V}\|_F^2\Big]. \tag{11}$$

At critical points of $\mathcal{L}_{VAE}$:

$$\frac{1}{2}\frac{\partial \mathcal{L}}{\partial \mathbf{V}} = \frac{1}{\eta_{\text{dec}}^2}(\mathbf{U}^\top(\mathbf{U}\mathbf{V} - \mathbf{Z}) + \beta c^2\mathbf{V}) = \mathbf{0}.$$

$$\frac{1}{2}\frac{\partial \mathcal{L}}{\partial \mathbf{U}} = \frac{1}{\eta_{\text{dec}}^2}((\mathbf{U}\mathbf{V} - \mathbf{Z})\mathbf{V}^\top + \mathbf{U}\mathbf{\Sigma}) = \mathbf{0}. \tag{12}$$

From $\frac{\partial \mathcal{L}}{\partial \mathbf{V}} = \mathbf{0}$, we have:

$$\mathbf{V} = (\mathbf{U}^\top\mathbf{U} + \beta c^2\mathbf{I})^{-1}\mathbf{U}^\top\mathbf{Z}, \tag{13}$$

and:

$$\beta c^2\mathbf{V}^\top\mathbf{V} = -\mathbf{V}^\top\mathbf{U}^\top(\mathbf{U}\mathbf{V} - \mathbf{Z}). \tag{14}$$

Denoting $\{\theta_i\}_{i=1}^{d_0}$ and $\{\omega_i\}_{i=1}^{\min(d_0,d_1)}$ with non-increasing order be the singular values of $\mathbf{Z}$ and $\mathbf{U}$, respectively. Let $\Theta$ and $\Omega$ be the singular matrices of $\mathbf{Z}$ and $\mathbf{U}$ with non-increasing diagonal, respectively. We also denote $\mathbf{\Sigma}' = \text{diag}(\sigma_1'^2, \ldots, \sigma_{d_1}'^2)$ as an rearrangements of $\mathbf{\Sigma}$ such that $\sigma_1'^2 \leq$

$\sigma_2'^2 \leq \ldots \leq \sigma_{d_1}'^2$. Thus, $\boldsymbol{\Sigma} = \mathbf{T}\boldsymbol{\Sigma}'\mathbf{T}^\top$ with some orthonormal matrix $\mathbf{T} \in \mathbb{R}^{d_1 \times d_1}$. It is clear that when all diagonal entries are distinct, $\mathbf{T}$ is a permutation matrix with only $\pm 1$'s and $0$'s. When there are some equal entries in $\boldsymbol{\Sigma}$, $\mathbf{T}$ may includes some orthonormal blocks on the diagonal when these equal entries are near to each other.

Plugging Eqn. (13) and Eqn. (14) into the loss function in Eqn. (11), we have:

$$
\begin{aligned}
\eta_{\text{dec}}^2 \mathcal{L}_{VAE} &= \|\mathbf{Z}\|_F^2 - \text{trace}(\mathbf{Z}\mathbf{V}^\top\mathbf{U}^\top) + \text{trace}(\mathbf{U}^\top\mathbf{U}\boldsymbol{\Sigma}) \\
&= \|\mathbf{Z}\|_F^2 - \text{trace}(\mathbf{Z}\mathbf{Z}^\top\mathbf{U}(\mathbf{U}^\top\mathbf{U} + \beta c^2\mathbf{I})^{-1}\mathbf{U}^\top) + \text{trace}(\mathbf{U}^\top\mathbf{U}\mathbf{T}\boldsymbol{\Sigma}'\mathbf{T}^\top) \\
&\geq \|\mathbf{Z}\|_F^2 - \text{trace}(\Theta\Theta^\top\Omega(\Omega^\top\Omega + \beta c^2\mathbf{I})^{-1}\Omega^\top) + \text{trace}(\Omega^\top\Omega\boldsymbol{\Sigma}') \\
&= \sum_{i=1}^{d_0} \theta_i^2 - \sum_{i=1}^{d_1} \frac{\omega_i^2\theta_i^2}{\omega_i^2 + \beta c^2} + \sum_{i=1}^{d_1} \omega_i^2\sigma_i'^2 \\
&= \sum_{i=d_1}^{d_0} \theta_i^2 + \sum_{i=1}^{d_1} \frac{\beta c^2\theta_i^2}{\omega_i^2 + \beta c^2} + \sum_{i=1}^{d_1} \omega_i^2\sigma_i'^2,
\end{aligned}
\tag{15}
$$

where we use two trace inequalities:

$$
\text{trace}(\mathbf{Z}\mathbf{Z}^\top\mathbf{U}(\mathbf{U}^\top\mathbf{U} + \beta c^2\mathbf{I})^{-1}\mathbf{U}^\top) \leq \text{trace}(\Theta\Theta^\top\Omega(\Omega^\top\Omega + \beta c^2\mathbf{I})^{-1}\Omega^\top), \tag{16}
$$

$$
\text{trace}(\mathbf{U}^\top\mathbf{U}\boldsymbol{\Sigma}) \geq \text{trace}(\Omega^\top\Omega\boldsymbol{\Sigma}'). \tag{17}
$$

The first inequality is from Von Neumann inequality with equality holds if and only if $\mathbf{Z}\mathbf{Z}^\top$ and $\mathbf{U}(\mathbf{U}^\top\mathbf{U} + \beta c^2\mathbf{I})^{-1}\mathbf{U}^\top$ are simultaneously ordering diagonalizable by some orthonormal matrix $\mathbf{R}$. The second inequality is Ruhe's trace inequality, with equality holds if and only if there exists an orthonormal matrix $\mathbf{T}$ that $\boldsymbol{\Sigma} = \mathbf{T}^\top\boldsymbol{\Sigma}'\mathbf{T}$ and $\mathbf{T}^\top\mathbf{U}^\top\mathbf{U}\mathbf{T}$ is diagonal matrix with decreasing entries.

By optimizing each $\omega_i$ in Eqn. (15), we have that:

$$
\omega_i^* = \sqrt{\max\left(0, \frac{\sqrt{\beta}c}{\sigma_1'}(\theta_i - \sqrt{\beta}c\sigma_1')\right)}. \tag{18}
$$

In order to let the inequalities above to become equality, we have that both $\mathbf{R}^\top\mathbf{U}\mathbf{U}^\top\mathbf{R}$ and $\mathbf{T}^\top\mathbf{U}^\top\mathbf{U}\mathbf{T}$ are diagonal matrix with decreasing entries. Thus, $\mathbf{U} = \mathbf{R}\Omega\mathbf{T}^\top$. From (13), by letting $\mathbf{Z} = \mathbf{R}\Theta\mathbf{S}$ with orthonormal matrix $\mathbf{S} \in \mathbb{R}^{d_0 \times d_0}$, we have the singular values (in decreasing order) of $\mathbf{V}$ as:

$$
\mathbf{V} = \mathbf{T}(\Omega^\top\Omega + \beta c^2\mathbf{I})^{-1}\Omega\Phi^{1/2}\mathbf{S},
$$

$$
\lambda_i^* = \sqrt{\max\left(0, \frac{\sigma_1'}{\sqrt{\beta}c}(\theta_i - \sqrt{\beta}c\sigma_1')\right)}. \tag{19}
$$

$\square$

**Remark 2.** *In the case when $\mathbf{V}$ has $d_1 - r$ zero singular values with $r := \text{rank}(\mathbf{V})$, it will depend on matrix $\mathbf{T}$ that decides whether posterior collapse happen or not. Specifically, if all values $\{\sigma_i\}_{i=1}^{d_1}$ are distinct, $\mathbf{T}$ is a permutation matrix and hence, $\mathbf{U}^\top\mathbf{U}$ is a diagonal matrix. Thus, using the same arguments as in the proof D.2, we have partial collapse will happen (although the variance of the posterior can be different from $\eta_{enc}^2$).*

*On the other hand, if $\boldsymbol{\Sigma}$ is chosen to be isotropic, $\mathbf{T}$ can be any orthonormal matrix and thus the number of zero rows of $\mathbf{V}$ can vary from $0$ (no posterior collapse) to $d_1 - r$ (partial posterior collapse). It is clear that when $\theta_i < c\sigma_i' \, \forall \, i$, $\mathbf{W} = 0$ and we observe a full posterior collapse.*

### D.2   LEARNABLE ENCODER VARIANCE $\boldsymbol{\Sigma}$

For learnable encoder variance $\boldsymbol{\Sigma}$ in linear standard VAE, we have the following results.

**Theorem 4** (Learnable $\boldsymbol{\Sigma}$). *Let $\mathbf{Z} := \mathbb{E}_x(x\tilde{x}^\top) = \mathbf{R}\Theta\mathbf{S}$ is the SVD of $\mathbf{Z}$ with singular values $\{\theta_i\}_{i=1}^{d_0}$ in non-increasing order and define $\mathbf{V} := \mathbf{W}\mathbf{P}_A\Phi^{1/2}$, the optimal solution of $(\mathbf{U}^*, \mathbf{W}^*, \boldsymbol{\Sigma}^*)$ of $\mathcal{L}_{VAE}$ is given by:*

$$
\mathbf{U}^* = \mathbf{R}\Omega\mathbf{T}^\top, \mathbf{V}^* = \mathbf{T}\Lambda\mathbf{S}^\top, \boldsymbol{\Sigma}^* = \beta\eta_{dec}^2(\mathbf{U}^\top\mathbf{U} + \beta c^2\mathbf{I})^{-1},
$$

*where $\mathbf{T} \in \mathbb{R}^{d_1 \times d_1}$ is an orthonormal matrices. The diagonal elements of $\Omega$ and $\Lambda$ are as follows, $\forall i \in [d_1]$:*

$$\omega_i^* = \frac{1}{\eta_{enc}}\sqrt{\max(0, \theta_i^2 - \beta\eta_{dec}^2)}, \quad \lambda_i^* = \frac{\eta_{enc}}{\theta_i}\sqrt{\max(0, \theta_i^2 - \beta\eta_{dec}^2)}.$$

*If $d_0 < d_1$, we denote $\theta_i = 0$ for $d_0 < i \leq d_1$.*

Now we prove Theorem 4.

*Proof of Theorem 4.* Recall the loss function in Eqn. (10):

$$\mathcal{L}_{VAE} = \frac{1}{\eta_{\text{dec}}^2}\Big[\|\mathbf{U}\mathbf{V} - \mathbf{Z}\|_F^2 + \text{trace}(\mathbf{U}^\top\mathbf{U}\boldsymbol{\Sigma}) + \beta c^2\|\mathbf{V}\|_F^2 + \beta c^2\,\text{trace}(\boldsymbol{\Sigma})\Big] - \beta\log|\boldsymbol{\Sigma}|. \quad (20)$$

We have at critical points of $\mathcal{L}_{VAE}$:

$$\frac{\partial\mathcal{L}}{\partial\boldsymbol{\Sigma}} = \frac{1}{\eta_{\text{dec}}^2}(\mathbf{U}^\top\mathbf{U} + \beta c^2\mathbf{I}) - \beta\boldsymbol{\Sigma}^{-1} = 0$$

$$\Rightarrow \boldsymbol{\Sigma} = \beta\eta_{\text{dec}}^2(\mathbf{U}^\top\mathbf{U} + \beta c^2\mathbf{I})^{-1}. \quad (21)$$

Plug $\boldsymbol{\Sigma} = \beta\eta_{\text{dec}}^2(\mathbf{U}^\top\mathbf{U} + \beta c^2\mathbf{I})^{-1}$ into $\mathcal{L}_{VAE}$ and dropping constant terms, we have:

$$\mathcal{L}'_{VAE} = \frac{1}{\eta_{\text{dec}}^2}\big[\|\mathbf{U}\mathbf{V} - \mathbf{Z}\|_F^2 + \beta c^2\|\mathbf{V}\|_F^2\big] + \beta\log|\mathbf{U}^\top\mathbf{U} + \beta c^2\mathbf{I}|. \quad (22)$$

At critical points of $\mathcal{L}'_{VAE}$:

$$\frac{1}{2}\frac{\partial\mathcal{L}'}{\partial\mathbf{V}} = \frac{1}{\eta_{\text{dec}}^2}(\mathbf{U}^\top(\mathbf{U}\mathbf{V} - \mathbf{Z}) + \beta c^2\mathbf{V}) = \mathbf{0},$$

$$\frac{1}{2}\frac{\partial\mathcal{L}'}{\partial\mathbf{U}} = \frac{1}{\eta_{\text{dec}}^2}(\mathbf{U}\mathbf{V} - \mathbf{Z})\mathbf{V}^\top + \mathbf{U}(\mathbf{U}^\top\mathbf{U} + \beta c^2\mathbf{I})^{-1} = \mathbf{0}. \quad (23)$$

From $\frac{\partial\mathcal{L}'}{\partial\mathbf{V}} = 0$, we have:

$$\mathbf{V} = (\mathbf{U}^\top\mathbf{U} + \beta c^2\mathbf{I})^{-1}\mathbf{U}^\top\mathbf{Z}, \quad (24)$$

and:

$$\beta c^2\mathbf{V}^\top\mathbf{V} = -\mathbf{V}^\top\mathbf{U}^\top(\mathbf{U}\mathbf{V} - \mathbf{Z}). \quad (25)$$

Denoting $\{\theta_i\}_{i=1}^{d_0}$ and $\{\omega_i\}_{i=1}^{\min(d_0,d_1)}$ with decreasing order be the singular values of $\mathbf{Z}$ and $\mathbf{U}$, respectively. Let $\Theta$ and $\Omega$ be the singular matrices of $\mathbf{Z}$ and $\mathbf{U}$, respectively. Plug Eqn. (24) and (25) to $\mathcal{L}'$, we have:

$$\begin{aligned}
\mathcal{L}'_{VAE} &= \frac{1}{\eta_{\text{dec}}^2}[\|\mathbf{Z}\|_F^2 - \text{trace}(\mathbf{Z}\mathbf{V}^\top\mathbf{U}^\top)] + \beta\log|\mathbf{U}^\top\mathbf{U} + \beta c^2\mathbf{I}| \\
&= \frac{1}{\eta_{\text{dec}}^2}[\|\mathbf{Z}\|_F^2 - \text{trace}(\mathbf{Z}\mathbf{Z}^\top\mathbf{U}(\mathbf{U}^\top\mathbf{U} + \beta c^2\mathbf{I})^{-1}\mathbf{U}^\top)] + \beta\log|\mathbf{U}^\top\mathbf{U} + \beta c^2\mathbf{I}|, \\
&\geq \frac{1}{\eta_{\text{dec}}^2}[\|\mathbf{Z}\|_F^2 - \text{trace}(\Theta\Theta^\top\Omega(\Omega^\top\Omega + \beta c^2\mathbf{I})\Omega^\top)] + \beta\log|\mathbf{U}^\top\mathbf{U} + \beta c^2\mathbf{I}| \\
&= \frac{1}{\eta_{\text{dec}}^2}\left[\sum_{i=1}^{d_0}\theta_i^2 - \sum_{i=1}^{d_1}\frac{\omega_i^2\theta_i^2}{\omega_i^2 + \beta c^2}\right] + \sum_{i=1}^{d_1}\beta\log(\omega_i^2 + \beta c^2) \\
&= \frac{1}{\eta_{\text{dec}}^2}\left[\sum_{i=d_1}^{d_0}\theta_i^2 + \sum_{i=1}^{d_1}\frac{\beta c^2\theta_i^2}{\omega_i^2 + \beta c^2}\right] + \sum_{i=1}^{d_1}\beta\log(\omega_i^2 + \beta c^2), \quad (26)
\end{aligned}$$

where we use Von Neumann inequality for $\mathbf{Z}\mathbf{Z}^\top$ and $\mathbf{U}(\mathbf{U}^\top\mathbf{U}+c^2\mathbf{I})^{-1}\mathbf{U}^\top$. The equality condition holds if these two symmetric matrices are simultaneous ordering diagonalizable (i.e., there exists an orthonormal matrix diagonalize both matrices such that the eigenvalues order of both matrices are in decreasing order).

Consider the function:

$$h(\omega) = \frac{c^2\theta^2/\eta_{\text{dec}}^2}{\omega^2 + \beta c^2} + \log(\omega^2 + \beta c^2), \quad \omega \geq 0 \tag{27}$$

This function is minimized at $\omega^* = \sqrt{\frac{c^2}{\eta_{\text{dec}}^2}(\theta^2 - \beta\eta_{\text{dec}}^2)}$ if $\theta^2 \geq \eta_{\text{dec}}^2$. Otherwise, if $\theta^2 < \eta_{\text{dec}}^2$, $\omega^* = 0$. Applying this result for each $\omega_i$ in Eqn. (26), we have:

$$\omega_i^* = \frac{1}{\eta_{\text{enc}}}\sqrt{\max(0, \theta_i^2 - \beta\eta_{\text{dec}}^2)}, \, \forall\, i \in [d_1] \tag{28}$$

Denote $\{\lambda_i\}_{i=1}^{\min(d_0,d_1)}$ as the singular values of $\mathbf{V}$ and $\Lambda$ as corresponding singular matrix. At the minimizer of $\mathcal{L}_{VAE}$, $\mathbf{U}$ and $\mathbf{Z}$ shares a set of left singular vectors due to equality of Von Neumann trace inequality. Thus, using these shared singular vectors, from Eqn. (24), the singular values of $\mathbf{V}$ are:

$$\lambda_i^* = \frac{\eta_{\text{enc}}}{\theta_i}\sqrt{\max(0, \theta_i^2 - \beta\eta_{\text{dec}}^2)}. \tag{29}$$

From Eqn. (21), now we consider $\Sigma$ that are diagonal in the set of global parameters of the loss function. We have:

$$\sigma_i' = \begin{cases} \sqrt{\beta}\eta_{\text{enc}}\eta_{\text{dec}}/\theta_i, & \text{if } \theta_i \geq \sqrt{\beta}\eta_{\text{dec}} \\ \eta_{\text{enc}}, & \text{if } \theta_i < \sqrt{\beta}\eta_{\text{dec}} \end{cases}, \tag{30}$$

where $\{\sigma_i'\}_{i=1}^{d_1}$ is a permutation of $\{\sigma_i\}_{i=1}^{d_1}$. Since $\Sigma = \beta\eta_{\text{dec}}^2(\mathbf{U}^\top\mathbf{U} + \beta c^2\mathbf{I})^{-1}$ at optimal, if we choose $\Sigma$ to be diagonal, we have $\mathbf{U}^\top\mathbf{U}$ diagonal at the global optimum. Thus, $\mathbf{U}$ can be decomposed as $\mathbf{U} = \mathbf{R}\Omega'$ with orthonormal matrix $\mathbf{R} \in \mathbb{R}^{D_0 \times D_0}$ and $\Omega'$ is a diagonal matrix with diagonal entries a permutation of diagonal entries of $\Omega$. Hence, $\mathbf{U}$ will have $d_1 - r$ zero columns with $r := \text{rank}(\mathbf{U})$. From the loss function in Eqn. (10), we see that the corresponding rows of $\mathbf{V}$ with the zero columns of $\mathbf{U}$, will have no effect on the term $\|\mathbf{U}\mathbf{V} - \mathbf{Z}\|$. Thus, the only term in the loss function that relates to these rows of $\mathbf{V}$ is $\|\mathbf{V}\|_F$. To minimize $\|\mathbf{V}\|_F$, these rows of $\mathbf{V}$ will converge to zero rows. Therefore, from $\mathbf{W}x = \mathbf{V}\tilde{x}$, we see that these zero rows will correspond with latent dimensions that collapse to the prior distribution of that dimension (partial posterior collapse). $\square$

## E  PROOFS FOR CONDITIONAL VAE

In this section, we prove Theorem 2 in main paper.

We consider the conditional VAE as described in Section 4.1. For any input data $(x, y)$, we have:

**Encoder:** $q(z|x,y) = \mathcal{N}(\mathbf{W}_1 x + \mathbf{W}_2 y, \Sigma), \mathbf{W}_1 \in \mathbb{R}^{d_1 \times D_0}, \mathbf{W}_2 \in \mathbb{R}^{d_1 \times D_2}$.

**Decoder:** $p(y|x,z) = \mathcal{N}(\mathbf{U}_1 z + \mathbf{U}_2 x, \eta_{\text{dec}}^2\mathbf{I}), \mathbf{U}_1 \in \mathbb{R}^{D_2 \times d_1}, \mathbf{U}_2 \in \mathbb{R}^{D_2 \times D_0}$.

**Prior:** $p(z|x) = \mathcal{N}(0, \eta_{\text{enc}}^2\mathbf{I})$,

note that we can write $z = \mathbf{W}_1 x + \mathbf{W}_2 y + \xi$ with $\xi \sim \mathcal{N}(0, \Sigma)$ for a given $(x, y)$.

To train CVAE, we minimize the following loss function (Sohn et al., 2015; Doersch, 2016; Walker et al., 2016):

$$\mathcal{L}_{CVAE} = -\mathbb{E}_{x,y}\big[\mathbb{E}_{q_\phi(z|x,y)}[p_\theta(y|x,z)] + \beta D_{KL}(q_\phi(z|x,y)\|p(z|x))\big]$$

$$= \mathbb{E}_{x,y,\xi}\left[\frac{1}{\eta_{\text{dec}}^2}\|\mathbf{U}_1 z + \mathbf{U}_2 x - y\|^2 - \beta\xi^\top\Sigma^{-1}\xi - \beta\log|\Sigma| + \frac{\beta}{\eta_{\text{enc}}^2}\|z\|^2\right]$$

$$= \frac{1}{\eta_{\text{dec}}^2}\mathbb{E}_{x,y}\big[\|(\mathbf{U}_1\mathbf{W}_1 + \mathbf{U}_2)x + (\mathbf{U}_1\mathbf{W}_2 - \mathbf{I})y\|^2 + \text{trace}(\mathbf{U}_1\Sigma_1\mathbf{U}_1^\top)$$

$$+ \beta c^2(\|\mathbf{W}_1 x + \mathbf{W}_2 y\|^2 + \text{trace}(\Sigma))\big] - \beta\log|\Sigma|,$$

where $c := \eta_{\text{dec}}^2/\eta_{\text{enc}}^2$. Note that we have dropped the multiplier $1/2$ and constants in the above derivation.

*Proof of Theorem 2.* For brevity in the subsequent analysis, we further denote $\mathbf{V}_1 = \mathbf{W}_1 \mathbf{P}_A \Phi^{1/2} \in \mathbb{R}^{d_1 \times d_0}$, $\mathbf{V}_2 = \mathbf{W}_2 \mathbf{P}_B \Psi^{1/2} \in \mathbb{R}^{d_1 \times d_2}$, $\mathbf{T}_2 = \mathbf{U}_2 \mathbf{P}_A \Phi^{1/2} \in \mathbb{R}^{D_2 \times d_0}$ and $\mathbf{D} = \mathbf{P}_B \Psi^{1/2} \in \mathbb{R}^{D_2 \times d_2}$, we have:

$$\mathbb{E}(\|(\mathbf{U}_1 \mathbf{W}_1 + \mathbf{U}_2)x + (\mathbf{U}_1 \mathbf{W}_2 - \mathbf{I})y\|^2) = \mathbb{E}(\|(\mathbf{U}_1 \mathbf{W}_1 + \mathbf{U}_2)\mathbf{P}_A \Phi^{1/2}\tilde{x} + (\mathbf{U}_1 \mathbf{W}_2 - \mathbf{I})\mathbf{P}_B \Psi^{1/2}\tilde{y}\|^2)$$
$$= \|\mathbf{U}_1 \mathbf{V}_1 + \mathbf{T}_2\|_F^2 + \|\mathbf{U}_1 \mathbf{V}_2 - \mathbf{D}\|_F^2 + 2\operatorname{trace}((\mathbf{U}_1 \mathbf{V}_1 + \mathbf{T}_2)\mathbf{Z}(\mathbf{U}_1 \mathbf{V}_2 - \mathbf{D})^\top),$$

$$\mathbb{E}(\|\mathbf{W}_1 x + \mathbf{W}_2 y\|^2) = \mathbb{E}(\|\mathbf{V}_1 \tilde{x} + \mathbf{V}_2 \tilde{y}\|^2) = \|\mathbf{V}_1\|_F^2 + \|\mathbf{V}_2\|_F^2 + 2\operatorname{trace}(\mathbf{V}_1 \mathbf{Z} \mathbf{V}_2^\top).$$

Therefore, the negative ELBO becomes

$$\mathcal{L}_{\text{CVAE}}(\mathbf{U}_1, \mathbf{V}_1, \mathbf{V}_2, \mathbf{T}_2, \mathbf{\Sigma}) =$$
$$\frac{1}{\eta_{\text{dec}}^2}\Bigg[ \underbrace{\|\mathbf{U}_1 \mathbf{V}_1 + \mathbf{T}_2\|_F^2 + \|\mathbf{U}_1 \mathbf{V}_2 - \mathbf{D}\|_F^2 + 2\operatorname{trace}((\mathbf{U}_1 \mathbf{V}_2 - \mathbf{D})^\top(\mathbf{U}_1 \mathbf{V}_1 + \mathbf{T}_2)\mathbf{Z})}_{= \|(\mathbf{U}_1 \mathbf{V}_1 + \mathbf{T}_2)\tilde{x} + (\mathbf{U}_1 \mathbf{V}_2 - \mathbf{I})\tilde{y}\|^2}$$
$$+ \operatorname{trace}((\mathbf{U}_1^\top \mathbf{U}_1 + \beta c^2 \mathbf{I})\mathbf{\Sigma}_1) + \beta c^2 \underbrace{(\|\mathbf{V}_1\|_F^2 + \|\mathbf{V}_2\|_F^2 + 2\operatorname{trace}(\mathbf{V}_1 \mathbf{Z} \mathbf{V}_2^\top))}_{= \|\mathbf{V}_1 \tilde{x} + \mathbf{V}_2 \tilde{y}\|^2}\Bigg] - \beta \log|\mathbf{\Sigma}|.$$

Next, we have at critical points of $\mathcal{L}_{CVAE}$:

$$\frac{\partial \mathcal{L}}{\partial \mathbf{\Sigma}} = \frac{1}{\eta_{\text{dec}}^2}(\mathbf{U}_1^\top \mathbf{U}_1 + \beta c^2 \mathbf{I}) - \beta \mathbf{\Sigma}^{-1} = \mathbf{0}$$
$$\Rightarrow \mathbf{\Sigma} = \beta \eta_{\text{dec}}^2 (\mathbf{U}_1^\top \mathbf{U}_1 + \beta c^2 \mathbf{I})^{-1} \tag{31}$$

Plugging $\mathbf{\Sigma} = \beta \eta_{\text{dec}}^2 (\mathbf{U}_1^\top \mathbf{U}_1 + \beta c^2 \mathbf{I})^{-1}$ in the loss function $\mathcal{L}_{CVAE}$ and dropping some constants, we have:

$$\mathcal{L}'_{CVAE} = \frac{1}{\eta_{\text{dec}}^2}\Bigg[\|\mathbf{U}_1 \mathbf{V}_1 + \mathbf{T}_2\|_F^2 + \|\mathbf{U}_1 \mathbf{V}_2 - \mathbf{D}\|_F^2 + 2\operatorname{trace}((\mathbf{U}_1 \mathbf{V}_1 + \mathbf{T}_2)\mathbf{Z}(\mathbf{U}_1 \mathbf{V}_2 - \mathbf{D})^\top)$$
$$+ \beta c^2 \left(\|\mathbf{V}_1\|_F^2 + \|\mathbf{V}_2\|_F^2 + 2\operatorname{trace}(\mathbf{V}_1 \mathbf{Z} \mathbf{V}_2^\top)\right)\Bigg] + \beta \log|\mathbf{U}_1^\top \mathbf{U}_1 + \beta c^2 \mathbf{I}|. \tag{32}$$

We have, at critical points of $\mathcal{L}'_{CVAE}$:

$$\frac{\eta_{\text{dec}}^2}{2} \frac{\partial \mathcal{L}'}{\partial \mathbf{T}_2} = (\mathbf{U}_1 \mathbf{V}_1 + \mathbf{T}_2) + (\mathbf{U}_1 \mathbf{V}_2 - \mathbf{D})\mathbf{Z}^\top = \mathbf{0}.$$
$$\frac{\eta_{\text{dec}}^2}{2} \frac{\partial \mathcal{L}'}{\partial \mathbf{V}_1} = \mathbf{U}_1^\top(\mathbf{U}_1 \mathbf{V}_1 + \mathbf{T}_2) + \mathbf{U}_1^\top(\mathbf{U}_1 \mathbf{V}_2 - \mathbf{D})\mathbf{Z}^\top + \beta c^2 \mathbf{V}_1 + \beta c^2 \mathbf{V}_2 \mathbf{Z}^\top = \mathbf{0}.$$
$$\frac{\eta_{\text{dec}}^2}{2} \frac{\partial \mathcal{L}'}{\partial \mathbf{V}_2} = \mathbf{U}_1^\top(\mathbf{U}_1 \mathbf{V}_2 - \mathbf{D}) + \mathbf{U}_1^\top(\mathbf{U}_1 \mathbf{V}_1 + \mathbf{T}_2)\mathbf{Z} + \beta c^2 \mathbf{V}_2 + \beta c^2 \mathbf{V}_1 \mathbf{Z} = \mathbf{0}.$$
$$\frac{\eta_{\text{dec}}^2}{2} \frac{\partial \mathcal{L}'}{\partial \mathbf{U}_1} = (\mathbf{U}_1 \mathbf{V}_1 + \mathbf{T}_2)\mathbf{V}_1^\top + (\mathbf{U}_1 \mathbf{V}_2 - \mathbf{D})\mathbf{V}_2^\top + \mathbf{U}_1(\mathbf{V}_2 \mathbf{Z}^\top \mathbf{V}_1^\top + \mathbf{V}_1 \mathbf{Z} \mathbf{V}_2^\top) - \mathbf{D}\mathbf{Z}^\top \mathbf{V}_1^\top$$
$$+ \mathbf{T}_2 \mathbf{Z} \mathbf{V}_2^\top + \beta \eta_{\text{dec}}^2 \mathbf{U}_1(\mathbf{U}_1^\top \mathbf{U}_1 + \beta c^2 \mathbf{I})^{-1} = \mathbf{0}. \tag{33}$$

From $\frac{\partial \mathcal{L}'}{\partial \mathbf{T}_2} = \mathbf{0}$, we have:

$$\mathbf{T}_2 = -\mathbf{U}_1 \mathbf{V}_1 - (\mathbf{U}_1 \mathbf{V}_2 - \mathbf{D})\mathbf{Z}^\top. \tag{34}$$

From $\frac{\partial \mathcal{L}'}{\partial \mathbf{V}_1} = \mathbf{0}$ and Eqn. (34), we have:

$$\mathbf{V}_1 + \mathbf{V}_2 \mathbf{Z}^\top = \mathbf{0}. \tag{35}$$

From $\frac{\partial \mathcal{L}'}{\partial \mathbf{V}_2} = \mathbf{0}$ and Eqn. (35), we have:

$$
\begin{aligned}
&\mathbf{U}_1^\top (\mathbf{U}_1 \mathbf{V}_2 - \mathbf{D}) - \mathbf{U}_1^\top (\mathbf{U}_1 \mathbf{V}_2 - \mathbf{D}) \mathbf{Z}^\top \mathbf{Z} + \beta c^2 \mathbf{V}_2 - \beta c^2 \mathbf{V}_2 \mathbf{Z}^\top \mathbf{Z} = \mathbf{0} \\
&\Leftrightarrow \mathbf{U}_1^\top (\mathbf{U}_1 \mathbf{V}_2 - \mathbf{D})(\mathbf{I} - \mathbf{Z}^\top \mathbf{Z}) = -\beta c^2 \mathbf{V}_2 (\mathbf{I} - \mathbf{Z}^\top \mathbf{Z}) \\
&\Leftrightarrow \mathbf{U}_1^\top (\mathbf{U}_1 \mathbf{W}_2 - \mathbf{I})\mathbf{D}(\mathbf{I} - \mathbf{Z}^\top \mathbf{Z}) = -\beta c^2 \mathbf{W}_2 \mathbf{D}(\mathbf{I} - \mathbf{Z}^\top \mathbf{Z}) \\
&\Leftrightarrow \mathbf{W}_2 \mathbf{D}(\mathbf{I} - \mathbf{Z}^\top \mathbf{Z}) = (\mathbf{U}_1^\top \mathbf{U}_1 + \beta c^2 \mathbf{I})^{-1} \mathbf{U}_1^\top \mathbf{D}(\mathbf{I} - \mathbf{Z}^\top \mathbf{Z}) \\
&\Rightarrow \mathbf{W}_2 \mathbf{E} = (\mathbf{U}_1^\top \mathbf{U}_1 + \beta c^2 \mathbf{I})^{-1} \mathbf{U}_1^\top \mathbf{E},
\end{aligned}
\tag{36}
$$

where we define $\mathbf{E} := \mathbf{D}(\mathbf{I} - \mathbf{Z}^\top \mathbf{Z})\mathbf{D}^\top$ for brevity.

Using above substitutions in Eqn. (34), Eqn. (35) and Eqn. (36), we have:

$$
\begin{aligned}
&\|\mathbf{U}_1 \mathbf{V}_1 + \mathbf{T}_2\|_F^2 + \|\mathbf{U}_1 \mathbf{V}_2 - \mathbf{D}\|_F^2 + 2\operatorname{trace}((\mathbf{U}_1 \mathbf{V}_1 + \mathbf{T}_2)\mathbf{Z}(\mathbf{U}_1 \mathbf{V}_2 - \mathbf{D})^\top) \\
&= \|(\mathbf{U}_1 \mathbf{V}_2 - \mathbf{D})\mathbf{Z}^\top\|_F^2 + \|\mathbf{U}_1 \mathbf{V}_2 - \mathbf{D}\|_F^2 - 2\operatorname{trace}((\mathbf{U}_1 \mathbf{V}_2 - \mathbf{D})\mathbf{Z}^\top \mathbf{Z}(\mathbf{U}_1 \mathbf{V}_2 - \mathbf{D})^\top) \\
&= \operatorname{trace}((\mathbf{U}_1 \mathbf{V}_2 - \mathbf{D})^\top (\mathbf{U}_1 \mathbf{V}_2 - \mathbf{D})(\mathbf{I} - \mathbf{Z}^\top \mathbf{Z})) \\
&= \operatorname{trace}((\mathbf{U}_1 \mathbf{W}_2 - \mathbf{I})^\top (\mathbf{U}_1 \mathbf{W}_2 - \mathbf{I})\mathbf{D}(\mathbf{I} - \mathbf{Z}^\top \mathbf{Z})\mathbf{D}^\top) \\
&= \operatorname{trace}(\mathbf{W}_2^\top \mathbf{U}_1^\top \mathbf{U}_1 \mathbf{W}_2 \mathbf{E}) - 2\operatorname{trace}(\mathbf{U}_1 \mathbf{W}_2 \mathbf{E}) + \operatorname{trace}(\mathbf{E}).
\end{aligned}
$$

$$
\begin{aligned}
&\|\mathbf{V}_1\|_F^2 + \|\mathbf{V}_2\|_F^2 + 2\operatorname{trace}(\mathbf{V}_1 \mathbf{Z} \mathbf{V}_2^\top) = \|\mathbf{V}_2 \mathbf{Z}^\top\|_F^2 + \|\mathbf{V}_2\|_F^2 - 2\operatorname{trace}(\mathbf{V}_2 \mathbf{Z}^\top \mathbf{Z} \mathbf{V}_2^\top) \\
&= \operatorname{trace}(\mathbf{V}_2 (\mathbf{I} - \mathbf{Z}^\top \mathbf{Z})\mathbf{V}_2^\top) = \operatorname{trace}(\mathbf{W}_2 \mathbf{E} \mathbf{W}_2^\top).
\end{aligned}
$$

$$
\begin{aligned}
&\operatorname{trace}(\mathbf{W}_2^\top \mathbf{U}_1^\top \mathbf{U}_1 \mathbf{W}_2 \mathbf{E}) - 2\operatorname{trace}(\mathbf{U}_1 \mathbf{W}_2 \mathbf{E}) + \beta c^2 \operatorname{trace}(\mathbf{W}_2 \mathbf{E} \mathbf{W}_2^\top) \\
&= \operatorname{trace}((\mathbf{U}_1^\top \mathbf{U}_1 + \beta c^2 \mathbf{I})\mathbf{W}_2 \mathbf{E} \mathbf{W}_2^\top) - 2\operatorname{trace}(\mathbf{U}_1 \mathbf{W}_2 \mathbf{E}) \\
&= \operatorname{trace}(\mathbf{U}_1^\top \mathbf{E} \mathbf{U}_1 (\mathbf{U}_1^\top \mathbf{U}_1 + \beta c^2 \mathbf{I})^{-1}) - 2\operatorname{trace}(\mathbf{U}_1 (\mathbf{U}_1^\top \mathbf{U}_1 + \beta c^2 \mathbf{I})^{-1} \mathbf{U}_1^\top \mathbf{E}) \\
&= -\operatorname{trace}(\mathbf{U}_1 (\mathbf{U}_1^\top \mathbf{U}_1 + \beta c^2 \mathbf{I})^{-1} \mathbf{U}_1^\top \mathbf{E}).
\end{aligned}
$$

Denote $\{\lambda_i\}_{i=1}^{d_1}$ and $\{\theta_i\}_{i=1}^{d_2}$ be the singular values of $\mathbf{U}_1$ and $\mathbf{E}$ in non-increasing order, respectively. If $d_2 < d_1$, we denote $\theta_i = 0$, with $d_2 < i \leq d_1$. We now have, after dropping constant $\operatorname{trace}(\mathbf{E})$:

$$
\begin{aligned}
\mathcal{L}'_{CVAE} &= -\frac{1}{\eta_{\text{dec}}^2} \operatorname{trace}(\mathbf{U}_1 (\mathbf{U}_1^\top \mathbf{U}_1 + \beta c^2 \mathbf{I})^{-1} \mathbf{U}_1^\top \mathbf{E}) + \beta \log |\mathbf{U}_1^\top \mathbf{U}_1 + \beta c^2 \mathbf{I}| \\
&\geq -\frac{1}{\eta_{\text{dec}}^2} \sum_{i=1}^{d_1} \frac{\lambda_i^2 \theta_i^2}{\lambda_i^2 + \beta c^2} + \sum_{i=1}^{d_1} \beta \log(\lambda_i^2 + \beta c^2) \\
&= -\frac{1}{\eta_{\text{dec}}^2} \sum_{i=1}^{d_1} \theta_i^2 + \frac{\beta}{\eta_{\text{dec}}^2} \sum_{i=1}^{d_1} \frac{c^2 \theta_i^2}{\lambda_i^2 + \beta c^2} + \sum_{i=1}^{d_1} \beta \log(\lambda_i^2 + \beta c^2),
\end{aligned}
$$

where we used Von Neumann trace inequality for $\mathbf{U}_1 (\mathbf{U}_1^\top \mathbf{U}_1 + \beta c^2 \mathbf{I})^{-1} \mathbf{U}_1^\top$ and $\mathbf{E}$. We consider the function below:

$$
g(t) = \frac{1}{\eta_{\text{dec}}^2} \frac{c^2 \theta^2}{t} + \log(t), \ t \geq \beta c^2. \tag{37}
$$

It is easy to see that $g(t)$ is minimized at $t^* = \frac{c^2 \theta^2}{\eta_{\text{dec}}^2}$ if $\theta^2 \geq \beta \eta_{\text{dec}}^2$. Otherwise, if $\theta^2 < \beta \eta_{\text{dec}}^2$, $t^* = \beta c^2$. If $\theta = 0$, clearly $\log(\lambda^2 + \beta c^2)$ is minimized at $\lambda = 0$. Applying this result for each $\lambda_i$, we have:

$$
\lambda_i^* = \frac{1}{\eta_{\text{enc}}} \sqrt{\max(0, \theta_i^2 - \beta \eta_{\text{dec}}^2)}, \ \forall \, i \in [d_1], \tag{38}
$$

note that the RHS can also be applied when $\theta_i = 0$ for $i \in [d_1]$.

Consider the global parameters that $\Sigma$ is diagonal. The entries of $\Sigma$ can be calculated from Eqn. (31):

$$\sigma_i' = \begin{cases} \sqrt{\beta}\eta_{\text{enc}}\eta_{\text{dec}}/\theta_i, & \text{if } \theta_i \geq \sqrt{\beta}\eta_{\text{dec}} \\ \eta_{\text{enc}}, & \text{if } \theta_i < \sqrt{\beta}\eta_{\text{dec}} \end{cases}, \tag{39}$$

where $\{\sigma_i'\}$ is a permutation of $\{\sigma_i\}$. $\qquad\qquad\square$

**Remark 3.** *Posterior collapse also exists in CVAE. When at the global parameters such that $\Sigma$ is diagonal, we have $\mathbf{U}_1^\top \mathbf{U}_1$ is diagonal. Thus, $\mathbf{U}_1$ can be decomposed as $\mathbf{U}_1 = \mathbf{R}\Omega'$ with orthonormal matrix $\mathbf{R} \in \mathbb{R}^{D_0 \times D_0}$ and $\Omega'$ is a diagonal matrix with diagonal entries are a permutation of diagonal entries of $\Omega$. Hence, $\mathbf{U}_1$ will have $d_1 - r$ zero columns with $r := \text{rank}(\mathbf{U}_1)$.*

*If there is a column (says, $i$-th column) of $\mathbf{U}_1$ is zero column, then in the loss function in Eqn. (5), the $i$-th row of $\mathbf{W}_1 x + \mathbf{W}_2 y$ will only appear in the term $\mathbb{E}_{x,y}(\|\mathbf{W}_1 x + \mathbf{W}_2 y\|_F^2)$. Thus, at the global minimum, $\mathbb{E}_{x,y}(\|\mathbf{W}_1 x + \mathbf{W}_2 y\|_F^2)$ will be pushed to $0$. Thus, for each pair of input $(x, y)$, we have $\mathbf{w}_i^{1\top} x + \mathbf{w}_i^{2\top} y = 0$ (posterior collapse).*

## F PROOFS FOR MARKOVIAN HIERARCHICAL VAE WITH 2 LATENTS

In this section, we prove Theorem 3 in Section F.1. We also analyze the case that both encoder variances $\Sigma_1$ and $\Sigma_2$ are unlearnable isotropic in Section F.2. We have:

$$\begin{aligned}
\textbf{Encoder: } & q(z_1|x) \sim \mathcal{N}(\mathbf{W}_1 x, \Sigma_1), \mathbf{W}_1 \in \mathbb{R}^{d_1 \times d_0}, \Sigma_1 \in \mathbb{R}^{d_1 \times d_1} \\
& q(z_2|z_1) \sim \mathcal{N}(\mathbf{W}_2 z_1, \Sigma_2), \mathbf{W}_2 \in \mathbb{R}^{d_2 \times d_1}, \Sigma_2 \in \mathbb{R}^{d_2 \times d_2} \\
\textbf{Decoder: } & p(z_1|z_2) \sim \mathcal{N}(\mathbf{U}_2 z_2, \eta_{\text{dec}}^2 \mathbf{I}), \mathbf{U}_2 \in \mathbb{R}^{d_1 \times d_2}, \\
& p(y|z_1) \sim \mathcal{N}(\mathbf{U}_1 z_1, \eta_{\text{dec}}^2 \mathbf{I}), \mathbf{U}_1 \in \mathbb{R}^{D \times d_1}. \\
\textbf{Prior: } & p(z_2) \sim \mathcal{N}(0, \eta_{\text{enc}}^2 \mathbf{I}),
\end{aligned} \tag{40}$$

Let $\mathbf{A} := \mathbb{E}_x(xx^\top) = \mathbf{P}_A \Phi \mathbf{P}_A^\top$, $\tilde{x} = \Phi^{-1/2} \mathbf{P}_A^\top x$ and $\mathbf{Z} := \mathbb{E}_x(x\tilde{x}^\top) \in \mathbb{R}^{D_0 \times d_0}$. Also, let $\mathbf{V}_1 = \mathbf{W}_1 \mathbf{P}_A \Phi^{1/2} \in \mathbb{R}^{d_1 \times D}$, thus $\mathbf{V}_1 \mathbf{V}_1^\top = \mathbf{W}_1 \mathbf{A} \mathbf{W}_1^\top = \mathbb{E}_x(\mathbf{W}_1 x x^\top \mathbf{W}_1^\top)$.

We minimize the negative ELBO loss function for MHVAE with 2 layers of latent:

$$\begin{aligned}
\mathcal{L}_{HVAE} &= -\mathbb{E}_x \big[ \mathbb{E}_{q_\phi(z_1|x)q_\phi(z_2|z_1)}(\log p_\theta(x|z_1)) - \beta_1 \mathbb{E}_{q_\phi(z_2|z_1)}(D_{\text{KL}}(q_\phi(z_1|x)\|p_\theta(z_1|z_2))) \\
&\quad - \beta_2 \mathbb{E}_x \mathbb{E}_{q_\phi(z_1|x)}(D_{\text{KL}}(q_\phi(z_2|z_1)\|p_\theta(z_2))) \big]. \\
&= -\mathbb{E}_x \mathbb{E}_{q_\phi(z_1|x)q_\phi(z_2|z_1)} \big[ \log p(x|z_1) + \beta_1 \log p(z_1|z_2) + \beta_2 \log p(z_2) - \beta_1 \log q(z_1|x) - \beta_2 \log q(z_2|z_1) \big] \\
&= \mathbb{E}_{x,\xi_1,\xi_2} \Big[ \frac{1}{\eta_{\text{dec}}^2} \|\mathbf{U}_1 z_1 - x\|^2 + \frac{\beta_1}{\eta_{\text{dec}}^2} \|\mathbf{U}_2 z_2 - z_1\|^2 + \frac{\beta_2}{\eta_{\text{enc}}^2} \|\mathbf{W}_2 z_1 + \xi_2\|^2 - \beta_1 \xi_1^\top \Sigma_1^{-1} \xi_1 \\
&\quad - \beta_1 \log|\Sigma_1| - \beta_2 \xi_2^\top \Sigma_2^{-1} \xi_2 - \beta_2 \log|\Sigma_2| \Big] \\
&= \frac{1}{\eta_{\text{dec}}^2} \mathbb{E}_x \big[ \|\mathbf{U}_1 \mathbf{W}_1 x - x\|^2 + \text{trace}(\mathbf{U}_1 \Sigma_1 \mathbf{U}_1^\top) + \beta_1 \|\mathbf{U}_2 \mathbf{W}_2 \mathbf{W}_1 x - \mathbf{W}_1 x\|^2 + \beta_1 \text{trace}(\mathbf{U}_2 \Sigma_2 \mathbf{U}_2^\top) \\
&\quad + \beta_1 \text{trace}((\mathbf{U}_2 \mathbf{W}_2 - \mathbf{I})\Sigma_1(\mathbf{U}_2 \mathbf{W}_2 - \mathbf{I})^\top) + c^2 \beta_2 \big( \|\mathbf{W}_2 \mathbf{W}_1 x\|^2 + \text{trace}(\mathbf{W}_2 \Sigma_1 \mathbf{W}_2^\top) + \text{trace}(\Sigma_2) \big) \big] \\
&\quad - \beta_1 d_1 - \beta_2 d_2 - \beta_1 \log|\Sigma_1| - \beta_2 \log|\Sigma_2| \\
&= \frac{1}{\eta_{\text{dec}}^2} \big[ \|\mathbf{U}_1 \mathbf{V}_1 - \mathbf{Z}\|_F^2 + \text{trace}(\mathbf{U}_1^\top \mathbf{U}_1 \Sigma_1) + \beta_1 \|(\mathbf{U}_2 \mathbf{W}_2 - \mathbf{I})\mathbf{V}_1\|_F^2 + \beta_1 \text{trace}(\mathbf{U}_2^\top \mathbf{U}_2 \Sigma_2) \\
&\quad + \beta_1 \text{trace}((\mathbf{U}_2 \mathbf{W}_2 - \mathbf{I})^\top (\mathbf{U}_2 \mathbf{W}_2 - \mathbf{I})\Sigma_1) + c^2 \beta_2 \|\mathbf{W}_2 \mathbf{V}_1\|_F^2 + c^2 \beta_2 \text{trace}(\mathbf{W}_2^\top \mathbf{W}_2 \Sigma_1) + c^2 \beta_2 \text{trace}(\Sigma_2) \big] \\
&\quad - \beta_1 d_1 - \beta_2 d_2 - \beta_1 \log|\Sigma_1| - \beta_2 \log|\Sigma_2|.
\end{aligned}$$

### F.1 LEARNABLE $\boldsymbol{\Sigma}_2$ AND UNLEARNABLE ISOTROPIC $\boldsymbol{\Sigma}_1$

*Proof of Theorem 3.* With unlearnable isotropic $\boldsymbol{\Sigma}_1 = \sigma_1^2 \mathbf{I}$, the loss function $\mathcal{L}_{HVAE}$ becomes (after dropping some constants):

$$\mathcal{L}_{HVAE} = \frac{1}{\eta_{\text{dec}}^2}\big[\|\mathbf{U}_1\mathbf{V}_1 - \mathbf{Z}\|_F^2 + \beta_1\|(\mathbf{U}_2\mathbf{W}_2 - \mathbf{I})\mathbf{V}_1\|_F^2 + \text{trace}(\mathbf{U}_1^\top\mathbf{U}_1\boldsymbol{\Sigma}_1) + \beta_1\,\text{trace}(\mathbf{U}_2^\top\mathbf{U}_2\boldsymbol{\Sigma}_2)$$

$$+ \beta_1\,\text{trace}((\mathbf{U}_2\mathbf{W}_2 - \mathbf{I})^\top(\mathbf{U}_2\mathbf{W}_2 - \mathbf{I})\boldsymbol{\Sigma}_1) + \beta_2 c^2\|\mathbf{W}_2\mathbf{V}_1\|_F^2 + \beta_2 c^2\,\text{trace}(\mathbf{W}_2^\top\mathbf{W}_2\boldsymbol{\Sigma}_1)$$

$$+ \beta_2 c^2\,\text{trace}(\boldsymbol{\Sigma}_2)\big] - \beta_2\log|\boldsymbol{\Sigma}_2|.$$

Taking the derivative of $\mathcal{L}_{HVAE}$ w.r.t $\boldsymbol{\Sigma}_2$:

$$\frac{1}{2}\frac{\partial\mathcal{L}}{\partial\boldsymbol{\Sigma}_2} = \frac{\beta_1}{\eta_{\text{dec}}^2}(\mathbf{U}_2^\top\mathbf{U}_2 + c^2\frac{\beta_2}{\beta_1}\mathbf{I}) - \beta_2\boldsymbol{\Sigma}_2^{-1} = \mathbf{0}$$

$$\Rightarrow \boldsymbol{\Sigma}_2 = \frac{\beta_2}{\beta_1}\eta_{\text{dec}}^2(\mathbf{U}_2^\top\mathbf{U}_2 + c^2\frac{\beta_2}{\beta_1}\mathbf{I})^{-1}. \tag{41}$$

Plugging this into the loss function and dropping constants yields:

$$\mathcal{L}'_{HVAE} = \frac{1}{\eta_{\text{dec}}^2}\big[\|\mathbf{U}_1\mathbf{V}_1 - \mathbf{Z}\|_F^2 + \beta_1\|(\mathbf{U}_2\mathbf{W}_2 - \mathbf{I})\mathbf{V}_1\|_F^2 + \text{trace}(\mathbf{U}_1^\top\mathbf{U}_1\boldsymbol{\Sigma}_1)$$

$$+ \beta_1\,\text{trace}((\mathbf{U}_2\mathbf{W}_2 - \mathbf{I})^\top(\mathbf{U}_2\mathbf{W}_2 - \mathbf{I})\boldsymbol{\Sigma}_1) + \beta_2 c^2\|\mathbf{W}_2\mathbf{V}_1\|_F^2 + \beta_2 c^2\,\text{trace}(\mathbf{W}_2^\top\mathbf{W}_2\boldsymbol{\Sigma}_1)\big]$$

$$+ \beta_2\log|\mathbf{U}_2^\top\mathbf{U}_2 + c^2\frac{\beta_2}{\beta_1}\mathbf{I}|. \tag{42}$$

At critical points of $\mathcal{L}'_{HVAE}$:

$$\frac{\eta_{\text{dec}}^2}{2}\frac{\partial\mathcal{L}'}{\partial\mathbf{V}_1} = \mathbf{U}_1^\top(\mathbf{U}_1\mathbf{V}_1 - \mathbf{Z}) + \beta_1(\mathbf{U}_2\mathbf{W}_2 - \mathbf{I})^\top(\mathbf{U}_2\mathbf{W}_2 - \mathbf{I})\mathbf{V}_1 + \beta_2 c^2\mathbf{W}_2^\top\mathbf{W}_2\mathbf{V}_1 = \mathbf{0}, \tag{43}$$

$$\frac{\eta_{\text{dec}}^2}{2}\frac{\partial\mathcal{L}'}{\partial\mathbf{U}_1} = (\mathbf{U}_1\mathbf{V}_1 - \mathbf{Z})\mathbf{V}_1^\top + \mathbf{U}_1\boldsymbol{\Sigma}_1 = \mathbf{0}, \tag{44}$$

$$\frac{\eta_{\text{dec}}^2}{2}\frac{\partial\mathcal{L}'}{\partial\mathbf{U}_2} = \beta_1(\mathbf{U}_2\mathbf{W}_2 - \mathbf{I})\mathbf{V}_1\mathbf{V}_1^\top\mathbf{W}_2^\top + \beta_2\eta_{\text{dec}}^2\mathbf{U}_2(\mathbf{U}_2^\top\mathbf{U}_2 + c^2\frac{\beta_2}{\beta_1}\mathbf{I})^{-1} + \beta_1(\mathbf{U}_2\mathbf{W}_2 - \mathbf{I})\boldsymbol{\Sigma}_1\mathbf{W}_2^\top = \mathbf{0}, \tag{45}$$

$$\frac{\eta_{\text{dec}}^2}{2}\frac{\partial\mathcal{L}'}{\partial\mathbf{W}_2} = \beta_1\mathbf{U}_2^\top(\mathbf{U}_2\mathbf{W}_2 - \mathbf{I})\mathbf{V}_1\mathbf{V}_1^\top + \beta_1\mathbf{U}_2^\top(\mathbf{U}_2\mathbf{W}_2 - \mathbf{I})\boldsymbol{\Sigma}_1 + \beta_2 c^2\mathbf{W}_2\mathbf{V}_1\mathbf{V}_1^\top + \beta_2 c^2\mathbf{W}_2\boldsymbol{\Sigma}_1 = \mathbf{0}. \tag{46}$$

From Eqn. (46), we have:

$$\big(\beta_1\mathbf{U}_2^\top(\mathbf{U}_2\mathbf{W}_2 - \mathbf{I}) + \beta_2 c^2\mathbf{W}_2\big)(\mathbf{V}_1\mathbf{V}_1^\top + \boldsymbol{\Sigma}_1) = \mathbf{0}$$

$$\Rightarrow \mathbf{W}_2 = -\frac{\beta_1}{\beta_2 c^2}\mathbf{U}_2^\top(\mathbf{U}_2\mathbf{W}_2 - \mathbf{I}) \quad \text{(since } \mathbf{V}_1\mathbf{V}_1^\top + \boldsymbol{\Sigma}_1 \text{ is positive definite)}$$

$$\Rightarrow \mathbf{W}_2 = (\mathbf{U}_2^\top\mathbf{U}_2 + c^2\frac{\beta_2}{\beta_1}\mathbf{I})^{-1}\mathbf{U}_2^\top$$

$$\Rightarrow \mathbf{W}_2 = \mathbf{U}_2^\top(\mathbf{U}_2\mathbf{U}_2^\top + c^2\frac{\beta_2}{\beta_1}\mathbf{I})^{-1}. \tag{47}$$

$$\Rightarrow \mathbf{U}_2\mathbf{W}_2 - \mathbf{I} = -c^2\frac{\beta_2}{\beta_1}(\mathbf{U}_2\mathbf{U}_2^\top + c^2\frac{\beta_2}{\beta_1}\mathbf{I})^{-1}. \tag{48}$$

From Eqn. (44), we have:

$$\mathbf{U}_1 = \mathbf{Z}\mathbf{V}_1^\top(\mathbf{V}_1\mathbf{V}_1^\top + \boldsymbol{\Sigma}_1)^{-1}, \tag{49}$$

From Eqn. (43), with the use of Eqn. (48) and (47), we have:

$$\mathbf{U}_1^\top \mathbf{U}_1 \mathbf{V}_1 - \mathbf{U}_1^\top \mathbf{Z} + c^4 \frac{\beta_2^2}{\beta_1}(\mathbf{U}_2\mathbf{U}_2^\top + c^2\frac{\beta_2}{\beta_1}\mathbf{I})^{-2}\mathbf{V}_1$$

$$+ c^2\beta_2(\mathbf{U}_2\mathbf{U}_2^\top + c^2\frac{\beta_2}{\beta_1}\mathbf{I})^{-1}\mathbf{U}_2\mathbf{U}_2^\top(\mathbf{U}_2\mathbf{U}_2^\top + c^2\frac{\beta_2}{\beta_1}\mathbf{I})^{-1}\mathbf{V}_1 = \mathbf{0}$$

$$\Rightarrow \mathbf{U}_1^\top \mathbf{U}_1 \mathbf{V}_1 + c^2\beta_2(\mathbf{U}_2\mathbf{U}_2^\top + c^2\frac{\beta_2}{\beta_1}\mathbf{I})^{-1}\mathbf{V}_1 = \mathbf{U}_1^\top \mathbf{Z} \tag{50}$$

$$\Rightarrow c^2\beta_2(\mathbf{U}_2\mathbf{U}_2^\top + c^2\frac{\beta_2}{\beta_1}\mathbf{I})^{-1}\mathbf{V}_1\mathbf{V}_1^\top = \mathbf{U}_1^\top \mathbf{Z}\mathbf{V}_1^\top - \mathbf{U}_1^\top\mathbf{U}_1\mathbf{V}_1\mathbf{V}_1^\top = \mathbf{U}_1^\top(\mathbf{Z} - \mathbf{U}_1\mathbf{V}_1)\mathbf{V}_1^\top = \mathbf{U}_1^\top\mathbf{U}_1\mathbf{\Sigma}_1, \tag{51}$$

where the last equality is from Eqn. (44).

We have the assumption that $\mathbf{\Sigma}_1 = \sigma_1^2\mathbf{I}$. From Eqn. (51), its LHS is symmetric because the RHS is symmetric, hence two symmetric matrices $\mathbf{V}_1\mathbf{V}_1^\top$ and $(\mathbf{U}_2\mathbf{U}_2^\top + c^2\frac{\beta_2}{\beta_1}\mathbf{I})^{-1}$ commute. Hence, they are orthogonally simultaneous diagonalizable. As a consequence, there exists an orthonormal matrix $\mathbf{P} \in \mathbb{R}^{d_1 \times d_1}$ such that $\mathbf{P}^\top\mathbf{V}_1\mathbf{V}_1^\top\mathbf{P}$ and $\mathbf{P}^\top(\mathbf{U}_2\mathbf{U}_2^\top + c^2\frac{\beta_2}{\beta_1}\mathbf{I})^{-1}\mathbf{P} = (\mathbf{P}^\top\mathbf{U}_2\mathbf{U}_2^\top\mathbf{P} + c^2\frac{\beta_2}{\beta_1}\mathbf{I})^{-1}$ are diagonal matrices (thus, $\mathbf{P}^\top\mathbf{U}_2\mathbf{U}_2^\top\mathbf{P}$ is also diagonal). Note that we choose $\mathbf{P}$ such that the eigenvalues of $\mathbf{V}_1\mathbf{V}_1^\top$ are in decreasing order, by altering the columns of $\mathbf{P}$. Therefore, we can write the SVD of $\mathbf{V}_1$ and $\mathbf{U}_2$ as $\mathbf{V}_1 = \mathbf{P}\mathbf{\Lambda}\mathbf{Q}^\top$ and $\mathbf{U}_2 = \mathbf{P}\mathbf{\Omega}\mathbf{N}^\top$ with orthonormal matrices $\mathbf{P} \in \mathbb{R}^{d_1 \times d_1}, \mathbf{Q} \in \mathbb{R}^{d_0 \times d_0}, \mathbf{N} \in \mathbb{R}^{d_2 \times d_2}$ and singular matrices $\mathbf{\Lambda} \in \mathbb{R}^{d_1 \times d_0}, \mathbf{\Omega} \in \mathbb{R}^{d_1 \times d_2}$. We denote the singular values of $\mathbf{V}_1, \mathbf{U}_2, \mathbf{Z}$ are $\{\lambda_i\}_{i=1}^{\min(d_0,d_1)}$, $\{\omega_i\}_{i=1}^{\min(d_1,d_2)}$ and $\{\theta_i\}_{i=1}^{d_0}$, respectively.

Next, by multiplying $\mathbf{U}_2^\top$ to the left of Eqn. (45), we have:

$$\beta_2\eta_{\text{dec}}^2\mathbf{U}_2^\top\mathbf{U}_2(\mathbf{U}_2^\top\mathbf{U}_2 + c^2\frac{\beta_2}{\beta_1}\mathbf{I})^{-1} = -\beta_1\mathbf{U}_2^\top(\mathbf{U}_2\mathbf{W}_2 - \mathbf{I})(\mathbf{V}_1\mathbf{V}_1^\top + \sigma_1^2\mathbf{I})\mathbf{W}_2^\top$$

$$= c^2\beta_2\mathbf{U}_2^\top(\mathbf{U}_2\mathbf{U}_2^\top + c^2\frac{\beta_2}{\beta_1}\mathbf{I})^{-1}(\mathbf{V}_1\mathbf{V}_1^\top + \sigma_1^2\mathbf{I})(\mathbf{U}_2\mathbf{U}_2^\top + c^2\frac{\beta_2}{\beta_1}\mathbf{I})^{-\top}\mathbf{U}_2. \tag{52}$$

Plugging the SVD forms of $\mathbf{V}_1$ and $\mathbf{U}_2$ in Eqn. (52) yields:

$$\eta_{\text{dec}}^2\mathbf{N}\mathbf{\Omega}^\top\mathbf{\Omega}(\mathbf{\Omega}^\top\mathbf{\Omega} + c^2\frac{\beta_2}{\beta_1}\mathbf{I})^{-1}\mathbf{N}^\top = c^2\mathbf{N}\mathbf{\Omega}^\top(\mathbf{\Omega}\mathbf{\Omega}^\top + c^2\frac{\beta_2}{\beta_1}\mathbf{I})^{-1}(\mathbf{\Lambda}\mathbf{\Lambda}^\top + \sigma_1^2\mathbf{I})(\mathbf{\Omega}\mathbf{\Omega}^\top + c^2\frac{\beta_2}{\beta_1}\mathbf{I})^{-1}\mathbf{\Omega}\mathbf{N}^\top$$

$$\Rightarrow \frac{\eta_{\text{dec}}^2\omega_i^2}{\omega_i^2 + c^2\frac{\beta_2}{\beta_1}} = c^2\frac{\omega_i^2(\lambda_i^2 + \sigma_1^2)}{(\omega_i^2 + c^2\frac{\beta_2}{\beta_1})^2}, \quad \forall i \in [\min(d_1,d_2)]. \tag{53}$$

$$\Rightarrow \omega_i = 0 \text{ or } \omega_i^2 + c^2\frac{\beta_2}{\beta_1} = \frac{c^2}{\eta_{\text{dec}}^2}(\lambda_i^2 + \sigma_1^2), \quad \forall i \in [\min(d_1,d_2)]. \tag{54}$$

Using above equations, the loss function can be simplified into:

$$\mathcal{L}'_{HVAE} = \frac{1}{\eta_{\text{dec}}^2}\left[\|\mathbf{Z}\|_F^2 - \text{trace}(\mathbf{U}_1^\top\mathbf{Z}\mathbf{V}_1^\top) + c^2\beta_2 \text{ trace}\left((\mathbf{U}_2\mathbf{U}_2^\top + c^2\frac{\beta_2}{\beta_1}\mathbf{I})^{-1}(\mathbf{V}_1\mathbf{V}_1^\top + \mathbf{\Sigma}_1)\right)\right]$$

$$+ \beta_2\log|\mathbf{U}_2^\top\mathbf{U}_2 + c^2\frac{\beta_2}{\beta_1}\mathbf{I}|. \tag{55}$$

Computing the components in $\mathcal{L}'_{HVAE}$, we have:

$$\log |\mathbf{U}_2^\top \mathbf{U}_2 + c^2 \frac{\beta_2}{\beta_1}\mathbf{I}| = \sum_{i=1}^{d_2} \log(\omega_i^2 + c^2\frac{\beta_2}{\beta_1}),$$

$$\text{trace}(\mathbf{U}_1^\top \mathbf{Z}\mathbf{V}_1^\top) = \text{trace}(\mathbf{V}_1^\top \mathbf{U}_1^\top \mathbf{Z})$$

$$= \text{trace}(\mathbf{V}_1^\top (\mathbf{V}_1\mathbf{V}_1^\top + \sigma_1^2\mathbf{I})^{-1}\mathbf{V}_1\mathbf{Z}^\top\mathbf{Z})$$

$$\le \sum_{i=1}^{d_0} \frac{\lambda_i^2\theta_i^2}{\lambda_i^2 + \sigma_1^2},$$

$$\text{trace}\left((\mathbf{U}_2\mathbf{U}_2^\top + c^2\frac{\beta_2}{\beta_1}\mathbf{I})^{-1}(\mathbf{V}_1\mathbf{V}_1^\top + \sigma_1^2\mathbf{I})\right) = \sum_{i=1}^{d_1} \frac{\lambda_i^2 + \sigma_1^2}{\omega_i^2 + c^2\frac{\beta_2}{\beta_1}},$$

where we used Von Neumann inequality for $\mathbf{V}_1^\top (\mathbf{V}_1\mathbf{V}_1^\top + \sigma_1^2\mathbf{I})^{-1}\mathbf{V}_1$ and $\mathbf{Z}^\top\mathbf{Z}$ with equality holds if and only if these two matrices are simultaneous ordering diagonalizable by some orthonormal matrix $\mathbf{R}$.

We assume $d_1 = d_2 \le d_0$ for now. Therefore, we have:

$$\eta_{\text{dec}}^2\mathcal{L}_{HVAE} \ge \sum_{i=1}^{d_0} \theta_i^2 - \sum_{i=1}^{d_1} \frac{\lambda_i^2\theta_i^2}{\lambda_i^2 + \sigma_1^2} + \sum_{i=1}^{d_1} c^2\beta_2 \frac{\lambda_i^2 + \sigma_1^2}{\omega_i^2 + c^2\frac{\beta_2}{\beta_1}} + \beta_2\eta_{\text{dec}}^2 \sum_{i=1}^{d_1} \log(\omega_i^2 + c^2\frac{\beta_2}{\beta_1})$$

$$= \sum_{i=d_1}^{d_0} \theta_i^2 + \sum_{i=1}^{d_1} \underbrace{\left(\frac{\sigma_1^2\theta_i^2}{\lambda_i^2 + \sigma_1^2} + c^2\beta_2 \frac{\lambda_i^2 + \sigma_1^2}{\omega_i^2 + c^2\frac{\beta_2}{\beta_1}} + \beta_2\eta_{\text{dec}}^2 \log(\omega_i^2 + c^2\frac{\beta_2}{\beta_1})\right)}_{g(\lambda_i,\omega_i)}. \quad (56)$$

Consider the function:

$$g(\lambda,\omega) = \frac{\sigma_1^2\theta^2}{\lambda^2 + \sigma_1^2} + c^2\beta_2 \frac{\lambda^2 + \sigma_1^2}{\omega^2 + c^2\frac{\beta_2}{\beta_1}} + \beta_2\eta_{\text{dec}}^2 \log(\omega^2 + c^2\frac{\beta_2}{\beta_1}).$$

We consider two cases:

- If $\omega = 0$, we have $g(\lambda, 0) = \frac{\sigma_1^2\theta^2}{\lambda^2 + \sigma_1^2} + \beta_1(\lambda^2 + \sigma_1^2) + \beta_2\eta_{\text{dec}}^2 \log(c^2\frac{\beta_2}{\beta_1})$. It is easy to see that $g(\lambda, 0)$ is minimized at:

$$\lambda^* = \sqrt{\max\left(0, \frac{\sigma_1}{\sqrt{\beta_1}}\left(\theta - \sqrt{\beta_1}\sigma_1\right)\right)}. \quad (57)$$

- If $\omega^2 + c^2\frac{\beta_2}{\beta_1} = c^2/\eta_{\text{dec}}^2(\lambda^2 + \sigma_1^2) = \frac{\lambda^2 + \sigma_1^2}{\eta_{\text{enc}}^2}$, then we have:

$$g = \frac{\sigma_1^2\theta^2}{\lambda^2 + \sigma_1^2} + \beta_2\eta_{\text{dec}}^2 \log(\lambda^2 + \sigma_1^2) + \beta_2\eta_{\text{dec}}^2 - \beta_2\eta_{\text{dec}}^2 \log(\eta_{\text{enc}}^2), \quad (58)$$

Since we require both $t := \lambda^2 + \sigma_1^2 \ge \sigma_1^2$ and $\omega^2 = (\lambda^2 + \sigma_1^2 - \frac{\beta_2}{\beta_1}\eta_{\text{dec}}^2)/\eta_{\text{enc}}^2 = (t - \frac{\beta_2}{\beta_1}\eta_{\text{dec}}^2)/\eta_{\text{enc}}^2 \ge 0$, we consider three cases:

  - If $\sigma_1^2 \ge \frac{\beta_2}{\beta_1}\eta_{\text{dec}}^2$, $g$ is minimized at the minima $t_0 = \frac{\sigma_1^2\theta^2}{\beta_2\eta_{\text{dec}}^2}$ if $t_0 \ge \sigma_1^2$, otherwise, $g$ is minimized at $t = \sigma_1^2$. Thus:

$$\lambda^* = \frac{\sigma_1}{\sqrt{\beta_2}\eta_{\text{dec}}}\sqrt{\max(0, \theta^2 - \beta_2\eta_{\text{dec}}^2)},$$

$$\omega^* = \sqrt{\max\left(\frac{\sigma_1^2 - \frac{\beta_2}{\beta_1} * \eta_{\text{dec}}^2}{\eta_{\text{enc}}^2}, \frac{\sigma_1^2\theta^2}{\beta_2\eta_{\text{enc}}^2\eta_{\text{dec}}^2} - c^2\frac{\beta_2}{\beta_1}\right)}. \quad (59)$$

    We can easily check that this solution is optimal after comparing with the case $\omega = 0$ above.

- If $\sigma_1^2 < \frac{\beta_2}{\beta_1}\eta_{\text{dec}}^2$, and if the minima of $g$ at $t_0 = \frac{\sigma_1^2\theta^2}{\beta_2\eta_{\text{dec}}^2} < \frac{\beta_2}{\beta_1}\eta_{\text{dec}}^2$, $g$ is minimized at $t = \frac{\beta_2}{\beta_1}\eta_{\text{dec}}^2$, thus $\omega = 0$ and we know from the case $\omega = 0$ above that:

$$\lambda^* = \sqrt{\max\left(0, \frac{\sigma_1}{\sqrt{\beta_1}}\left(\theta - \sqrt{\beta_1}\sigma_1\right)\right)},$$
$$\omega^* = 0. \tag{60}$$

- If $\sigma_1^2 < \frac{\beta_2}{\beta_1}\eta_{\text{dec}}^2$, and if the minima of $g$ at $t_0 = \frac{\sigma_1^2\theta^2}{\beta_2\eta_{\text{dec}}^2} \geq \frac{\beta_2}{\beta_1}\eta_{\text{dec}}^2$, $g$ is minimized at $t_0 = \lambda^2 + \sigma_1^2 = \frac{\sigma_1^2\theta^2}{\beta_2\eta_{\text{dec}}^2}$ and thus:

$$\lambda^* = \frac{\sigma_1}{\sqrt{\beta_2}\eta_{\text{dec}}}\sqrt{\theta^2 - \beta_2\eta_{\text{dec}}^2},$$
$$\omega^* = \sqrt{\frac{\sigma_1^2\theta^2}{\beta_2\eta_{\text{enc}}^2\eta_{\text{dec}}^2} - \frac{\beta_2}{\beta_1}c^2}. \tag{61}$$

We can easily check this solution is optimal in this case after comparing with the case $\omega = 0$ above.

We call the optimal singulars above the "standard" case. For other relations between $d_0, d_1$ and $d_2$, we consider below cases:

- If $d_0 < d_1 < d_2$: For index $i \leq d_0$, the optimal values follow standard case. For $d_0 < i \leq d_1$, clearly $\lambda_i = 0$ (recall $\mathbf{V}_1 \in \mathbb{R}^{d_1 \times d_0}$), then $\omega_i = \sqrt{\max(0, \frac{\sigma_i^2 - \frac{\beta_2}{\beta_1}*\eta_{\text{dec}}^2}{\eta_{\text{enc}}^2})}$. For $i > d_1$, it is clear that $\lambda_i = \omega_i = 0$.

- If $d_0 < d_2 \leq d_1$: For index $i \leq d_0$, the optimal values follow standard case. For $d_0 < i \leq d_2$, clearly $\lambda_i = 0$, then $\omega_i = \sqrt{\max(0, \frac{\sigma_i^2 - \frac{\beta_2}{\beta_1}*\eta_{\text{dec}}^2}{\eta_{\text{enc}}^2})}$. For $i > d_2$, it is clear that $\lambda_i = \omega_i = 0$.

- If $d_1 \leq \min(d_0, d_2)$: For index $i \leq d_1$, the optimal values follow standard case. For $d_1 < i$, $\lambda_i = \omega_i = 0$ (recall $\mathbf{V}_1 \in \mathbb{R}^{d_1 \times d_0}$ and $\mathbf{U}_2 \in \mathbb{R}^{d_1 \times d_2}$ ).

- If $d_2 \leq d_0 < d_1$: For index $i \leq d_2$, the optimal values follow standard case. For $d_2 < i \leq d_0$, $\omega_i = 0$ and $\lambda_i = \sqrt{\max\left(0, \frac{\sigma_1}{\sqrt{\beta_1}}\left(\theta - \sqrt{\beta_1}\sigma_1\right)\right)}$. For $i > d_0$, $\lambda_i = \omega_i = 0$.

- If $d_2 < d_1 \leq d_0$: For index $i \leq d_2$, the optimal values follow standard case. For $d_2 < i \leq d_1$, $\omega_i = 0$ and $\lambda_i = \sqrt{\max\left(0, \frac{\sigma_1}{\sqrt{\beta_1}}\left(\theta - \sqrt{\beta_1}\sigma_1\right)\right)}$. For $i > d_1$, $\lambda_i = \omega_i = 0$

$\square$

**Remark 4.** *If $\Sigma_2$ is diagonal, we can easily calculate the optimal $\Sigma_2^*$ via the equation $\Sigma_2^* = \eta_{dec}^2(\mathbf{U}_2^\top\mathbf{U}_2 + c^2\mathbf{I})^{-1}$. Also, in this case, $\mathbf{U}_2$ will have orthogonal columns and can be written as $\mathbf{U}_2 = \mathbf{T}\Omega'$ with orthonormal matrix $\mathbf{T}$ and $\Omega'$ is a diagonal matrix. Therefore, $\mathbf{W}_2 = \mathbf{U}_2^\top(\mathbf{U}_2\mathbf{U}_2^\top + c^2\mathbf{I})^{-1} = \Omega'(\Omega'\Omega'^\top + c^2\mathbf{I})^{-1}\mathbf{T}^\top$ will have zero rows (posterior collapse in second latent variable) if $\text{rank}(\mathbf{W}_2) = \text{rank}(\mathbf{U}_2) < d_2$.*

*For the first latent variable, posterior collapse may not exist. Indeed, we have $\mathbf{V}_1$ has the form $\mathbf{V}_1 = \mathbf{P}\Lambda\mathbf{R}^\top$. Thus, $\mathbf{P}$ will determine the number of zero rows of $\mathbf{V}_1$. The number of zero rows of $\mathbf{V}_1$ will vary from 0 to $d_1 - \text{rank}(\mathbf{V}_1)$ since $\mathbf{P}$ can be chosen arbitrarily.*

### F.2 UNLEARNABLE ISOTROPIC ENCODER VARIANCES $\Sigma_1, \Sigma_2$

In this section, with the setting as in Eqn. (40), we derive the results for MHVAE two latents with both encoder variances are unlearnable and isotropic $\Sigma_1 = \sigma_1^2 \mathbf{I}$, $\Sigma_2 = \sigma_2^2 \mathbf{I}$. We have the following results.

**Theorem 5.** *Assume* $\Sigma_1 = \sigma_1^2 \mathbf{I}$, $\Sigma_2 = \sigma_2^2 \mathbf{I}$ *for some* $\sigma_1, \sigma_2 > 0$. *Assuming* $d_0 \geq d_1 = d_2$, *the optimal solution of* $(\mathbf{U}_1^*, \mathbf{U}_2^*, \mathbf{V}_1^*, \mathbf{W}_2^*)$ *of* $\mathcal{L}_{HVAE}$ *is given by:*

$$\mathbf{V}_1^* = \mathbf{P}\Lambda\mathbf{R}^\top, \mathbf{U}_2^* = \mathbf{P}\Omega\mathbf{Q}^\top, \mathbf{W}_2^* = \mathbf{U}_2^{*\top}(\mathbf{U}_2^*\mathbf{U}_2^{*\top} + c^2\mathbf{I})^{-1}, \mathbf{U}_1^* = \mathbf{Z}\mathbf{V}_1^{*\top}(\mathbf{V}_1^*\mathbf{V}_1^{*\top} + \Sigma_1)^{-1},$$

*where* $\mathbf{Z} = \mathbf{R}\Theta\mathbf{S}^\top$ *is the SVD of* $\mathbf{Z}$ *and orthonormal matrix* $\mathbf{P} \in \mathbb{R}^{d_1 \times d_1}$. *The diagonal elements of* $\Lambda$ *and* $\Omega$ *are as follows, with* $i \in [d_1]$:

- *If* $\theta_i \geq \max\{\sqrt{\sqrt{\beta_1\beta_2}c\sigma_1\sigma_2}, \frac{\beta_2}{\sqrt{\beta_1}}\frac{c^2\sigma_2^2}{\sigma_1}\}$:

$$\lambda_i^* = \sqrt[3]{\frac{\sigma_1^2}{\sqrt{\beta_1\beta_2}c\sigma_2}}\sqrt{\sqrt[3]{\theta_i^4} - \sqrt[3]{\beta_1\beta_2c^2\sigma_1^2\sigma_2^2}},$$

$$\omega_i^* = \sqrt[3]{\frac{\sqrt{\beta_2}c\sigma_1}{\beta_1\sigma_2^2}}\sqrt{\sqrt[3]{\theta_i^2} - \sqrt[3]{\frac{\beta_2^2c^4\sigma_2^4}{\beta_1\sigma_1^2}}}.$$

- *If* $\theta_i < \max\{\sqrt{\sqrt{\beta_1\beta_2}c\sigma_1\sigma_2}, \frac{\beta_2}{\sqrt{\beta_1}}\frac{c^2\sigma_2^2}{\sigma_1}\}$ *and* $\sqrt{\beta_1}\sigma_1 \geq \sqrt{\beta_2}c\sigma_2$:

$$\lambda_i^* = 0,$$

$$\omega_i^* = \sqrt{\sqrt{\frac{\beta_2}{\beta_1}}\frac{c\sigma_1}{\sigma_2} - \frac{\beta_2}{\beta_1}c^2}.$$

- *If* $\theta_i < \max\{\sqrt{\sqrt{\beta_1\beta_2}c\sigma_1\sigma_2}, \frac{\beta_2}{\sqrt{\beta_1}}\frac{c^2\sigma_2^2}{\sigma_1}\}$ *and* $\sqrt{\beta_1}\sigma_1 < \sqrt{\beta_2}c\sigma_2$:

$$\lambda_i^* = \sqrt{\max\left(0, \frac{\sigma_1}{\sqrt{\beta_1}}(\theta - \sqrt{\beta_1}\sigma_1)\right)},$$

$$\omega_i^* = 0.$$

Now we prove Theorem 5.

*Proof of Theorem 5.* The loss function is this setting will be:

$$\mathcal{L}_{HVAE} = \frac{1}{\eta_{\text{dec}}^2}\Big[\|\mathbf{U}_1\mathbf{V}_1 - \mathbf{Z}\|_F^2 + \beta_1\|(\mathbf{U}_2\mathbf{W}_2 - \mathbf{I})\mathbf{V}_1\|_F^2 + \text{trace}(\mathbf{U}_1^\top\mathbf{U}_1\Sigma_1) + \beta_1\,\text{trace}(\mathbf{U}_2^\top\mathbf{U}_2\Sigma_2)$$

$$+ \beta_1\,\text{trace}((\mathbf{U}_2\mathbf{W}_2 - \mathbf{I})^\top(\mathbf{U}_2\mathbf{W}_2 - \mathbf{I})\Sigma_1) + \beta_2c^2\|\mathbf{W}_2\mathbf{V}_1\|_F^2 + \beta_2c^2\,\text{trace}(\mathbf{W}_2^\top\mathbf{W}_2\Sigma_1)\Big].$$

We have at critical points of $\mathcal{L}_{HVAE}$:

$$\frac{\eta_{\text{dec}}^2}{2}\frac{\partial\mathcal{L}}{\partial\mathbf{V}_1} = \mathbf{U}_1^\top(\mathbf{U}_1\mathbf{V}_1 - \mathbf{Z}) + \beta_1(\mathbf{U}_2\mathbf{W}_2 - \mathbf{I})^\top(\mathbf{U}_2\mathbf{W}_2 - \mathbf{I})\mathbf{V}_1 + c^2\beta_2\mathbf{W}_2^\top\mathbf{W}_2\mathbf{V}_1 = \mathbf{0},$$

$$\tag{62}$$

$$\frac{\eta_{\text{dec}}^2}{2}\frac{\partial\mathcal{L}}{\partial\mathbf{U}_1} = (\mathbf{U}_1\mathbf{V}_1 - \mathbf{Z})\mathbf{V}_1^\top + \mathbf{U}_1\Sigma_1 = \mathbf{0}, \tag{63}$$

$$\frac{\eta_{\text{dec}}^2}{2}\frac{\partial\mathcal{L}}{\partial\mathbf{U}_2} = \beta_1(\mathbf{U}_2\mathbf{W}_2 - \mathbf{I})\mathbf{V}_1\mathbf{V}_1^\top\mathbf{W}_2^\top + \beta_1\mathbf{U}_2\Sigma_2 + \beta_1(\mathbf{U}_2\mathbf{W}_2 - \mathbf{I})\Sigma_1\mathbf{W}_2^\top = \mathbf{0}, \tag{64}$$

$$\frac{\eta_{\text{dec}}^2}{2}\frac{\partial\mathcal{L}}{\partial\mathbf{W}_2} = \beta_1\mathbf{U}_2^\top(\mathbf{U}_2\mathbf{W}_2 - \mathbf{I})\mathbf{V}_1\mathbf{V}_1^\top + \beta_1\mathbf{U}_2^\top(\mathbf{U}_2\mathbf{W}_2 - \mathbf{I})\Sigma_1 + c^2\beta_2\mathbf{W}_2\mathbf{V}_1\mathbf{V}_1^\top + c^2\beta_2\mathbf{W}_2\Sigma_1 = \mathbf{0}.$$

$$\tag{65}$$

From Eqn. (65), we have:

$$\left(\beta_1 \mathbf{U}_2^\top (\mathbf{U}_2 \mathbf{W}_2 - \mathbf{I}) + c^2 \beta_2 \mathbf{W}_2\right)(\mathbf{V}_1 \mathbf{V}_1^\top + \mathbf{\Sigma}_1) = \mathbf{0}$$

$$\Rightarrow \mathbf{W}_2 = -\frac{\beta_1}{c^2 \beta_2} \mathbf{U}_2^\top (\mathbf{U}_2 \mathbf{W}_2 - \mathbf{I}) \quad (\text{since } \mathbf{V}_1 \mathbf{V}_1^\top + \mathbf{\Sigma}_1 \text{ is PD})$$

$$\Rightarrow \mathbf{W}_2 = (\mathbf{U}_2^\top \mathbf{U}_2 + c^2 \frac{\beta_2}{\beta_1} \mathbf{I})^{-1} \mathbf{U}_2^\top$$

$$\Rightarrow \mathbf{W}_2 = \mathbf{U}_2^\top (\mathbf{U}_2 \mathbf{U}_2^\top + c^2 \frac{\beta_2}{\beta_1} \mathbf{I})^{-1}. \tag{66}$$

$$\Rightarrow \mathbf{U}_2 \mathbf{W}_2 - \mathbf{I} = -c^2 \frac{\beta_2}{\beta_1} (\mathbf{U}_2 \mathbf{U}_2^\top + c^2 \frac{\beta_2}{\beta_1} \mathbf{I})^{-1}. \tag{67}$$

From Eqn. (63), we have:

$$\mathbf{U}_1 = \mathbf{Z} \mathbf{V}_1^\top (\mathbf{V}_1 \mathbf{V}_1^\top + \mathbf{\Sigma}_1)^{-1}. \tag{68}$$

From Eqn. (62) with the use of Eqn. (67) and (66), we have:

$$\mathbf{U}_1^\top \mathbf{U}_1 \mathbf{V}_1 - \mathbf{U}_1^\top \mathbf{Z} + c^4 \frac{\beta_2^2}{\beta_1} (\mathbf{U}_2 \mathbf{U}_2^\top + c^2 \frac{\beta_2}{\beta_1} \mathbf{I})^{-2} \mathbf{V}_1$$

$$+ c^2 \beta_2 (\mathbf{U}_2 \mathbf{U}_2^\top + c^2 \frac{\beta_2}{\beta_1} \mathbf{I})^{-1} \mathbf{U}_2 \mathbf{U}_2^\top (\mathbf{U}_2 \mathbf{U}_2^\top + c^2 \frac{\beta_2}{\beta_1} \mathbf{I})^{-1} \mathbf{V}_1 = \mathbf{0}$$

$$\Rightarrow \mathbf{U}_1^\top \mathbf{U}_1 \mathbf{V}_1 + c^2 \beta_2 (\mathbf{U}_2 \mathbf{U}_2^\top + c^2 \frac{\beta_2}{\beta_1} \mathbf{I})^{-1} \mathbf{V}_1 = \mathbf{U}_1^\top \mathbf{Z} \tag{69}$$

$$\Rightarrow c^2 \beta_2 (\mathbf{U}_2 \mathbf{U}_2^\top + c^2 \frac{\beta_2}{\beta_1} \mathbf{I})^{-1} \mathbf{V}_1 \mathbf{V}_1^\top = \mathbf{U}_1^\top \mathbf{Z} \mathbf{V}_1^\top - \mathbf{U}_1^\top \mathbf{U}_1 \mathbf{V}_1 \mathbf{V}_1^\top = \mathbf{U}_1^\top (\mathbf{Z} - \mathbf{U}_1 \mathbf{V}_1) \mathbf{V}_1^\top = \mathbf{U}_1^\top \mathbf{U}_1 \mathbf{\Sigma}_1, \tag{70}$$

where the last equality is from Eqn. (63).

Now, go back to the loss function $\mathcal{L}_{HVAE}$, we have:

$$\|\mathbf{U}_1 \mathbf{V}_1 - \mathbf{Z}\|_F^2 + \beta_1 \|(\mathbf{U}_2 \mathbf{W}_2 - \mathbf{I}) \mathbf{V}_1\|_F^2 + c^2 \beta_2 \|\mathbf{W}_2 \mathbf{V}_1\|_F^2$$

$$= \text{trace}(\mathbf{U}_1 \mathbf{V}_1 \mathbf{V}_1^\top \mathbf{U}_1^\top - 2\mathbf{Z} \mathbf{V}_1^\top \mathbf{U}_1^\top) + \|\mathbf{Z}\|_F^2 + \beta_1 \|(\mathbf{U}_2 \mathbf{W}_2 - \mathbf{I}) \mathbf{V}_1\|_F^2$$

$$+ c^2 \beta_2 \|\mathbf{U}_2^\top (\mathbf{U}_2 \mathbf{U}_2^\top + c^2 \mathbf{I})^{-1} \mathbf{V}_1\|_F^2$$

$$= \text{trace}(\mathbf{U}_1 \mathbf{V}_1 \mathbf{V}_1^\top \mathbf{U}_1^\top - 2\mathbf{Z} \mathbf{V}_1^\top \mathbf{U}_1^\top) + \|\mathbf{Z}\|_F^2$$

$$+ c^2 \beta_2 \, \text{trace}\left(\mathbf{V}_1^\top (\mathbf{U}_2 \mathbf{U}_2^\top + c^2 \frac{\beta_2}{\beta_1} \mathbf{I})^{-1} (\mathbf{U}_2 \mathbf{U}_2^\top + c^2 \frac{\beta_2}{\beta_1} \mathbf{I})(\mathbf{U}_2 \mathbf{U}_2^\top + c^2 \frac{\beta_2}{\beta_1} \mathbf{I})^{-1} \mathbf{V}_1\right)$$

$$= \text{trace}(\mathbf{U}_1^\top \mathbf{U}_1 \mathbf{V}_1 \mathbf{V}_1^\top - 2\mathbf{U}_1^\top \mathbf{Z} \mathbf{V}_1^\top) + \|\mathbf{Z}\|_F^2 + c^2 \beta_2 \, \text{trace}\left(\mathbf{V}_1^\top (\mathbf{U}_2 \mathbf{U}_2^\top + c^2 \frac{\beta_2}{\beta_1} \mathbf{I})^{-1} \mathbf{V}_1\right)$$

$$= -\text{trace}(\mathbf{U}_1^\top \mathbf{U}_1 \mathbf{\Sigma}_1) - \text{trace}(\mathbf{U}_1^\top \mathbf{Z} \mathbf{V}_1^\top) + \|\mathbf{Z}\|_F^2 + c^2 \beta_2 \, \text{trace}\left(\mathbf{V}_1^\top (\mathbf{U}_2 \mathbf{U}_2^\top + c^2 \frac{\beta_2}{\beta_1} \mathbf{I})^{-1} \mathbf{V}_1\right), \tag{71}$$

where we use Eqn. (68) in the last equation.

We also have:

$$\beta_1 \, \text{trace}((\mathbf{U}_2 \mathbf{W}_2 - \mathbf{I}) \mathbf{\Sigma}_1 (\mathbf{U}_2 \mathbf{W}_2 - \mathbf{I})^\top) + c^2 \beta_2 \, \text{trace}(\mathbf{W}_2 \mathbf{\Sigma}_1 \mathbf{W}_2^\top)$$

$$= \frac{c^4 \beta_2^2}{\beta_1} \, \text{trace}((\mathbf{U}_2 \mathbf{U}_2^\top + c^2 \frac{\beta_2}{\beta_1} \mathbf{I})^{-1} \mathbf{\Sigma}_1 (\mathbf{U}_2 \mathbf{U}_2^\top + c^2 \frac{\beta_2}{\beta_1} \mathbf{I})^{-1})$$

$$+ c^2 \beta_2 \, \text{trace}(\mathbf{U}_2^\top (\mathbf{U}_2 \mathbf{U}_2^\top + c^2 \frac{\beta_2}{\beta_1} \mathbf{I})^{-1} \mathbf{\Sigma}_1 (\mathbf{U}_2 \mathbf{U}_2^\top + c^2 \frac{\beta_2}{\beta_1} \mathbf{I})^{-1} \mathbf{U}_2)$$

$$= c^2 \beta_2 \, \text{trace}(\mathbf{\Sigma}_1 (\mathbf{U}_2 \mathbf{U}_2^\top + c^2 \frac{\beta_2}{\beta_1} \mathbf{I})^{-1}). \tag{72}$$

Thus, using Eqn. (71) and (72) to the loss function $\mathcal{L}_{HVAE}$ yields:

$$
\begin{aligned}
\mathcal{L}_{HVAE} &= \frac{1}{\eta_{\text{dec}}^2}\big[\|\mathbf{Z}\|_F^2 - \operatorname{trace}(\mathbf{U}_1^\top \mathbf{Z}\mathbf{V}_1^\top) + c^2\beta_2 \operatorname{trace}\big(\mathbf{V}_1^\top(\mathbf{U}_2\mathbf{U}_2^\top + c^2\tfrac{\beta_2}{\beta_1}\mathbf{I})^{-1}\mathbf{V}_1\big) \\
&\quad + \beta_1 \operatorname{trace}(\mathbf{U}_2\boldsymbol{\Sigma}_2\mathbf{U}_2^\top) + c^2\beta_2 \operatorname{trace}\big(\boldsymbol{\Sigma}_1(\mathbf{U}_2\mathbf{U}_2^\top + \tfrac{\eta_{\text{dec}}^2}{\eta_{\text{enc}}^2}\mathbf{I})^{-1}\big)\big] \\
&= \frac{1}{\eta_{\text{dec}}^2}\big[\|\mathbf{Z}\|_F^2 - \operatorname{trace}(\mathbf{U}_1^\top \mathbf{Z}\mathbf{V}_1^\top) + \beta_1 \operatorname{trace}(\mathbf{U}_2\boldsymbol{\Sigma}_2\mathbf{U}_2^\top) \\
&\quad + c^2\beta_2 \operatorname{trace}\big((\mathbf{U}_2\mathbf{U}_2^\top + c^2\tfrac{\beta_2}{\beta_1}\mathbf{I})^{-1}(\mathbf{V}_1\mathbf{V}_1^\top + \boldsymbol{\Sigma}_1))\big], \quad (73)
\end{aligned}
$$

We have the assumption that $\boldsymbol{\Sigma}_1 = \sigma_1^2\mathbf{I}$. From Eqn. (70), its LHS is symmetric because the RHS is symmetric, hence two symmetric matrices $\mathbf{V}_1\mathbf{V}_1^\top$ and $(\mathbf{U}_2\mathbf{U}_2^\top + c^2\tfrac{\beta_2}{\beta_1}\mathbf{I})^{-1}$ commute. Hence, they are orthogonally simultaneous diagonalizable. As a consequence, there exists an orthonormal matrix $\mathbf{P} \in \mathbb{R}^{d_1 \times d_1}$ such that $\mathbf{P}^\top\mathbf{V}_1\mathbf{V}_1^\top\mathbf{P}$ and $\mathbf{P}^\top(\mathbf{U}_2\mathbf{U}_2^\top + c^2\tfrac{\beta_2}{\beta_1}\mathbf{I})^{-1}\mathbf{P} = (\mathbf{P}^\top\mathbf{U}_2\mathbf{U}_2^\top\mathbf{P} + c^2\tfrac{\beta_2}{\beta_1}\mathbf{I})^{-1}$ are diagonal matrices (thus, $\mathbf{P}^\top\mathbf{U}_2\mathbf{U}_2^\top\mathbf{P}$ is also diagonal). Note that we choose $\mathbf{P}$ such that the eigenvalues of $\mathbf{V}_1\mathbf{V}_1^\top$ are in decreasing order, by altering the columns of $\mathbf{P}$. Therefore, we can write the SVD of $\mathbf{V}_1$ and $\mathbf{U}_2$ as $\mathbf{V}_1 = \mathbf{P}\Lambda\mathbf{Q}^\top$ and $\mathbf{U}_2 = \mathbf{P}\Omega\mathbf{N}^\top$ with orthonormal matrices $\mathbf{P} \in \mathbb{R}^{d_1 \times d_1}, \mathbf{Q} \in \mathbb{R}^{d_0 \times d_0}, \mathbf{N} \in \mathbb{R}^{d_2 \times d_2}$ and singular matrices $\Lambda \in \mathbb{R}^{d_1 \times d_0}, \Omega \in \mathbb{R}^{d_1 \times d_2}$. Specifically, the singular values of $\mathbf{V}_1, \mathbf{U}_2, \mathbf{Z}$ are $\{\lambda_i\}_{i=1}^{\min(d_0,d_1)}$, $\{\omega_i\}_{i=1}^{\min(d_1,d_2)}$ and $\{\theta_i\}_{i=1}^{\min(d_0,d_3)}$, respectively.

Next, from Eqn. (64), we have:

$$
\begin{aligned}
\sigma_2^2\mathbf{U}_2^\top\mathbf{U}_2 &= -\mathbf{U}_2^\top(\mathbf{U}_2\mathbf{W}_2 - \mathbf{I})(\mathbf{V}_1\mathbf{V}_1^\top + \sigma_1^2\mathbf{I})\mathbf{W}_2^\top \\
&= c^2\tfrac{\beta_2}{\beta_1}\mathbf{U}_2^\top(\mathbf{U}_2\mathbf{U}_2^\top + c^2\tfrac{\beta_2}{\beta_1}\mathbf{I})^{-1}(\mathbf{V}_1\mathbf{V}_1^\top + \sigma_1^2\mathbf{I})(\mathbf{U}_2\mathbf{U}_2^\top + c^2\tfrac{\beta_2}{\beta_1}\mathbf{I})^{-\top}\mathbf{U}_2, \quad (74)
\end{aligned}
$$

where we use Eqn. (67) and Eqn. (66) in the last equality. Using the SVD forms of $\mathbf{V}_1$ and $\mathbf{U}_2$ in Eqn. (74) yields:

$$
\sigma_2^2\mathbf{N}\Omega^\top\Omega\mathbf{N}^\top = c^2\tfrac{\beta_2}{\beta_1}\mathbf{N}\Omega^\top(\Omega\Omega^\top + c^2\tfrac{\beta_2}{\beta_1}\mathbf{I})^{-1}(\Lambda\Lambda^\top + \sigma_1^2\mathbf{I})(\Omega\Omega^\top + c^2\tfrac{\beta_2}{\beta_1}\mathbf{I})^{-1}\Omega\mathbf{N}^\top
$$

$$
\Rightarrow \sigma_2^2\omega_i^2 = c^2\frac{\beta_2}{\beta_1}\frac{\omega_i^2(\lambda_i^2 + \sigma_1^2)}{(\omega_i^2 + c^2\frac{\beta_2}{\beta_1})^2}, \quad \forall i \in [\min(d_1, d_2)].
$$

$$
\Rightarrow \omega_i = 0 \text{ or } c^2\frac{\beta_2}{\beta_1}(\lambda_i^2 + \sigma_1^2) = \sigma_2^2(\omega_i^2 + c^2\tfrac{\beta_2}{\beta_1})^2, \quad \forall i \in [\min(d_1, d_2)]. \quad (75)
$$

Computing the components in the loss function in Eqn. (73), we have:

$$
\begin{aligned}
\operatorname{trace}(\mathbf{U}_2\boldsymbol{\Sigma}_2\mathbf{U}_2^\top) &= \operatorname{trace}(\mathbf{U}_2^\top\mathbf{U}_2\boldsymbol{\Sigma}_2) \\
&= \operatorname{trace}(\mathbf{N}\Omega^\top\Omega\mathbf{N}^\top\boldsymbol{\Sigma}_2) = \sigma_2^2\operatorname{trace}(\Omega^\top\Omega) = \sigma_2^2\sum_{i=1}^{d_2}\omega_i^2, \\
\operatorname{trace}(\mathbf{U}_1^\top\mathbf{Z}\mathbf{V}_1^\top) &= \operatorname{trace}(\mathbf{V}_1^\top\mathbf{U}_1^\top\mathbf{Z}) \\
&= \operatorname{trace}(\mathbf{V}_1^\top(\mathbf{V}_1\mathbf{V}_1^\top + \sigma_1^2\mathbf{I})^{-1}\mathbf{V}_1\mathbf{Z}^\top\mathbf{Z}) \\
&\leq \sum_{i=1}^{d_0}\frac{\lambda_i^2\theta_i^2}{\lambda_i^2 + \sigma_1^2}, \\
\operatorname{trace}\big((\mathbf{U}_2\mathbf{U}_2^\top + c^2\tfrac{\beta_2}{\beta_1}\mathbf{I})^{-1}(\mathbf{V}_1\mathbf{V}_1^\top + \sigma_1^2\mathbf{I})\big) &= \sum_{i=1}^{d_1}\frac{\lambda_i^2 + \sigma_1^2}{\omega_i^2 + c^2\frac{\beta_2}{\beta_1}},
\end{aligned}
$$

where we denote $\{\theta_i\}_{i=1}^{d_0}$ are the singular values of $\mathbf{Z}$ and we use Von Neumann inequality for $\mathbf{Z}^\top\mathbf{Z}$ and $\mathbf{V}_1^\top(\mathbf{V}_1\mathbf{V}_1^\top + \sigma_1^2\mathbf{I})^{-1}\mathbf{V}_1$. The equality condition holds if these two symmetric matrices are simultaneous ordering diagonalizable.

We assume that $d_1 = d_2 \leq d_0$. From the loss function in Eqn. (73) and above calculation, we have:

$$
\begin{aligned}
\eta_{\text{dec}}^2 \mathcal{L}_{HVAE} &\geq \sum_{i=1}^{d_0} \theta_i^2 - \sum_{i=1}^{d_0} \frac{\lambda_i^2 \theta_i^2}{\lambda_i^2 + \sigma_1^2} + \beta_1 \sigma_2^2 \sum_{i=1}^{d_2} \omega_i^2 + \sum_{i=1}^{d_1} c^2 \beta_2 \frac{\lambda_i^2 + \sigma_1^2}{\omega_i^2 + c^2 \frac{\beta_2}{\beta_1}} \\
&= \sum_{i=1}^{d_0} \theta_i^2 - \sum_{i=1}^{d_1} \frac{\lambda_i^2 \theta_i^2}{\lambda_i^2 + \sigma_1^2} + \beta_1 \sigma_2^2 \sum_{i=1}^{d_1} \omega_i^2 + \sum_{i=1}^{d_1} c^2 \beta_2 \frac{\lambda_i^2 + \sigma_1^2}{\omega_i^2 + c^2 \frac{\beta_2}{\beta_1}} \\
&= \sum_{i=d_1}^{d_0} \theta_i^2 + \sum_{i=1}^{d_1} \underbrace{\left( \frac{\sigma_1^2 \theta_i^2}{\lambda_i^2 + \sigma_1^2} + \frac{c^2 \beta_2 (\lambda_i^2 + \sigma_1^2)}{\omega_i^2 + c^2 \frac{\beta_2}{\beta_1}} + \beta_1 \sigma_2^2 \omega_i^2 \right)}_{g(\lambda_i, \omega_i)},
\end{aligned}
\tag{76}
$$

where we denote $c := \eta_{\text{dec}} / \eta_{\text{enc}}$. We consider two cases derived from Eqn. (75):

- If $\omega = 0$:

$$
g(\lambda, 0) = \frac{\sigma_1^2 \theta^2}{\lambda^2 + \sigma_1^2} + \beta_1 (\lambda^2 + \sigma_1^2).
\tag{77}
$$

  We can easily see that if $\theta \geq \sqrt{\beta_1} \sigma_1$, we have $g$ is minimized at $\lambda^* = \sqrt{\frac{\sigma_1}{\sqrt{\beta_1}} (\theta - \sqrt{\beta_1} \sigma_1)}$ with $g^* = 2\sqrt{\beta_1} \sigma_1 \theta := g_1$. Otherwise, $\lambda^* = 0$ and $g^* = \theta^2 + \beta_1 \sigma_1^2 := g_2$.

- If $\omega^2 + c^2 \frac{\beta_2}{\beta_1} = \frac{c}{\sigma_2} \sqrt{\frac{\beta_2}{\beta_1}} \sqrt{\lambda^2 + \sigma_1^2}$. Let $t = \sqrt{\lambda^2 + \sigma_1^2}$ ($t \geq \sigma_1$), we have that $\omega^2 = \frac{c}{\sigma_2} \sqrt{\frac{\beta_2}{\beta_1}} t - c^2 \frac{\beta_2}{\beta_1} \geq 0$, hence $t \geq c\sigma_2 \sqrt{\frac{\beta_2}{\beta_1}}$. We have:

$$
\begin{aligned}
g(\lambda, \omega) &= \frac{\sigma_1^2 \theta^2}{\lambda^2 + \sigma_1^2} + 2c\sigma_2 \sqrt{\beta_1 \beta_2} \sqrt{\lambda^2 + \sigma_1^2} - \beta_2 c^2 \sigma_2^2 \\
&= \frac{\sigma_1^2 \theta^2}{t^2} + 2c\sigma_2 \sqrt{\beta_1 \beta_2} t - \beta_2 c^2 \sigma_2^2 := h(t, \theta), \quad t \geq \max(\sigma_1, c\sigma_2)
\end{aligned}
\tag{78}
$$

Taking the derivative of $h$ w.r.t $t$ yields:

$$
\frac{\partial h}{\partial t} = -\frac{2\sigma_1^2 \theta^2}{t^3} + 2c\sigma_2 \sqrt{\beta_1 \beta_2},
\tag{79}
$$

$$
\frac{\partial h}{\partial t} = 0
$$

$$
\Rightarrow t = \sqrt[3]{\frac{\sigma_1^2 \theta^2}{c\sigma_2 \sqrt{\beta_1 \beta_2}}} := t_0.
\tag{80}
$$

We can see that $t_0$ is the minimum of $h(t, \theta)$ if $t_0 \geq \max(\sigma_1, c\sigma_2 \sqrt{\frac{\beta_2}{\beta_1}})$. Otherwise, we will prove that the minimum of $h(t, \theta)$ is achieved at $t^* = \max(\sigma_1, c\sigma_2 \sqrt{\frac{\beta_2}{\beta_1}})$. We consider three following cases about $t_0$:

- If $t_0 \geq \max(\sigma_1, c\sigma_2 \sqrt{\frac{\beta_2}{\beta_1}}) \Leftrightarrow \begin{cases} \theta \geq \sqrt{c\sigma_1 \sigma_2 \sqrt{\beta_1 \beta_2}} \\ \theta \geq \frac{c^2 \sigma_2^2}{\sigma_1} \frac{\beta_2}{\sqrt{\beta_1}} \end{cases}$.

  The minimum of $g$ is achieved at $t^* = t_0$ with corresponding function value $g_3 = 3(\sqrt[3]{c\sigma_1 \sigma_2 \theta \sqrt{\beta_1 \beta_2}})^2 - c^2 \sigma_2^2 \beta_2$. We compare $g_3$ with $g_1$ and $g_2$ from the case $\omega = 0$:

$$
\begin{aligned}
g_1 - g_3 &= 2\sqrt{\beta_1} \sigma_1 \theta + \beta_2 c^2 \sigma_2^2 - 3\sqrt[3]{\beta_1 \beta_2 c^2 \sigma_1^2 \sigma_2^2 \theta^2} \\
&= \sqrt{\beta_1} \sigma_1 \theta + \sqrt{\beta_1} \sigma_1 \theta + \beta_2 c^2 \sigma_2^2 - 3\sqrt[3]{\beta_1 \beta_2 c^2 \sigma_1^2 \sigma_2^2 \theta^2} \geq 0. \quad \text{(Cauchy-Schwarz inequality)}
\end{aligned}
$$

$$
g_2 - g_3 = \theta_i^2 + \beta_1 \sigma_1^2 + \beta_2 c^2 \sigma_2^2 - 3\sqrt[3]{\beta_1 \beta_2 c^2 \sigma_1^2 \sigma_2^2 \theta^2} \geq 0. \quad \text{(Cauchy-Schwarz inequality)}.
$$

Thus, $g^* = g_3$ and:

$$\lambda^* = \sqrt[3]{\frac{\sigma_1^2}{\sqrt{\beta_1\beta_2}c\sigma_2}}\sqrt{\sqrt[3]{\theta^4} - \sqrt[3]{\beta_1\beta_2c^2\sigma_1^2\sigma_2^2}},$$

$$\omega^* = \sqrt[3]{\frac{\sqrt{\beta_2}c\sigma_1}{\beta_1\sigma_2^2}}\sqrt{\sqrt[3]{\theta^2} - \sqrt[3]{\frac{\beta_2^2c^4\sigma_2^4}{\beta_1\sigma_1^2}}}. \tag{81}$$

- If $\sigma_1 \geq \sqrt{\frac{\beta_2}{\beta_1}}c\sigma_2 > t_0 \Leftrightarrow \theta_i < \frac{\beta_2}{\sqrt{\beta_1}}\frac{c^2\sigma_2^2}{\sigma_1} \leq \sqrt{\sqrt{\beta_1\beta_2}c\sigma_1\sigma_2}$. Thus, $\sqrt{\beta_1}\sigma_1 \geq \sqrt{\beta_2}c\sigma_2$.

  From the condition $t \geq \max\{\sigma_1, \sqrt{\frac{\beta_2}{\beta_1}}c\sigma_2\}$, we obtain $t \geq \sigma_1 > t_0$. For all $t > t_0$, we have:

  $$\frac{\partial h(t)}{\partial t} > \frac{\partial h(t_0)}{\partial t} = 0. \quad (\text{since } \frac{\partial h}{\partial t} \text{ is an increasing function w.r.t } t)$$

  So the minimum of $g$ is achieved at $t^* = \sigma_1$ with corresponding function value:

  $$g_4 = \theta^2 + 2c\sigma_1\sigma_2\sqrt{\beta_1\beta_2} - \beta_2c^2\sigma_2^2. \tag{82}$$

  We only need to compare $g_4$ with $g_2$ since $\theta_i < \sqrt{\sqrt{\beta_1\beta_2}c\sigma_1\sigma_2} \leq \sqrt{(\sqrt{\beta_1}\sigma_1)^2} = \sqrt{\beta_1}\sigma_1$. We have:

  $$g_2 - g_4 = \beta_1\sigma_1^2 + \beta_2c^2\sigma_2^2 - 2c\sigma_1\sigma_2\sqrt{\beta_1\beta_2} \geq 0. \quad (\text{Cauchy-Schwarz inequality})$$

  Thus, $g^* = g_4$ and:

  $$\lambda^* = 0,$$

  $$\omega^* = \sqrt{\sqrt{\frac{\beta_2}{\beta_1}}\frac{c\sigma_1}{\sigma_2} - \frac{\beta_2}{\beta_1}c^2}.$$

- If $\sigma_1 > t_0 \geq \sqrt{\frac{\beta_2}{\beta_1}}c\sigma_2 \Leftrightarrow \frac{\beta_2}{\sqrt{\beta_1}}\frac{c^2\sigma_2^2}{\sigma_1} \leq \theta_i < \sqrt{\sqrt{\beta_1\beta_2}c\sigma_1\sigma_2}$. Thus, $\sqrt{\beta_1}\sigma_1 \geq \sqrt{\beta_2}c\sigma_2$.

  Similar to the previous case, the minimum of $g$ is achieved at $t^* = \sigma_1$ with corresponding function value:

  $$g^* = g_4 = \theta^2 + 2c\sigma_1\sigma_2\sqrt{\beta_1\beta_2} - \beta_2c^2\sigma_2^2, \tag{83}$$

  and minimizers:

  $$\lambda^* = 0,$$

  $$\omega^* = \sqrt{\sqrt{\frac{\beta_2}{\beta_1}}\frac{c\sigma_1}{\sigma_2} - \frac{\beta_2}{\beta_1}c^2}.$$

- If $\sqrt{\frac{\beta_2}{\beta_1}}c\sigma_2 > \max\{t_0, \sigma_1\}$, $t^* = \sqrt{\frac{\beta_2}{\beta_1}}c\sigma_2$ and thus, $\omega = 0$. We already know that when $\omega^* = 0$, $\lambda^* = 0$ when $\theta < \sqrt{\beta_1}\sigma_1$ and $\lambda^* = \sqrt{\frac{\sigma_1\theta}{\sqrt{\beta_1}} - \sigma_1^2}$ when $\theta \geq \sqrt{\beta_1}\sigma_1$.

In conclusion:

- If $\theta \geq \max\{\sqrt{\sqrt{\beta_1\beta_2}c\sigma_1\sigma_2}, \frac{\beta_2}{\sqrt{\beta_1}}\frac{c^2\sigma_2^2}{\sigma_1}\}$:

  $$\lambda^* = \sqrt[3]{\frac{\sigma_1^2}{\sqrt{\beta_1\beta_2}c\sigma_2}}\sqrt{\sqrt[3]{\theta^4} - \sqrt[3]{\beta_1\beta_2c^2\sigma_1^2\sigma_2^2}},$$

  $$\omega^* = \sqrt[3]{\frac{\sqrt{\beta_2}c\sigma_1}{\beta_1\sigma_2^2}}\sqrt{\sqrt[3]{\theta^2} - \sqrt[3]{\frac{\beta_2^2c^4\sigma_2^4}{\beta_1\sigma_1^2}}}.$$

- If $\theta < \max\{\sqrt{\sqrt{\beta_1\beta_2}c\sigma_1\sigma_2}, \frac{\beta_2}{\sqrt{\beta_1}}\frac{c^2\sigma_2^2}{\sigma_1}\}$ and $\sqrt{\beta_1}\sigma_1 \geq \sqrt{\beta_2}c\sigma_2$:

$$\lambda^* = 0,$$

$$\omega^* = \sqrt{\sqrt{\frac{\beta_2}{\beta_1}}\frac{c\sigma_1}{\sigma_2} - \frac{\beta_2}{\beta_1}c^2}.$$

- If $\theta < \max\{\sqrt{\sqrt{\beta_1\beta_2}c\sigma_1\sigma_2}, \frac{\beta_2}{\sqrt{\beta_1}}\frac{c^2\sigma_2^2}{\sigma_1}\}$ and $\theta < \sqrt{\beta_1}\sigma_1 < \sqrt{\beta_2}c\sigma_2$:

$$\lambda^* = \omega^* = 0$$

- If $\theta < \max\{\sqrt{\sqrt{\beta_1\beta_2}c\sigma_1\sigma_2}, \frac{\beta_2}{\sqrt{\beta_1}}\frac{c^2\sigma_2^2}{\sigma_1}\}$ and $\sqrt{\beta_1}\sigma_1 \leq \theta$ and $\sqrt{\beta_1}\sigma_1 < \sqrt{\beta_2}c\sigma_2$:

$$\omega^* = 0,$$

$$\lambda^* = \sqrt{\frac{\sigma_1}{\sqrt{\beta_1}}(\theta - \sqrt{\beta_1}\sigma_1)}.$$

$\square$

**Remark 5.** *The singular values of $\mathbf{W}_2$ and $\mathbf{U}_1$ can be calculated via Eqn. (66) and Eqn. (68). From Eqn. (66), we have that:*

$$\mathbf{W}_2 = \mathbf{U}_2^\top(\mathbf{U}_2\mathbf{U}_2^\top + c^2\mathbf{I})^{-1} = \mathbf{N}\Omega(\Omega\Omega^\top + c^2\mathbf{I})^{-1}\mathbf{P}^\top. \tag{84}$$

*Since $\mathbf{N}$ can be arbitrary orthonormal matrix, the number of zero rows of $\mathbf{W}_2$ can vary from $0$ (no posterior collapse) to $d_2 - \text{rank}(\mathbf{W}_2)$. Similar argument can be made for $\mathbf{V}_1$. Indeed, we have $bv_1$ has the form $\mathbf{V}_1 = \mathbf{P}\Lambda\mathbf{R}^\top$. Thus, $\mathbf{P}$ will determine the number of zero rows of $\mathbf{V}_1$. The number of zero rows of $\mathbf{V}_1$ will vary from $0$ to $d_1 - \text{rank}(\mathbf{V}_1)$ since $\mathbf{P}$ can be chosen arbitrarily.*

