# OpenReview forum: "Beyond Vanilla Variational Autoencoders: Detecting Posterior Collapse in Conditional and Hierarchical Variational Autoencoders"
_ICLR.cc/2024/Conference — ICLR 2024 poster_

### Official Review · Reviewer_1CHE · 2023-10-30

**Soundness:** 3 good
**Presentation:** 3 good
**Contribution:** 3 good
**Rating:** 6
**Confidence:** 4

**Summary:**

In this paper, the authors analyze the issue of posterior mode collapse for conditional and hierarchical linear VAEs. They demonstrate conditions required to observe posterior collapse, which provides meaningful insight into the design and training of more complex VAEs.

**Strengths:**

- Technical derivations seem correct, and the paper methodology is solid.
- Theoretical insights into the understanding and design of VAEs.

**Weaknesses:**

The reviewer appreciates little contribution in the paper. Yes, the authors generalize the results obtained for the linear VAE by Wang & Ziying (2022) and Dai et al. 2020 to linear CVAEs and hierarchical linear VAES, and this contribution is acknowledged. However, I wonder if generalizing the analysis to these new models resulted in different conclusions or insights about avoiding posterior collapse. In other words, is anything in Table 1 significantly different from what we learned from Wang and Ziying? At the end, the conclusion is that both $\beta$ and $\eta_{dec}$ should be small ...

**Questions:**

Please clarify what new designing insights are obtained from your results that couldn't be guessed from other works in the literature on analytical understanding methods for VAEs.

---

> ### Author Response · Authors · 2023-11-18
> **Response to Reviewer 1CHE - Part 1**
>
> Dear Reviewer 1CHE,
>
> Thank you for your thoughtful review and valuable feedback. Below we address your concerns. We have also revised the current manuscript according to the suggestions of the reviewers. The changes are in blue color. Please kindly see the revised manuscript, as well as our General Response and Summary of Revision.
>
> **Q1**: The reviewer appreciates little contribution in the paper. Yes, the authors generalize the results obtained for the linear VAE by Wang & Ziyin (2022) and Dai et al. 2020 to linear CVAEs and hierarchical linear VAES, and this contribution is acknowledged.  However, I wonder if generalizing the analysis to these new models resulted in different conclusions or insights about avoiding posterior collapse. In other words, is anything in Table 1 significantly different from what we learned from Wang and Ziyin? At the end, the conclusion is that both $\beta$  and $\eta_{dec}$  should be small.
>
> **A1**: We respectfully disagree with the reviewer that our contributions are little. Please allow us to clarify our contributions below.
>
> 1/ **Implications of our theoretical results for CVAE and MHVAE.**
>
> For CVAE, our theoretical analysis is the first to prove the existence of posterior collapse in CVAE, an important and popularly used variant of VAE [1, 2, 3]. We rigorously study the effect of parameters to the rank of the models and the level of posterior collapse at optimality, including $\theta$’s (the singular values of matrix $Z = \mathbb{E}(x \tilde{x}^{\top})$, defined at Theorem 2), $\beta$ (scaling scalar of the KL term of ELBO at Equation (5) in our paper) and $\eta_{dec}$ (magnitude of the variance of the generating distribution). The results imply that a sufficiently small $\beta$ and/or $\eta_{dec}$ can mitigate posterior collapse. Especially, we prove that **the correlation of the input condition and output of training data relates to the collapse level** (see the paragraph after Theorem 2 in our manuscript). This insight obviously cannot be derived from the standard VAE setting which does not incorporate input conditions into the generative process.
>
> For MHVAE, our theoretical analysis is again the first to prove the existence of posterior collapse in MHVAE, another important and popularly used variant of VAE [4, 5, 6, 7]. We also rigorously study the effect of parameters to the rank of the models and the level of posterior collapse at optimality, including $\theta$’s, $\beta_1$, $\beta_2$ and $\eta_{dec}$. Moreover, since the ELBO loss of MHVAE consists of multiple KL-regularized terms, we study the effect of each KL-regularized term and the trade-offs between latents via the magnitude of the hyperparameters $\beta_1$ and $\beta_2$. We find that **decreasing $\beta_2$ can help to alleviate posterior collapse and increase the rank of the model, while decreasing $\beta_1$ will have the opposite effect**. This insight cannot be derived from the standard VAE setting which has only one latent and one KL-regularized term in the ELBO. In addition, we observe that **when any latent variable of MHVAE suffers complete collapse, its higher-level latents become useless**. Thus, we suggest creating additional mapping between the input data and every latent to prevent this adverse situation.
>
> Furthermore, we find that **unlearnable encoder variance can prevent posterior collapse** (see Section 3 in our paper) for VAE, CVAE and MHVAE. This insight is also novel.
>
> 2/ **As per Reviewer vZoP comment, our paper “provides an important step forward for the understanding of the inner workings of VAEs and provides the basis and theoretical tool set for future research”**. Specifically, current state-of-the-art VAE models are mostly CVAE and HVAE [1, 2, 3, 4, 5, 6, 7], not standard VAE with only one latent, so our settings are more practical compared with prior works and provide the basis and theoretical tool set for future research of CVAE and HVAE. Furthermore, as the discussion in Section 4.2 in our manuscript, diffusion models share many similarities with deep MHVAE model where the encoding process of diffusion models also consists of consecutive linear maps with injected noise and with unlearnable isotropic variance, and their training loss function is also ELBO. Therefore, our framework potentially sheds light on the understanding of diffusion models.

---

> > ### Author Response · Authors · 2023-11-18
> > **Response to Reviewer 1CHE - Part 2**
> >
> > **A1 (Continue)**:
> >
> > 3/ **Our proofs are methodologically novel and not just a generalization from linear VAE**.
> >
> > For example, for MHVAE, both the architecture and the training problem are more complicated than the standard VAEs with only one latent variable. Adding one more level of latent requires two more mappings to map the first latent to the second latent in the encoding process and vice versa for the decoding process. The training problem of maximizing the ELBO must have an extra KL-regularizer term between the two latents, i.e., $D_{\text{KL}}(q(z_1 | x) || p(z_1 | z_{2}))$. This term significantly complicates the training problem and our proof uses new techniques to handle it. Specifically, we prove a property that $\mathbf{V}\_{1} \mathbf{V}\_{1}^{\top}$ and $\mathbf{U}\_{2} \mathbf{U}\_{2}^{\top}$ are simultaneously diagonalizable (see page 38 in our manuscript), and thus, we are able to convert the zero gradient condition into relations of their singular values $\lambda$'s and $\omega$’s (see Eqn. (86) at page 38). Thanks to these relations between $\lambda$'s and $\omega$’s, the loss function now can be converted to a function of singular values. Again, the formulations for MHVAE are significantly more cumbersome and require different calculations compared to linear VAE cases, which can be seen via the gradient calculation from Eqn. (75)-(78) and how we leverage these critical points conditions to simplify the loss function in Eqn. (79)-(88) in our manuscript.
> >
> > **Q2**: Please clarify what new designing insights are obtained from your results that couldn't be guessed from other works in the literature on analytical understanding methods for VAEs.
> >
> > **A2**: We believe the Table 1 in Section 4.3 of our manuscript summarizes and clarifies our new insights to prevent posterior collapse for CVAE and HVAE models. Although you may say that “$\beta$ and $\eta_{dec}$  should be small” is a similar insight drawn from standard VAE, in our opinion, it is **still a new insight since CVAE is a different variant of VAE with different architecture and mechanism**. Furthermore, **our results even suggest insights that cannot be drawn from the standard VAE setting**. Specifically, for CVAE, we prove that **the correlation of the input condition and output of training data relates to the collapse level** (see the paragraph after Theorem 2 in our manuscript). This insight obviously cannot be derived from the standard VAE setting which does not incorporate input conditions into the generative process. For MHVAE, we find that **decreasing $\beta_2$ can help to alleviate posterior collapse and increase the rank of the model, while decreasing $\beta_1$ will have the opposite effect**. This insight also cannot be derived from the standard VAE setting which has only one latent and one KL-regularized term in the ELBO. In addition, for both models, we find that **unlearnable encoder variance can prevent posterior collapse**, which is also a new insight in the literature.
> >
> > **References:**
> >
> > [1] Fang, Le, et al. "Transformer-based conditional variational autoencoder for controllable story generation." arXiv preprint arXiv:2101.00828
> >
> > [2] Sohn, Kihyuk, Honglak Lee, and Xinchen Yan. "Learning structured output representation using deep conditional generative models", NeurIPS 2015
> >
> > [3] Walker, Jacob, et al. "An uncertain future: Forecasting from static images using variational autoencoders.", ECCV 2016
> >
> > [4] Child, Rewon. "Very deep vaes generalize autoregressive models and can outperform them on images." arXiv preprint arXiv:2011.10650 (2020)
> >
> > [5] Vahdat, Arash, and Jan Kautz. "NVAE: A deep hierarchical variational autoencoder", NeurIPS 2020
> >
> > [6] Sønderby, Casper Kaae, et al. "Ladder variational autoencoders”, NeurIPS 2016
> >
> > [7] Maaløe, Lars, Marco Fraccaro, and Ole Winther. "Semi-supervised generation with cluster-aware generative models." arXiv preprint arXiv:1704.00637 (2017).

---

> > > ### Comment · Reviewer_1CHE · 2023-11-19
> > >
> > > Dear authors,
> > >
> > > Thanks for you detail response. Upon reading all the answers, I am willing to admit that I probably overlooked all the paper contributions, which are also better emphasized in the revised version of the paper.  In this regard, I am willing to increase my rating of the paper once the rebuttal period finishes.

---

> > > > ### Author Response · Authors · 2023-11-19
> > > > **Thanks for your endorsement!**
> > > >
> > > > Dear Reviewer 1CHE,
> > > >
> > > > Thanks for your response and we appreciate your endorsement. Please let us know if you need any further information/clarification about our work.
> > > >
> > > > Best Regards,
> > > >
> > > > The Authors

---

### Official Review · Reviewer_fUP2 · 2023-10-30

**Soundness:** 3 good
**Presentation:** 2 fair
**Contribution:** 3 good
**Rating:** 6
**Confidence:** 4

**Summary:**

The paper presents the first theoretical analyses of posterior collapse in the conditional VAE and markovian hierarchical VAE. Both models contrast with standard VAEs, which have theoretical results related to posterior collapse, in that they have more complex latent structure. Mathematical analyses derive conditions under which we should see posterior collapse for both classes of models. Qualitative observations are extracted from the theorems. Experimental results on MNIST test and validate the qualitative observations.

**Strengths:**

- The VAE is a widely used, classic generative modeling framework. Theoretical results providing a better understanding of performance, and lack thereof, are interesting and valuable.
- The theoretical results and qualitative observations provide satisfying and interesting insights into the behavior of the model.
- The empirical experiments provide a nice, if modest, evaluation of the predictions.

**Weaknesses:**

- The primary weakness was in exposition. The paper tends to discuss intuitions after theorems, which makes the theorems a bit challenging on first read.
- Moreover, at several point the authors assume a lot of the reader. Specifically, consider the case of unlearnable Sigma, and comparisons between theorem 1 and the previous work (the result from which is not stated in the text).
Taken together the paper was more difficult than need be to understand.

Minor comments:
"Interestingly, we find that the correlation of the training input and training output is one of the factors that decides the collapse level" Why is this interesting?
- "We study model with" typo
- "beside the eigenvalue" typo
- " i) we characterize the global solutions of linear VAE training problem for unlearnable Σ case, which generalizes the result in (Wang & Ziyin, 2022) where only the unlearnable isotropic Σ is considered, and ii) we prove that for the case of unlearnable Σ, even when the encoder matrix is low-rank, posterior collapse may not happen. Thus, learnable latent variance is among the causes of posterior collapse, opposite to the results in Wang & Ziyin (2022) that it is not the cause of posterior collapse." This warrants some explanation. Why is the conclusion opposite?
- What does it mean for Sigma to be unlearnable? This isn't properly explained.
- What is the meaning of Theorem 1? It isn't yet clear to me what we are proving or why?
- "We note that our results generalize Theorem 1 in (Wang & Ziyin, 2022) where σi’s are all equal to a constant." It would really be helpful to the reader to present the previous result and explain clearly why.
- Ok the explanation after the theorem is helpful. It would be preferable to help readers see what is coming ahead of time, too.
- I am still unsure about what an unlearnable Sigma is.

**Questions:**

My main questions are related to the limitations. I would like to hear how the authors would plan to revise to clarify and sharpen exposition.

---

> ### Author Response · Authors · 2023-11-18
> **Response to Reviewer fUP2 - Part 1**
>
> Dear Reviewer fUP2,
>
> Thank you for your thoughtful review and valuable feedback. Below we address your concerns. We have also revised the current manuscript according to the suggestions of the reviewers. The changes are in blue color. Please kindly see the revised manuscript, as well as our General Response and Summary of Revision.
>
> **Q1**: The primary weakness was in exposition. The paper tends to discuss intuitions after theorems, which makes the theorems a bit challenging on first read.
>
> **Q2**: Ok the explanation after the theorem is helpful. It would be preferable to help readers see what is coming ahead of time, too.
>
> **A1 + A2**: Thanks for your suggestions. We have added the description before every theorem of what the incoming theorems are about and what they try to accomplish to facilitate the reading.
>
> **Q3**: Moreover, at several points the authors assume a lot of the reader. Specifically, consider the case of unlearnable Sigma, and comparisons between theorem 1 and the previous work (the result from which is not stated in the text). Taken together the paper was more difficult than need be to understand.
>
> **Q4**: "We note that our results generalize Theorem 1 in (Wang & Ziyin, 2022) where σi’s are all equal to a constant." It would really be helpful to the reader to present the previous result and explain clearly why.
>
> **A3 + A4**: Thank you for your valuable comment. We have revised the paper to clarify the previous results whenever we mention them. For the comparison between our Theorem 1 and previous results, we have updated the paper to describe explicitly the differences between our Theorem 1 and Proposition 2 in (Wang & Ziyin, 2022), please see the paragraphs before and after Theorem 1 in our manuscript.
>
> Specifically, our Theorem 1 allows for arbitrary predefined values of $\\{ \sigma_i \\}\_{i=1}^{d_1}$, thus is more general than the Proposition 2 in (Wang & Ziyin, 2022) where $\sigma_i$'s are all equal to a constant. Under broader settings, there are two notable points from Theorem 1 that has not been captured in the previous result of (Wang & Ziyin, 2022): i) at optimality, the singular matrix $\mathbf{T}$ of the encoder map $\mathbf{V}$ sorts the set $\\{ \sigma_i \\}\_{i=1}^{d_1}$ in non-decreasing order, and ii) singular values $\omega\_{i}$ of the decoder map $\mathbf{U}$ and $\lambda\_{i}$ of the encoder map $\mathbf{V}$ are calculated via the $i$-th smallest value $\sigma^{\prime}\_{i}$ of the set $\\{ \sigma_i \\}\_{i=1}^{d_1}$, not necessarily the $i$-th element $\sigma_i$.
>
> **Q5**: "Interestingly, we find that the correlation of the training input and training output is one of the factors that decides the collapse level" Why is this interesting?
> And some typos: "We study model with", "beside the eigenvalue"
>
> **A5**: Thanks for your suggestion and for pointing out the typos. We used the word “Interestingly” because we found that the finding was novel. In our revision, we removed this word. We have also fixed the typos that the reviewer mentioned.
>
> **Q6**: What is the meaning of Theorem 1? It isn't yet clear to me what we are proving or why?
>
> **Q7**: “Thus, learnable latent variance is among the causes of posterior collapse, opposite to the results in Wang & Ziyin (2022) that it is not the cause of posterior collapse." This warrants some explanation. Why is the conclusion opposite?
>
> **A6 + A7**: Our Theorem 1 characterizes the global minima of the ELBO training problem, which includes the parameters from the matrix $\mathbf{U}$ (the decoder), $\mathbf{W}$ (the encoder) as optimization variables. The encoder variance $\mathbf{\Sigma}$ is fixed/unlearnable in this setting. We prove that the optimal parameters $(\mathbf{U}^{\ast}, \mathbf{W}^{\ast})$ have their singular values can be calculated via the closed-form formula stated in Theorem 1 in our manuscript. Thus, the ranks of the encoder map and the decoder map depend on $\theta$’s (the singular values of the matrix $\mathbf{Z} := \mathbb{E}(x \tilde{x}^{\top})$ ), $\beta$ (scaling factor of the KL term in ELBO in Eqn. (3)) and $\eta_{dec}$ (the variance magnitude of the generating distribution $p(x | z)$). After Theorem 1, in Section 3, we discuss the conditions for posterior collapse to occur, including the rank of the encoder/decoder and learnability of the encoder variance.

---

> > ### Author Response · Authors · 2023-11-18
> > **Response to Reviewer fUP2 - Part 2**
> >
> > **A6 + A7 (Continue)**:
> > Compared to previous results, specifically Proposition 2 in (Wang & Ziyin, 2022), our Theorem 1 characterizes the global solutions of standard linear VAE for unlearnable $\mathbf{\Sigma}$ with arbitrary elements $\sigma_i$ on the diagonal while Proposition 2 in (Wang & Ziyin, 2022) studies the isotropic $\mathbf{\Sigma} = \sigma^2 \mathbf{I} $ case only. **Another new and interesting result from Theorem 1 in our manuscript is that we prove the left singular matrix $\mathbf{T}$ of the encoder map $\mathbf{V}$ is the sorting matrix that sorts the values of $\mathbf{\Sigma}$ in non-decreasing order**. . This property of $\mathbf{T}$ is one of the factors that decide whether posterior collapse happens or not, which has not been considered in (Wang & Ziyin, 2022).
> >
> > Regarding the opposite conclusion, after Theorem 1 in (Wang & Ziyin, 2022), they claim that “optimizable encoder variance is not essential to posterior collapse problem” (Section 4), “a learnable (data-dependent or not) latent variance is not the cause of posterior collapse” (Section 4.5). Moreover, in Section 4.2 in [40], after Theorem 1, under the setting of unlearnable encoder variance $\sigma\_i = \eta\_{dec}, \forall i$, they claimed the existence of posterior collapse when the learned model is low-rank (i.e., some singular values $\lambda$ and $\theta$ are 0’s). In particular, they argue “... the learned model becomes low-rank. Namely, some of the dimensions collapse with the prior”. **We observe that these claims are not totally correct**, and we explain in our manuscript why posterior collapse may not happen in the above unlearnable setting (see the paragraph after our Theorem 1). For convenience, we repeat the explanation below.
> >
> > We first let the SVD of the encoder matrix $\mathbf{V}$ (the encoder) to be $\mathbf{V} =  \mathbf{T} \Lambda \mathbf{S}^{\top}$ then at the global minimizer of the training problem, we prove that $\mathbf{T}$ is the matrix that sorts the diagonal values of $\Sigma$ in non-decreasing order. When $\mathbf{V}$ is low-rank, $\Lambda$ will have zero rows, then $\Lambda \mathbf{S}^{\top}$ will have zero rows. However, in the case where the entries on the diagonal of encoder variance matrix $\Sigma$ are equal (as considered in Theorem 1 in Wang), $\mathbf{T}$ can be any orthonormal matrix (since
> > $\sigma\_i$’s are all equal and in non-decreasing order already), and thus, $\mathbf{V} = \mathbf{T} \Lambda \mathbf{S}^{\top}$ may have no zero rows. Then, for any data $x$, the mean $\mathbf{V} \tilde{x}$ of the approximate posterior $q(z|x) =  \mathcal{N} (\mathbf{V} \tilde{x}, \Sigma)$ might have no zero component, and thus **the posterior may not collapse to the prior $\mathcal{N}(0, \eta^2_{enc})$**.
> >
> > For the case of **learnable** $\mathbf{\Sigma}$, low-rank condition leads to posterior collapse. This is because at optimality, the diagonality of $\mathbf{\Sigma}$ makes $\mathbf{U}^{\top} \mathbf{U}$ diagonal. Therefore, $\mathbf{U}$ has zero columns, and subsequently,  $\mathbf{V}$ will have zero rows (see detailed explanation at the paragraph before Section 4 in our paper). Finally, the corresponding dimensions of $z = \mathbf{V} \tilde{x} + \epsilon$ collapse to its prior $\mathcal{N}(0, \eta^2_{\text{enc}})$. **Therefore, our findings suggest that the learnability of encoder variance plays an important role in the existence of posterior collapse**. We have updated the manuscript to describe carefully the differences between our Theorem 1 and Proposition 2 in (Wang & Ziyin, 2022) in the paragraphs after Theorem 1 in our revision.
> >
> > **Q8**: What does it mean for Sigma to be unlearnable? This isn't properly explained.
> >
> > **A8**: It means the elements of $\mathbf{\Sigma}$ are predefined and not updated during the training of the model. We have clarified this in the first paragraph of Section 3 in our revised manuscript.

---

> > > ### Comment · Reviewer_fUP2 · 2023-11-22
> > > **Thanks for the detailed response**
> > >
> > > Thank you for the detailed response!

---

### Official Review · Reviewer_VR5S · 2023-11-01

**Soundness:** 4 excellent
**Presentation:** 3 good
**Contribution:** 3 good
**Rating:** 6
**Confidence:** 3

**Summary:**

This paper presents a theoretical and empirical analysis of the problem of posterior collapse in two types of VAE, specifically conditional and hierarchical. Under a linear VAE setting they prove certain specific conditions as contributors to posterior collapse. They then conduct various experiments on MNIST, focusing in their empirical work on non-linear VAEs, and demonstrate that their theoretical results line up with the experimental findings of what settings lead to collapse.

**Strengths:**

- The paper puts forward theoretical analysis of posterior collapse in previously unstudied models that are more complex than vanilla VAEs
- The paper performs experiments that corroborate the theoretical claims, albeit in a more realistic nonlinear setting

**Weaknesses:**

- Much of the empirical results are relegated to the appendix with the paper itself being somewhat light on experiments. This isn’t necessarily a bad thing but it would be nice to see more thorough experiments in the main paper, e.g. vanilla VAE, linear CVAE/MHVAE, learnable vs. unlearnable variance, just for comparison.
- The theoretical results are generally restricted to highly specific scenarios, such as linear and two latent for MHVAE, and the setting of both variances being learnable in MHVAE is not considered. It would be nice to see more theoretical motivation for why these results might generalize to more realistic training settings, but instead that is largely directed towards prior work.
- Even in the empirical setting the majority of models studied are restricted to fairly shallow networks. The results are entirely on a simple dataset, MNIST, and therefore it’s hard to get a sense for how the experiments would behave in a more complex setting with deeper networks, modern architectures, and less standardized data.
- One of the most noticeable absences is that this paper doesn’t consider the effects of SGD and optimization strategies. The setting where global solutions are directly obtainable is not really representative of most real world VAEs which would involve deep nonlinear networks that are most likely to only reach a local optimum. While the experiments themselves do use Adam to train the models, there’s not much analysis of how the optimization strategy interacts with other phenomena.
- Many of these weaknesses could be levied against prior work in this space. For what it’s worth this paper does go beyond those works in that they do rigorous theoretical analysis of more complex model types.

**Questions:**

You claim that Wang & Ziyin 2022 arrived at the opposite conclusion as you regarding the role of learnable latent variance. Can you say more about why you would have found different results and what that means?

---

> ### Author Response · Authors · 2023-11-18
> **Response to Reviewer VR5S - Part 1**
>
> Dear Reviewer VR5S,
>
> Thank you for your thoughtful review and valuable feedback. Below we address your concerns. We have also revised the current manuscript according to the suggestions of the reviewers. The changes are in blue color. Please kindly see the revised manuscript, as well as our General Response and Summary of Revision.
>
> **Q1**:  Much of the empirical results are relegated to the appendix with the paper itself being somewhat light on experiments. This isn’t necessarily a bad thing but it would be nice to see more thorough experiments in the main paper, e.g. vanilla VAE, linear CVAE/MHVAE, learnable vs. unlearnable variance, just for comparison.
>
> **A1**: Thank you for your comment. We have moved an additional experiment to the main paper, where we compare the level of posterior collapse for VAE with learnable and unlearnable encoder variance to validate our claim about the role of learnability encoder variance to posterior collapse (Section 3). Please kindly see Figure 2 in the revised manuscript. Due to the main text’s space limit of 9 pages and our work consists of many theoretical results that require detailed explanation, we currently include other experimental results in the Appendix.
>
> **Q2**: The theoretical results are generally restricted to highly specific scenarios, such as linear and two latent for MHVAE, and the setting of both variances being learnable in MHVAE is not considered. It would be nice to see more theoretical motivation for why these results might generalize to more realistic training settings, but instead that is largely directed towards prior work.
>
> **A2**: We agree with the reviewer that our theoretical results focus on the settings of linear networks. However, the theoretical analysis of nonlinear networks is very challenging and, in fact, there has been no rigorous theory for deep nonlinear networks yet to the best of our knowledge. Even for the linear case, a complete theoretical understanding of the VAE setting is still lacking. We aim to fill this gap and use the insights from our work for deep linear VAEs to empirically examine and improve deep nonlinear VAEs: we have experiments at Figure 2 + Figure 3 in our main paper and more experiments on nonlinear networks in the Appendix that support our theoretical insights.
>
> We also want to emphasize that **analyzing deep linear networks has been an important step in studying deep nonlinear networks**. [1, 2, 3, 4] show that the optimization of deep linear models exhibits similar properties to those of the optimization of deep nonlinear models. In practice, deep linear networks can help improve the training and performance of deep nonlinear networks [5, 6, 7]. Specifically, [5] empirically proves that linear overparameterization in nonlinear networks improves generalization on classification tasks (see Section 4 in [5]). In particular, [5] expands each linear layer into a succession of multiple linear layers and does not include any non-linearities in between, which results in a considerable increase in performance. [6] applies a similar strategy for compact networks, and their experiments show that training such expanded networks yields better results than training the original compact networks. We discuss these importances of studying linear networks in the Related Work section in Appendix B in our paper, which is also our theoretical motivation to study this setting.
>
> **Regarding the motivation for the settings of CVAE and MHVAE**, we note that current state-of-the-art VAE models are mostly deep HVAE and CVAE [8, 9, 10, 11, 12], not standard VAE with only one latent, so our settings are practical and we provide the basis and theoretical tool set for future research of CVAE and HVAE (as per Reviewer vZoP comment). Furthermore, as discussed in Section 4.2 in our paper, **diffusion models** share many similarities with MHVAE model where the encoding process of diffusion models also consists of **consecutive linear maps** with injected noise and **with unlearnable isotropic variances**, and their training loss function is also the ELBO. Therefore, our framework potentially sheds light on the understanding of diffusion models.
>
> We note that MHVAE with two latents is cumbersome to study theoretically. The complexities added when increasing the number of latent variables from one (standard VAE) to two can be seen via the ELBO training problem of two models in our paper: $L_{VAE}$ in Eqn. (3) and $L_{HVAE}$ in page 6. Due to some technical challenges, we currently leave the setting of both learnable encoders as future work.

---

> > ### Author Response · Authors · 2023-11-18
> > **Response to Reviewer VR5S - Part 2**
> >
> > **Q3**: Even in the empirical setting the majority of models studied are restricted to fairly shallow networks. The results are entirely on a simple dataset, MNIST, and therefore it’s hard to get a sense for how the experiments would behave in a more complex setting with deeper networks, modern architectures, and less standardized data.
> >
> > **A3**: To further validate our theoretical insights on more complicated dataset and deeper networks with modern architectures, we have conducted two additional experiments for standard VAE and two-latent MHVAE on CIFAR10, in which we use ResNet18 in the encoder to map the input image to latent space.
> >
> > In particular, for VAE, we conduct an experiment to show the effect of learnability of the encoder’s variance to posterior collapse (similar as Fig. 2a experiment). The results in Figure 5 in Appendix A.2.1 shows that the model with unlearnable variance $\Sigma = \mathbf{I}$ exhibits a lower degree of posterior collapse compared to the model with learnable variance. This experiment strongly validates our theoretical prediction that unlearnable variance can mitigate posterior collapse, even for deeper networks and more complicated datasets. More details on the model architecture and hyperparameters for this experiment can be found in Appendix A.2.1.
> >
> > For MHVAE, we conduct an experiment to assess the impact of varying $\beta_1, \beta_2, \eta_{\text{dec}}$ on posterior collapse. Figure 10 in Appendix A.2.3 demonstrates that reducing $\beta_2$ and $\eta_{\text{dec}}$ mitigates posterior collapse, while decreasing $\beta_1$ yields the opposite effect. This result aligns with our theoretical prediction in Theorem 3 and experimental results on MNIST dataset in Figure 2c in the main paper. More details on the model architecture and hyperparameters for this experiment can be found in Appendix A.2.3.
> >
> > **Q4**: One of the most noticeable absences is that this paper doesn’t consider the effects of SGD and optimization strategies. The setting where global solutions are directly obtainable is not really representative of most real world VAEs which would involve deep nonlinear networks that are most likely to only reach a local optimum. While the experiments themselves do use Adam to train the models, there’s not much analysis of how the optimization strategy interacts with other phenomena.
> >
> > **A4**: We agree with the reviewer that our work does not consider the training dynamics, which is similar to most of the theoretical works of VAE to the best of our knowledge, due to the technical challenges of analyzing the training dynamics. We leave the study of training dynamics of VAE for future research. One of few examples that consider the training dynamics of VAE is [13], where the paper only empirically observes a behavior of the inference network that leads to posterior collapse at the initial stage of training. Below are our thoughts about the theoretical analysis of VAE’s training dynamics:
> >
> > 1.  The work [14] studies the dynamic of deep linear networks under MSE loss and has strong evidence that SGD biases the model towards low-rank solutions, which is also one of the key conditions for posterior collapse occurrence, as stated in our theorems. However, the ELBO training problem is significantly more complex compared to MSE and thus, more challenging to analyze its dynamics.
> >
> > 2. One of the recent phenomena observed in deep neural networks that are also proven to occur at the global optimum is Neural Collapse. Most theoretical results for Neural Collapse characterize the global solutions without analyzing the training dynamics [15, 16, 17, 18]. One of few works in this context that study the training dynamics is [19]. Specifically, [19] restricts the model parameters to the “central path”, where the linear classifier stays as an optimal function of the features throughout the dynamics. Then, by studying renormalized gradient flow along the central path, the authors derive exact dynamics that predict neural collapse. We see that a similar approach may be applied for posterior collapse, and we are leaving it for future work.
> >
> > **Q5**: Many of these weaknesses could be levied against prior work in this space. For what it’s worth this paper does go beyond those works in that they do rigorous theoretical analysis of more complex model types.
> >
> > **A5**: Thank you for your comment and appreciation of our work.

---

> > > ### Author Response · Authors · 2023-11-18
> > > **Response to Reviewer VR5S - Part 3**
> > >
> > > **Q6**: You claim that Wang & Ziyin 2022 arrived at the opposite conclusion as you regarding the role of learnable latent variance. Can you say more about why you would have found different results and what that means?
> > >
> > > **A6**: Compared to Proposition 2 in (Wang & Ziyin, 2022), our Theorem 1 characterizes the global solutions of standard linear VAE for unlearnable $\mathbf{\Sigma}$ with arbitrary elements $\sigma_i$ on the diagonal while Proposition 2 in (Wang & Ziyin, 2022) studies the isotropic $\mathbf{\Sigma} = \sigma^2 \mathbf{I} $ case only. **Another new and interesting result from Theorem 1 in our manuscript is that we prove the left singular matrix $\mathbf{T}$ of the encoder map $\mathbf{V}$ is the sorting matrix that sorts the values of $\mathbf{\Sigma}$ in non-decreasing order**. This property of $\mathbf{T}$ is one of factors that decide whether posterior collapse happens or not, which has not been considered in (Wang & Ziyin, 2022).
> > >
> > > After the Theorem 1 in (Wang & Ziyin, 2022), they claim that “optimizable encoder variance is not essential to posterior collapse problem” (Section 4), “a learnable (data-dependent or not) latent variance is not the cause of posterior collapse” (Section 4.5). Moreover, in Section 4.2 in (Wang & Ziyin, 2022), after Theorem 1, under the setting of unlearnable isotropic variance $\sigma_i = \eta_{enc}, \forall i$, they claimed the existence of posterior collapse when the learned model is low-rank (i.e., some singular values $\lambda$ and $\theta$ are 0’s). In particular, they argue “... the learned model becomes low-rank. Namely, some of the dimensions collapse with the prior”. **We observe that these claims are not totally correct** and we explain in our manuscript why posterior collapse may not happen in the above unlearnable setting (see the first paragraph after our Theorem 1 in the paper). For convenience, we repeat the explanation here.
> > >
> > > We first let the SVD of the encoder matrix $\mathbf{V}$ (i.e, the encoder map) to be $\mathbf{V} =  \mathbf{T} \Lambda \mathbf{S}^{\top}$ then at the global minimizer of the training problem, we prove in Theorem 1 that $\mathbf{T}$ is the matrix that sorts the diagonal values of $\Sigma$ in non-decreasing order. When $\mathbf{V}$ is low-rank, the singular matrix $\Lambda$ has zero rows, then $\Lambda \mathbf{S}^{\top}$ will have zero rows. However, in the case of isotropic $\mathbf{\Sigma} = \eta^{2}\_{enc} \mathbf{I}$ (as considered in Theorem 1 in (Wang & Ziyin, 2022)), $\mathbf{T}$ can be any orthonormal matrix (since $\sigma_i$’s are all equal and in non-decreasing order already), and thus, $\mathbf{V} = \mathbf{T} \Lambda \mathbf{S}^{\top}$ may have no zero rows. Then, for any data $x$, the mean $\mathbf{V} \tilde{x}$ of the approximate posterior $q(z|x) =  \mathcal{N} (\mathbf{V} \tilde{x}, \Sigma)$ might have no zero component, and thus **the posterior may not collapse to the prior $\mathcal{N}(0, \eta^2_{enc})$**.
> > >
> > > For the case of **learnable** and diagonal $\mathbf{\Sigma}$, low-rank condition leads to posterior collapse. This is because at optimality, the diagonality of $\mathbf{\Sigma}$ makes $\mathbf{U}^{\top} \mathbf{U}$ diagonal. Therefore, $\mathbf{U}$ has zero columns and subsequently,  $\mathbf{V}$ will have zero rows (see detailed explanation at the paragraph before Section 4 in our paper). Finally, the corresponding dimensions of $z = \mathbf{V} \tilde{x} + \epsilon$ collapse to its prior $\mathcal{N}(0, \eta^2_{\text{enc}})$. Therefore, our findings suggest that learnability of encoder variance plays an important role in the existence of posterior collapse. We have updated the manuscript to describe carefully the differences between our Theorem 1 and Proposition 2 in (Wang & Ziyin, 2022) in the paragraphs after Theorem 1 in our revision.

---

> > > > ### Author Response · Authors · 2023-11-18
> > > > **Response to Reviewer VR5S - Part 4**
> > > >
> > > > **References:**
> > > >
> > > > [1] Andrew M. Saxe, James L. McClelland, Surya Ganguli. “Exact solutions to the nonlinear dynamics of learning in deep linear neural networks”, 2013
> > > >
> > > > [2] Kenji Kawaguchi. “Deep Learning without Poor Local Minima”, NeurIPS 2016
> > > >
> > > > [3] Moritz Hardt, Tengyu Ma. “Identity Matters in Deep Learning”, ICLR 2017
> > > >
> > > > [4] Thomas Laurent, James von Brecht. “Deep linear neural networks with arbitrary loss: All local minima are global”, ICML 2018
> > > >
> > > > [5] Minyoung Huh et al. “The Low-Rank Simplicity Bias in Deep Networks”, TMLR 2022
> > > >
> > > > [6] Shuxuan Guo, Jose M. Alvarez, Mathieu Salzmann. “ExpandNets: Linear Over-parameterization to Train Compact Convolutional Networks”, NeurIPS 2020
> > > >
> > > > [7] Sanjeev Arora, Nadav Cohen, Elad Hazan. “On the Optimization of Deep Networks: Implicit Acceleration by Overparameterization”, ICML 2018
> > > >
> > > > [8] Child, Rewon. "Very deep vaes generalize autoregressive models and can outperform them on images." arXiv preprint arXiv:2011.10650 (2020)
> > > >
> > > > [9] Vahdat, Arash, and Jan Kautz. "NVAE: A deep hierarchical variational autoencoder", NeurIPS 2020
> > > >
> > > > [10] Sønderby, Casper Kaae, et al. "Ladder variational autoencoders”, NeurIPS 2016
> > > >
> > > > [11] Fang, Le, et al. "Transformer-based conditional variational autoencoder for controllable story generation." arXiv preprint arXiv:2101.00828
> > > >
> > > > [12] Sohn, Kihyuk, Honglak Lee, and Xinchen Yan. "Learning structured output representation using deep conditional generative models", NeurIPS 2015
> > > >
> > > > [13] He, Junxian, et al. "Lagging inference networks and posterior collapse in variational autoencoders." arXiv preprint arXiv:1901.05534
> > > >
> > > > [14] Arora, Sanjeev, et al. "Implicit regularization in deep matrix factorization", NeurIPS 2019
> > > >
> > > > [15] Zhu, Zhihui, et al. "A geometric analysis of neural collapse with unconstrained features", NeurIPS 2021
> > > >
> > > > [16] Tirer, Tom, and Joan Bruna. "Extended unconstrained features model for exploring deep neural collapse", ICML 2022
> > > >
> > > > [17] Dang, Hien, et al. "Neural collapse in deep linear networks: From balanced to imbalanced data", ICML 2023
> > > >
> > > > [18] Súkeník, Peter, Marco Mondelli, and Christoph Lampert. "Deep Neural Collapse Is Provably Optimal for the Deep Unconstrained Features Model." arXiv preprint arXiv:2305.13165.
> > > >
> > > > [19] Han, X. Y., Vardan Papyan, and David L. Donoho. "Neural collapse under mse loss: Proximity to and dynamics on the central path”, ICLR 2022

---

### Official Review · Reviewer_Bn1w · 2023-11-05

**Soundness:** 4 excellent
**Presentation:** 3 good
**Contribution:** 3 good
**Rating:** 6
**Confidence:** 2

**Summary:**

This work extends previous theoretical work on the linear variational autoencoder (VAE), showing the conditions under which linear conditional VAEs (CVAEs) and Markovian hierarchical VAEs (MHVAEs) will exhibit posterior collapse. Empirical experiments are conducted that supports theoretical results in both linear and nonlinear cases.

**Strengths:**

- **The results are potentially novel and can have interesting implications.** This work extends the previous results on more complex VAE models which are widely used in practice. This might give practical directions for people deploying these variant of VAEs that previous work did not cover.
- **Experiments clearly support the main theoretical results.** Both linear and nonlinear models are trained for different parameters and subsets of data. The results very clearly support the theoretical predictions and shows evidence on how the linear results can be predictive of the general nonlinear case.

**Weaknesses:**

- **The paper seems to claim a result that has already been proven.** Theorem 1 seems to be exactly identical to proposition 2 in (Wang & Ziying, 2022). Maybe the functional form of the result is identical but the authors have more general assumptions on $\sigma_i$ (I doubt this after having scanned the proof in Wang & Ziying). In either case, the statement that this theorem is one of the novel contributions is false or at best misleading.
- **It is unclear how much novelty there is in extending the proof to CVAEs and MHVAEs.** Since I did not follow the proof for theorem 2 and 3 closely, and I am also not familiar with the literature, I cannot be certain whether extending the existing results on linear VAEs to CVAEs and MHVAEs is methodologically novel. In particular, the expression for $\omega^*$ in theorem 2 seems quite similar to that of theorem 1 (which I do not consider novel, see previous point), and it is unclear to me if the derivation follows almost identical strategies. I think it might be helpful if the authors can discuss why extending the proof to the CVAE and MHVAE cases are interesting and nontrivial. This might just be a minor concern as one can argue that the implications of the results might be the more important and interesting aspect of this work.
- **Graphics are not very easy to read.** It is difficult to see how the results shown in the plots support the theoretical results. This is only a minor critique and can be easily improved by using a different coloring scheme or just rearranging the labels in a vertical list and/or remind the reader what the theoretical predictions are (larger $\beta1$ and smaller $\beta2$ is good, etc.).

**Questions:**

- Can you clarify on the novelty claim for theorem 1?
- On the hierarchical VAE results, you mentioned that it is advisible to create separate maps between the input data and each level of hierarchy, which seems closely related to the ideas in (Sønderby et al., 2016; Zhao et al., 2017; Vahdat & Kautz, 2020). Maybe it will be helpful to discuss these models since the theoretical results here seem to provide motivation for them?

References:

Casper Kaae Sønderby, Tapani Raiko, Lars Maaløe, Søren Kaae Sønderby, and Ole Winther. Ladder variational autoencoders. Advances in neural information processing systems, 29, 2016.

Zhao, S., Song, J., & Ermon, S. (2017). Learning hierarchical features from generative models. arXiv preprint arXiv:1702.08396.

Arash Vahdat and Jan Kautz. Nvae: A deep hierarchical variational autoencoder. Advances in neural information processing systems, 33:19667–19679, 2020.

---

> ### Author Response · Authors · 2023-11-18
> **Response to Reviewer Bn1w - Part 1**
>
> Dear Reviewer Bn1w,
>
> Thank you for your thoughtful review and valuable feedback. Below we address your concerns. We have also revised the current manuscript according to the suggestions of the reviewers. The changes are in blue color. Please kindly see the revised manuscript, as well as our General Response and Summary of Revision.
>
> **Q1**: The paper seems to claim a result that has already been proven. Theorem 1 seems to be exactly identical to proposition 2 in (Wang & Ziying, 2022). Maybe the functional form of the result is identical but the authors have more general assumptions on $\sigma_i$. In either case, the statement that this theorem is one of the novel contributions is false or at best misleading.
>
> **Q2**: Can you clarify on the novelty claim for theorem 1?
>
> **A1 + A2**: Thank you for your comment. We agree with the reviewer that the word “novel” may be inappropriate here. In our revision, we have changed the wording from “Our novel contributions” to “Our contributions” in Section 3. Please allow us to clarify the contributions of Theorem 1 in our manuscript and the differences with Proposition 2 in (Wang & Ziyin, 2022) as follows.
>
> First, compared to Proposition 2 in (Wang & Ziyin, 2022), our Theorem 1 characterizes the global solutions of standard linear VAE for unlearnable $\mathbf{\Sigma}$ with arbitrary elements $\sigma_i$ on the diagonal while Proposition 2 and Theorem 1 in (Wang & Ziyin, 2022) studies the isotropic $\mathbf{\Sigma} = \sigma^2 \mathbf{I} $ case only. **Another new and interesting result from Theorem 1 in our manuscript is that we prove the left singular matrix $\mathbf{T}$ of the encoder map $\mathbf{V}$ is the sorting matrix that sorts the values of $\mathbf{\Sigma}$ in non-decreasing order**. This property of $\mathbf{T}$ is one of the factors that decide whether posterior collapse happens or not, which has not been considered in (Wang & Ziyin, 2022).
>
> Second, after Theorem 1 in (Wang & Ziyin, 2022), they claim that “optimizable encoder variance is not essential to posterior collapse problem” (Section 4) and “a learnable (data-dependent or not) latent variance is not the cause of posterior collapse” (Section 4.5). Moreover, in Section 4.2 of (Wang & Ziyin, 2022), after Theorem 1, under the setting of unlearnable isotropic variance $\sigma_i = \eta_{enc}, \forall i$, they claim the existence of posterior collapse when the learned model is low-rank (i.e., some singular values $\lambda$ and $\theta$ are 0’s). In particular, they argue “...the learned model becomes low-rank. Namely, some of the dimensions collapse with the prior”. **We observe that these claims are not totally correct**, and we explain in our manuscript why posterior collapse may not happen in the above unlearnable setting (see the paragraph after our Theorem 1). For convenience, we repeat the explanation below.
>
> We first let the SVD of the encoder matrix $\mathbf{V}$ (i.e, the encoder map) to be $\mathbf{V} =  \mathbf{T} \Lambda \mathbf{S}^{\top}$ then at the global minimizer of the training problem, we prove in Theorem 1 that $\mathbf{T}$ is the matrix that sorts the diagonal values of $\Sigma$ in non-decreasing order. When $\mathbf{V}$ is low-rank, $\Lambda$ will have zero rows, then $\Lambda \mathbf{S}^{\top}$ will have zero rows. However, in the case of isotropic $\mathbf{\Sigma} = \eta_{enc}^{2} \mathbf{I}$ (as considered in Theorem 1 in (Wang & Ziyin, 2022)), $\mathbf{T}$ can be any orthonormal matrix (since $\sigma_i$’s are all equal and in non-decreasing order already), and thus, $\mathbf{V} = \mathbf{T} \Lambda \mathbf{S}^{\top}$ may have no zero rows. Then, for any data $x$, the mean $\mathbf{V} \tilde{x}$ of the approximate posterior $q(z|x) =  \mathcal{N} (\mathbf{V} \tilde{x}, \Sigma)$ might have no zero component, and thus the **posterior may not collapse to the prior $\mathcal{N}(0, \eta^2_{enc})$**. On the other hand, if all values $\sigma_i$’s are distinct, Theorem 1 suggests that $\mathbf{T}$ is a permutation matrix and hence, $\mathbf{V}$ will have zero rows, which corresponds with the dimensions of latent $z$ that collapse.
>
> For the case of **learnable** $\mathbf{\Sigma}$, low-rank condition leads to posterior collapse. This is because at optimality, the diagonality of $\mathbf{\Sigma}$ makes $\mathbf{U}^{\top} \mathbf{U}$ diagonal. Therefore, $\mathbf{U}$ has zero columns, and subsequently,  $\mathbf{V}$ will have zero rows (see detailed explanation in the paragraph before Section 4 in our manuscript). Finally, the corresponding dimensions of $z = \mathbf{V} \tilde{x} + \epsilon$ collapse to its prior $\mathcal{N}(0, \eta^2_{\text{enc}})$. Thus, our findings suggest that the learnability of encoder variance plays an important role in the existence of posterior collapse. We have updated the manuscript to describe carefully the differences between our Theorem 1 and Proposition 2 in (Wang & Ziyin, 2022) in the paragraphs after Theorem 1 in our revision.

---

> > ### Author Response · Authors · 2023-11-18
> > **Response to Reviewer Bn1w - Part 2**
> >
> > **Q3**: **It is unclear how much novelty there is in extending the proof to CVAEs and MHVAEs**. Since I did not follow the proof for theorem 2 and 3 closely, and I am also not familiar with the literature, I cannot be certain whether extending the existing results on linear VAEs to CVAEs and MHVAEs is methodologically novel. I think it might be helpful if the authors can discuss why extending the proof to the CVAE and MHVAE cases are interesting and nontrivial.
> >
> > This might just be a minor concern as one can argue that the implications of the results might be the more important and interesting aspect of this work.
> >
> > **A3**: Thank you for the question. We offer an understanding of more complex and practical VAE architectures (as you mention) and “provide the basis and theoretical toolset for future research” (as per Reviewer vZoP comment). The proofs to find the global minima of linear CVAE and MHVAE are methodologically novel and more challenging than those in linear VAE due to the following reasons.
> >
> > For our proof in this work, under the setting of **learnable encoder variance $\Sigma$**, we use the zero gradient condition of the loss function to convert $\Sigma$ into a function of the other variables to reduce the number of variables. Compared to (Wang & Ziyin, 2022), their proof first considers the unlearnable and pre-defined $\mathbf{\Sigma}$ case (see their proof at Appendix Section E.2) and optimizes the encoder and decoder map singular values $\lambda$’s and $\theta$’s as a function of $\sigma$’s. Then, they plug that optimal $\lambda$’s and $\theta$’s back to the loss function. Next, they treat $\sigma$’s as learnable variables and find the optimal $\sigma$’s. However, for the proof for unlearnable $\Sigma$ case in Section E.2 at page 21 of (Wand & Ziyin, 2022), the term $\operatorname{Trace} (\mathbf{U} \Sigma \mathbf{U}^{\top})$ in Eqn (77) and the term $\operatorname{Trace} ( \Lambda^{\top} \Lambda \Sigma)$ in Eqn. (79) are equal based on their proof. We see this is **not entirely correct** because if they let the SVD of $\mathbf{U}$ is $\mathbf{U} = \mathbf{Q} \Lambda \mathbf{P}$, we have the term $\operatorname{Trace} (\mathbf{U} \Sigma \mathbf{U}^{\top}) = \operatorname{Trace}(\Lambda \mathbf{P} \Sigma \mathbf{P}^{\top} \Lambda^{\top}) = \operatorname{Trace} (\Lambda^{\top} \Lambda \mathbf{P} \Sigma \mathbf{P}^{\top}) \neq \operatorname{Trace}( \Lambda^{\top} \Lambda \Sigma)$ in general. One trivial case where these two terms are equal is $\Sigma$ is isotropic, i.e., $\Sigma = \sigma^2 \mathbf{I}$, while if $\Sigma$ contains distinct values, this equation is **generally not correct**. Thus, the derivation of optimal $\sigma$’s for learnable $\mathbf{\Sigma}$ in Proposition 3 in (Wang & Ziyin, 2022) with distinct values of $\sigma$’s, is not entirely correct. Our approach alleviates this problem. As a result, regarding the optimal values of $\sigma_i$, we find that the $i$-th element of $\Sigma$ is not necessarily computed via the $i$-th largest singular value $\lambda_i$ as in Proposition 3 in (Wang & Ziyin, 2022). Instead, the ordering between $\{ \sigma_i \}$ and $\{ \theta_i \}$ will depend on right singular matrix $\mathbf{P}$ of $\mathbf{U}^{*}$ (see Eqn. (7) of our Theorem 2).
> >
> > Furthermore, since CVAE has two additional mappings compared to VAE, i.e., the map from the output $y$ to latent $z$ in the encoder and from input condition $x$ to output $y$ in the decoder, the loss function is more cumbersome and requires different calculations compared to the standard VAE case. This can be demonstrated via the larger number of optimizable variables of the two additional mappings, the gradient calculations in Eqn. (13) at Appendix C.1 on page 22, and subsequent calculations to simplify the loss function on page 23 in our manuscript. We note that for standard VAE, using only one zero gradient condition $\frac{\partial L_{VAE}}{\partial \mathbf{V}} = 0$ can deduce the encoder $\mathbf{V}$ as a function of the decoder $\mathbf{U}$ and the matrix $\mathbf{Z} = \mathbb{E} (x \tilde{x}^{\top})$. Thus, simplifying the optimization problem of standard VAE is much easier compared to CVAE.

---

> > > ### Author Response · Authors · 2023-11-18
> > > **Response to Reviewer Bn1w - Part 3**
> > >
> > > **A3 (Continue)**:
> > >
> > > For MHVAE, both the architecture and the training problem are more complicated than the standard VAEs with only one latent variable. Adding one more level of latent requires two more mappings to map the first latent to the second latent in the encoding process and vice versa for the decoding process. The training problem of maximizing the ELBO must have an extra KL-regularizer term between the two latents, i.e., $D_{\text{KL}}(q(z_1 | x) || p(z_1 | z_{2}))$. This term significantly complicates the training problem, and our proof uses new techniques to handle it. Specifically, we prove a property that $\mathbf{V}\_{1} \mathbf{V}\_{1}^{\top}$ and $\mathbf{U}\_{2} \mathbf{U}\_{2}^{\top}$ are **simultaneously diagonalizable** (see page 38 in our manuscript). Hence, we are able to convert the zero gradient condition into relations of their singular values $\lambda$’s and $\omega$’s (see Eqn. (86) at page 38). Thanks to these relations between $\lambda$’s and $\omega$’s, the loss function now can be converted to a function of singular values. Again, the formulations for MHVAE are significantly more cumbersome and require different calculations compared to linear VAE cases, which can be seen via the loss $L_{HVAE}$ in Section 4.2, the gradient calculation from Eqn. (75)-(78), and how we leverage these critical points conditions to simplify the loss function in Eqn. (79)-(88) in our manuscript.
> > >
> > > **Q4**: **Graphics are not very easy to read**. It is difficult to see how the results shown in the plots support the theoretical results. This is only a minor critique and can be easily improved by using a different coloring scheme or just rearranging the labels in a vertical list and/or remind the reader what the theoretical predictions are.
> > >
> > > **A4**: Thanks for your suggestion. We have revised the figures in our manuscript as follows.
> > > 1. In the revision paper, we added more detailed captions to recall the theoretical predictions and what the figures are implying. For the experiments in the main paper: Figure 2a shows that unlearnable encoder variance can prevent posterior collapse, compared with learnable one; Figure 2b shows that a small $\beta$ or $\eta_{dec}$ can mitigate collapse for ReLU CVAE; Figure 2c and Figure 3b imply that a small $\beta_2$ or $\eta_{dec}$ can mitigate collapse for ReLU MHVAE, while a small $\beta_1$ will have the opposite effect; and Figure 3a implies that dataset with smaller value of singular values $\theta$ of the matrix $\mathbf{E}$ defined in Theorem 2 is more prone to posterior collapse.
> > >
> > > 2. We rearrange the labels in the vertical list for Figure 2 in main text for easier tracking of the graph.
> > >
> > > 3.  We allocate more space for Figure 2 for easier reading.
> > >
> > > **Q5**: On the hierarchical VAE results, you mentioned that it is advisible to create separate maps between the input data and each level of hierarchy, which seems closely related to the ideas in (Sønderby et al., 2016; Zhao et al., 2017; Vahdat & Kautz, 2020). Maybe it will be helpful to discuss these models since the theoretical results here seem to provide motivation for them?
> > >
> > > **A5**: Thanks for your suggestion and the references. We have added some discussion regarding this insight in the Remark 1 at Section 4.2 in our revised manuscript. We state that many HVAE models use additional connections (e.g., skip-connection) between the input and multiple layers of latent variables [1, 2, 3, 4]. Furthermore, we also discuss that [5] empirically showed 5-latent MHVAE cannot train higher-level latents to be active and suggested using other methods such as batch normalization or “warm-up” training scheme can mitigate collapse for this model.
> > >
> > > **References**:
> > >
> > > [1] Vahdat, Arash, and Jan Kautz. "NVAE: A deep hierarchical variational autoencoder." NeurIPS 2020
> > >
> > > [2] Child, Rewon. "Very deep vaes generalize autoregressive models and can outperform them on images." arXiv preprint arXiv:2011.10650 (2020).
> > >
> > > [3] Maaløe, Lars, Marco Fraccaro, and Ole Winther. "Semi-supervised generation with cluster-aware generative models." arXiv preprint arXiv:1704.00637 (2017).
> > >
> > > [4] Zhao, Shengjia, Jiaming Song, and Stefano Ermon. "Learning hierarchical features from generative models.", ICML 2017
> > >
> > > [5] Sønderby, Casper Kaae, et al. "Ladder variational autoencoders.", ICML 2016

---

> > > > ### Author Response · Authors · 2023-11-22
> > > > **Response to Reviewer Bn1w - Any further questions on our current draft?**
> > > >
> > > > Dear Reviewer Bn1w,
> > > >
> > > > We would like to thank you again for your thoughtful reviews and valuable feedback.
> > > >
> > > > We would appreciate it if you could let us know if our responses have addressed your concerns and whether you still have any other questions on the current draft and our rebuttal.
> > > >
> > > > We would be happy to do any follow-up discussion or address any additional comments.
> > > >
> > > > Best regards,
> > > >
> > > > Authors

---

### Official Review · Reviewer_vZoP · 2023-11-10

**Soundness:** 4 excellent
**Presentation:** 4 excellent
**Contribution:** 4 excellent
**Rating:** 8
**Confidence:** 4

**Summary:**

This work provides a theoretical analysis of posterior collapse in Variational Autoencoder (VAE) models. First the authors extend previous results for the linear VAE case (Variational Autoencoders with only a single linear layer as encoder and decoder) and provide a novel condition for posterior collapse. Then these results are extended and the authors provide conditions for posterior collapse for two further VAE models: the Conditional VAE (CVAE), and the Markovian Hierarchical VAE (MHVAE).

**Strengths:**

This work provides important theoretical insights into the conditions for posterior collapse in Variational Autoencoders. Although these conditions are only presented for the linear case, and for MHVAE the case that only one of the two encoder variances is learnable, this work provides an important step forward for the understanding of the inner workings of VAEs and provides the basis and theoretical tool set for future research. The empirical evaluation confirms that the theoretical insights gained for the single layer encoder and decoder case extend, at least empirically, to the nonlinear (multi layer) case.

**Weaknesses:**

The conditions provided in the paper are only for the linear (single layer encoder and decoder) case, which of course limits the practical applicability of the framework. Yet, I believe that the work provides an important step forward in the understanding of VAEs and do not see a big drawback in this.

**Questions:**

In Fig. 3 (b) the case for beta_1=1.0 and beta_2=1.0 has been left out. Just for completeness it might be interesting to visualize the result for this "standard" ELBO (beta_1=beta_2=1.0).

**Details Of Ethics Concerns:**

No ethical concerns

---

> ### Author Response · Authors · 2023-11-18
> **Response to Reviewer vZoP**
>
> Dear Reviewer vZoP,
>
> Thank you for your thoughtful review and valuable feedback. Below we address your concerns. We have also revised our manuscript according to the suggestions of the reviewers. The changes are in blue color. Please kindly see the revised manuscript, as well as our General Response and Summary of Revision.
>
> **Q1**: The conditions provided in the paper are only for the linear (single layer encoder and decoder) case, which of course limits the practical applicability of the framework. Yet, I believe that the work provides an important step forward in the understanding of VAEs and do not see a big drawback in this.
>
> **A1**: Thank you for your comment and appreciation of our work. We agree with the reviewer that our theoretical results focus on the settings of linear networks. However, the theoretical analysis of nonlinear networks is very challenging and, in fact, there has been no rigorous theory for deep nonlinear networks yet to the best of our knowledge. Even for the linear case, the complete theoretical understanding in the VAE setting is still lacking. We aim to fill this gap and use the insights from our work for deep linear VAEs to empirically examine and improve the deep nonlinear VAEs: we have experiments at Figure 2 + Figure 3 in our main paper and more experiments on nonlinear networks in the Appendix that support our theoretical insights. Throughout this process, our frameworks and theorems play an important role as a starting point for further theoretical and empirical study of the nonlinear conditional VAEs, hierarchical VAEs or even diffusion models, potentially.
>
> We also want to note that analyzing deep linear networks has been an important step in studying deep nonlinear networks. For example, using only linear regression, [1] can recover several phenomena observed in large-scale deep nonlinear networks, including the double descent phenomenon [2]. [3, 4, 5, 6] show that the optimization of deep linear models exhibits similar properties to those of the optimization of deep nonlinear models. In practice, deep linear networks can help improve the training and performance of deep nonlinear networks [7, 8, 9]. Specifically, [7] empirically proves that linear overparameterization in nonlinear networks improves generalization on classification tasks (see Section 4 in [7]). In particular, [7] expands each linear layer into a succession of multiple linear layers and does not include any non-linearities in between, which results in a considerable increase in performance. [8] applies a similar strategy for compact networks, and their experiments show that training such expanded networks yields better results than training the original compact networks. [9] shows that linear overparameterization, i.e., the use of a deep linear network in place of a classic linear model, induces on gradient descent a particular preconditioning scheme that can accelerate optimization. The preconditioning scheme that deep linear layers introduce can be interpreted as using momentum and adaptive learning rate.
>
> **Q2**: In Fig. 3 (b) the case for beta_1=1.0 and beta_2=1.0 has been left out. Just for completeness it might be interesting to visualize the result for this "standard" ELBO (beta_1=beta_2=1.0).
>
> **A2**: Thanks for your suggestion. We already have this setting in the paper. In particular, we conducted the experiment for either $\beta_1$ or  $\beta_2$ is 1 and included the results in Figure 4 on page 14 of the Appendix. We agree with the reviewer that this setting is standard, and we have moved it back to the main text in our revision.
>
>
> **References**:
>
> [1] Trevor Hastie et al. “Surprises in High-Dimensional Ridgeless Least Squares Interpolation”, Annals of statistics, 2022
>
> [2] Preetum Nakkiran et al. “Deep Double Descent: Where Bigger Models and More Data Hurt”, Journal of Statistical Mechanics, 2021
>
> [3] Andrew M. Saxe, James L. McClelland, Surya Ganguli. “Exact solutions to the nonlinear dynamics of learning in deep linear neural networks”, 2013
>
> [4] Kenji Kawaguchi. “Deep Learning without Poor Local Minima”, NeurIPS 2016
>
> [5] Moritz Hardt, Tengyu Ma. “Identity Matters in Deep Learning”, ICLR 2017
>
> [6] Thomas Laurent, James von Brecht. “Deep linear neural networks with arbitrary loss: All local minima are global”, ICML 2018
>
> [7] Minyoung Huh et al. “The Low-Rank Simplicity Bias in Deep Networks”, TMLR 2022
>
> [8] Shuxuan Guo, Jose M. Alvarez, Mathieu Salzmann. “ExpandNets: Linear Over-parameterization to Train Compact Convolutional Networks”, NeurIPS 2020
>
> [9] Sanjeev Arora, Nadav Cohen, Elad Hazan. “On the Optimization of Deep Networks: Implicit Acceleration by Overparameterization”, ICML 2018

---

### Author Response · Authors · 2023-11-18
**General Response - Part 1**

Dear AC and Reviewers,

Thanks for your thoughtful reviews and valuable comments, which have helped us improve the paper significantly. We are encouraged by the endorsements that:

1) Our work provides important theoretical insights into the conditions for posterior collapse in VAE (Reviewer vZoP, Bn1w, fUP2)

2) Our work puts forward theoretical analysis on more complex VAE models which are widely used in practice (Reviewer Bn1w, VR5S), provides the basis and theoretical tool set for future research (Reviewer vZoP) and our methodology is solid (Reviewer 1CHE)

3) The experiments clearly support theoretical results and shows evidence on how the linear results can be predictive of the general nonlinear case (Reviewer vZoP, Bn1w, VR5S)

There are two main concerns from the reviewers about 1) the theoretical motivations of our analysis for linear CVAE and MHVAE, as well as the implications over previous works that study linear VAE, and 2) the opposite conclusion with previous work regarding the role of learnable latent variance. We summarize our response to these concerns below.

**Q1**:  What are the theoretical motivations of our analysis for linear CVAE and MHVAE, and the implications over previous works that study linear VAE?

**A1**:

1/ **Implications of our theoretical results for CVAE and MHVAE over previous works that study linear VAE**

For CVAE, our theoretical analysis is the first to prove the existence of posterior collapse in CVAE, an important and popularly used variant of VAE [1, 2, 3]. We rigorously study the effect of parameters to the rank of the models and the level of posterior collapse at optimality, including $\theta$’s (the singular values of matrix $E$ , defined in Theorem 2), $\beta$ (scaling scalar of the KL term of ELBO at Equation (5) in our paper) and $\eta_{dec}$ (scaling scalar of the variance of the generating distribution). The results imply that a sufficiently small $\beta$ and/or $\eta_{dec}$ can mitigate posterior collapse. Although this insight sounds similar to an insight drawn from standard VAE, in our opinion, it is **still a new insight since CVAE is a different variant of VAE with different architecture and mechanism**. Especially, we prove that **the correlation of the input condition and output of training data relates to the collapse level** (see the paragraph after Theorem 2 in our paper). This insight obviously cannot be derived from the standard VAE setting which does not incorporate input conditions into the generative process.

For MHVAE, our theoretical analysis is again the first to prove the existence of posterior collapse in MHVAE, another important and popularly used variant of VAE [4, 5, 6, 7]. We also rigorously study the effect of parameters to the rank of the models and the level of posterior collapse at optimality, including $\theta$’s (the singular values of matrix $Z = \mathbb{E}(x \tilde{x})$), $\beta_1$, $\beta_2$ and $\eta_{dec}$. Moreover, since the ELBO loss of MHVAE consists of multiple KL-regularized terms, we study the effect of each KL term and the trade-offs between latents via the magnitude of the $\beta_1$ and $\beta_2$. We find that **decreasing $\beta_2$ can help to alleviate posterior collapse and increase the rank of the encoder and decoder maps, while decreasing $\beta_1$ will have the opposite effect**. This insight cannot be derived from the standard VAE setting which has only one latent and one KL-regularized term in the ELBO. We also observe that **when any latent variable of MHVAE suffers complete collapse, its higher-level latents become useless**. Thus, we suggest creating additional mapping between the input data and every latent to prevent this adverse situation.

Furthermore, we find that **unlearnable encoder variance can prevent posterior collapse for VAE, CVAE and MHVAE** (see Section 3 in our paper). This insight is also novel.

2/ **Theoretical contributions and motivations**

As per Reviewer vZoP comment, our paper **“provides an important step forward for the understanding of the inner workings of VAEs and provides the basis and theoretical tool set for future research”**. Specifically, current state-of-the-art VAE models are mostly CVAE and HVAE [1, 2, 3, 4, 5, 6, 7], not standard VAE with only one latent, so our settings are more practical compared with prior works, and provide the basis and theoretical tool set for future research of CVAE and HVAE. Furthermore, as the discussion in Section 4.2 in our paper, **diffusion models** share many similarities with deep MHVAE model where the encoding process of diffusion models also consists of **consecutive linear maps** with injected noise and with **unlearnable isotropic variance**, and their training loss function is also ELBO. Therefore, our framework potentially sheds light on the understanding of diffusion models.

---

> ### Author Response · Authors · 2023-11-18
> **General Response - Part 2**
>
> **A1 (Continue)**:
>
> 2/ **Theoretical contributions and motivations**
>
> We acknowledge that our theoretical results study the simplistic settings of **linear networks**. However, the theoretical analysis of nonlinear networks is very challenging and, in fact, there has been no rigorous theory for deep nonlinear networks yet to the best of our knowledge. Even for the linear case, a complete theoretical understanding of the VAE setting is still lacking. We aim to fill this gap and use the insights from our work for deep linear VAEs to empirically examine and improve deep nonlinear VAEs.
>
> We also want to emphasize that analyzing deep linear networks has been an important step in studying deep nonlinear networks. [8, 9, 10, 11] show that the optimization of deep linear models exhibits similar properties to those of the optimization of deep nonlinear models. In practice, deep linear networks can help improve the training and performance of deep nonlinear networks [12, 13, 14]. Specifically, [12] empirically proves that linear overparameterization in nonlinear networks improves generalization on classification tasks (see Section 4 in [12]). In particular, [12] expands each linear layer into a succession of multiple linear layers and does not include any non-linearities in between, which results in a considerable increase in performance.
>
> 3/ **Our proofs are methodologically novel and are not just a generalization from linear VAE**
>
> For example, for MHVAE, both the architecture and the training problem are more complicated than the standard VAEs with only one latent variable. Adding one more level of latent requires two more mappings to map the first latent to the second latent in the encoding process and vice versa for the decoding process. The training problem of maximizing the ELBO must have an extra KL-regularizer term between the two latents, i.e., $D_{\text{KL}}(q(z_1 | x) || p(z_1 | z_{2}))$. This term significantly complicates the training problem and our proof uses new techniques to handle it. Specifically, we prove a property that $\mathbf{V}\_{1} \mathbf{V}\_{1}^{\top}$ and $\mathbf{U}\_{2} \mathbf{U}\_{2}^{\top}$ are simultaneously diagonalizable (see page 38), and thus, we are able to convert the zero gradient condition into relations of their singular values $\lambda$’s and $\omega$’s (see Eqn. (86) at page 38). Thanks to these relations between $\lambda$’s and $\omega$’s , the loss function now can be converted to a function of singular values. Again, the formulations for MHVAE are significantly more cumbersome and require different calculations compared to linear VAE cases, which can be seen via the gradient calculation from Eqn. (75)-(78) and how we leverage these critical points conditions to simplify the loss function in Eqn. (79)-(88).
>
> **Q2**: The opposite conclusion with previous work (Wang & Ziyin, 2022) regarding the role of learnable latent variance.
>
> **A2**: After the Theorem 1 in (Wang & Ziying, 2022), they claim that “optimizable encoder variance is not essential to posterior collapse problem” (Section 4), “a learnable (data-dependent or not) latent variance is not the cause of posterior collapse” (Section 4.5). Moreover, in Section 4.2 in (Wang & Ziying, 2022), after Theorem 1, under the setting of unlearnable isotropic variance $\sigma_i = \eta_{enc}, \forall i$, they claimed the existence of posterior collapse when the learned model is low-rank (i.e., some singular values $\lambda$ and $\theta$ are 0’s). In particular, they argue “... the learned model becomes low-rank. Namely, some of the dimensions collapse with the prior”. **We observe that these claims are not totally correct** and we explain in our manuscript why posterior collapse may not happen in the above unlearnable setting (see the first paragraph after our Theorem 1 in the paper). For convenience, we repeat the explanation below.

---

> > ### Author Response · Authors · 2023-11-18
> > **General Response - Part 3**
> >
> > **A2 (Continue)**:
> >
> > We first let the SVD of the encoder matrix $\mathbf{V}$ (i.e, the encoder map) to be $\mathbf{V} =  \mathbf{T} \Lambda \mathbf{S}^{\top}$ then at the global minimizer of the training problem, we prove in Theorem 1 that $\mathbf{T}$ is the matrix that sorts the diagonal values of $\Sigma$ in non-decreasing order. This property of $\mathbf{T}$ is one of factors that decide whether posterior collapse happens or not, which has not been considered in (Wang & Ziyin, 2022). When $\mathbf{V}$ is low-rank, the singular matrix $\Lambda$ has zero rows, then $\Lambda \mathbf{S}^{\top}$ will have zero rows. However, in the case of isotropic $\mathbf{\Sigma} = \eta^{2}\_{enc} \mathbf{I}$ (as considered in Theorem 1 in (Wang & Ziying\, 2022)), $\mathbf{T}$ can be any orthonormal matrix (since $\sigma_i$’s are all equal and in non-decreasing order already), and thus, $\mathbf{V} = \mathbf{T} \Lambda \mathbf{S}^{\top}$ may have no zero rows. Then, for any data $x$, the mean $\mathbf{V} \tilde{x}$ of the approximate posterior $q(z|x) =  \mathcal{N} (\mathbf{V} \tilde{x}, \Sigma)$ might have no zero component, and thus **the posterior may not collapse to the prior $\mathcal{N}(0, \eta^2_{enc})$**.
> >
> > For the case of **learnable** and diagonal $\mathbf{\Sigma}$, low-rank condition leads to posterior collapse. This is because at optimality, the diagonality of $\mathbf{\Sigma}$ makes $\mathbf{U}^{\top} \mathbf{U}$ diagonal. Therefore, $\mathbf{U}$ has zero columns and subsequently,  $\mathbf{V}$ will have zero rows (see detailed explanation at the paragraph before Section 4 in our paper). Finally, the corresponding dimensions of $z = \mathbf{V} \tilde{x} + \epsilon$ collapse to its prior $\mathcal{N}(0, \eta^2_{\text{enc}})$. **Therefore, our findings suggest that learnability of encoder variance plays an important role in the existence of posterior collapse.**
> >
> > **References:**
> >
> > [1] Fang, Le, et al. "Transformer-based conditional variational autoencoder for controllable story generation." arXiv preprint arXiv:2101.00828
> >
> > [2] Sohn, Kihyuk, Honglak Lee, and Xinchen Yan. "Learning structured output representation using deep conditional generative models", NeurIPS 2015
> >
> > [3] Walker, Jacob, et al. "An uncertain future: Forecasting from static images using variational autoencoders.", ECCV 2016
> >
> > [4] Child, Rewon. "Very deep vaes generalize autoregressive models and can outperform them on images." arXiv preprint arXiv:2011.10650 (2020)
> >
> > [5] Vahdat, Arash, and Jan Kautz. "NVAE: A deep hierarchical variational autoencoder", NeurIPS 2020
> >
> > [6] Sønderby, Casper Kaae, et al. "Ladder variational autoencoders”, NeurIPS 2016
> >
> > [7] Maaløe, Lars, Marco Fraccaro, and Ole Winther. "Semi-supervised generation with cluster-aware generative models." arXiv preprint arXiv:1704.00637 (2017).
> >
> > [8] Andrew M. Saxe, James L. McClelland, Surya Ganguli. “Exact solutions to the nonlinear dynamics of learning in deep linear neural networks”, 2013
> >
> > [9] Kenji Kawaguchi. “Deep Learning without Poor Local Minima”, NeurIPS 2016
> >
> > [10] Moritz Hardt, Tengyu Ma. “Identity Matters in Deep Learning”, ICLR 2017
> >
> > [11] Thomas Laurent, James von Brecht. “Deep linear neural networks with arbitrary loss: All local minima are global”, ICML 2018
> >
> > [12] Minyoung Huh et al. “The Low-Rank Simplicity Bias in Deep Networks”, TMLR 2022
> >
> > [13] Shuxuan Guo, Jose M. Alvarez, Mathieu Salzmann. “ExpandNets: Linear Over-parameterization to Train Compact Convolutional Networks”, NeurIPS 2020
> >
> > [14] Sanjeev Arora, Nadav Cohen, Elad Hazan. “On the Optimization of Deep Networks: Implicit Acceleration by Overparameterization”, ICML 2018

---

### Author Response · Authors · 2023-11-18
**Summary of Revision**

Dear AC and Reviewers,

Incorporating the comments and suggestions from all reviewers, besides fixing typos, we have made the following main changes in the revised paper:

- In Section 3, before and after Theorem 1, we gave a more detailed description of the differences between our Theorem 1 and the result in (Wang & Ziyin, 2022) about the role of learnability encoder variance to posterior collapse.

- Before the Theorems, we added the descriptions about what the theorems try to accomplish to facilitate the reading.

- We moved more experiments from the Appendix to the main paper. Please see Figure 2a and section 5.1 regarding the new added experiments. The experiment compares the degree of posterior collapse of linear VAE under learnable and unlearnable encoder variance settings.

- To validate our theoretical insights on more complex dataset and deeper architecture, we performed two more experiments on CIFAR10 dataset that use ResNet18 in the encoder and include the results in Figure 5 and Figure 10 in the Appendix.

- We allocated more spaces for the figures for easier reading and recalled our theoretical predictions for a better comparison between the theoretical and experimental results in the captions of figures.

- Due to space consideration, we shortened the Related works section in the main paper and moved the full Related works section into Appendix B. We also added more discussion about relevant literature in this section.

---

### Author Response · Authors · 2023-11-21
**Any Questions from the Reviewers before the Rebuttal/Discussion Period Ends?**

Dear reviewers,

We would like to thank all reviewers again for your thoughtful reviews and valuable feedback. We have updated our manuscript and added replies to your comments and questions with our latest results. We have summarized the changes we made in the manuscript in the Summary of Revision below.

We would appreciate it if you could let us know if there are additional questions or concerns about our revision and rebuttal.

We would be happy to do any follow-up discussion or address any additional comments.

Best regards,

Authors

---

### Meta-Review · Area_Chair_CYUu · 2023-12-14

**Metareview:**

This paper extends prior theoretical analysis of posterior collapse in simple linear VAEs to more complex VAE variants: linear conditional VAEs and Markovian hierarchal VAEs. Experiment are also provided and their results align with the theoretical analysis. Reviewers are generally in favor of acceptance. Reviewers valued both the rigor of the theoretical analysis and the extension of this type of analysis to more commonly used VAE variants. The main weakness brought up by reviewers was related to novelty: how different are the takeaways of this work compared to the original theoretical results for linear VAEs they were based on? The authors, in rebuttal and in revision, clarified their contributions and how these results make a step forward towards future analysis of non-linear models. Reviewers were mostly satisfied with the improvements made to the draft.

**Justification For Why Not Higher Score:**

While the theoretical results here are valuable, they are still limited to restricted model variants like linear conditional VAEs.

**Justification For Why Not Lower Score:**

The theoretical analysis provided is rigorous, and makes a meaningful step towards broader theoretical analysis of more complex model variants.

---

### Decision · Program_Chairs · 2024-01-16

Accept (poster)